# Intrinsic electrical activity drives small-cell lung cancer progression

Paola Peinado[1,11], Marco Stazi[1,11], Claudio Ballabio[1,11], Michael-Bogdan Margineanu[1], Zhaoqi Li[2], Caterina I. Colón[2], Min-Shu Hsieh[3], Shreoshi Pal Choudhuri[4,5], Victor Stastny[4,5], Seth Hamilton[4,5], Alix Le Marois[6], Jodie Collingridge[7], Linus Conrad[7], Yinxing Chen[2], Sheng Rong Ng[2], Margaret Magendantz[2], Arjun Bhutkar[2], Jin-Shing Chen[8], Erik Sahai[6], Benjamin J. Drapkin[4,5], Tyler Jacks[2], Matthew G. Vander Heiden[2,9], Maksym V. Kopanitsa[1,10], Hugh P. C. Robinson[1,7] & Leanne Li[1,2✉]

Elevated or ectopic expression of neuronal receptors promotes tumour progression in many cancer types[1,2]; neuroendocrine (NE) transformation of adenocarcinomas has also been associated with increased aggressiveness[3]. Whether the defining neuronal feature, namely electrical excitability, exists in cancer cells and impacts cancer progression remains mostly unexplored. Small-cell lung cancer (SCLC) is an archetypal example of a highly aggressive NE cancer and comprises two major distinct subpopulations: NE cells and non-NE cells[4,5]. Here we show that NE cells, but not non-NE cells, are excitable, and their action potential firing directly promotes SCLC malignancy. However, the resultant high ATP demand leads to an unusual dependency on oxidative phosphorylation in NE cells. This finding contrasts with the properties of most cancer cells reported in the literature, which are non-excitable and rely heavily on aerobic glycolysis. Additionally, we found that non-NE cells metabolically support NE cells, a process akin to the astrocyte–neuron metabolite shuttle[6]. Finally, we observed drastic changes in the innervation landscape during SCLC progression, which coincided with increased intratumoural heterogeneity and elevated neuronal features in SCLC cells, suggesting an induction of a tumour-autonomous vicious cycle, driven by cancer cell-intrinsic electrical activity, which confers long-term tumorigenic capability and metastatic potential.

Small-cell lung cancer (SCLC) is one of the most aggressive cancer types. Two-thirds of patients with SCLC present distant metastasis at initial diagnosis, and the median survival of these patients is only slightly over half a year[7]. SCLC is highly heterogeneous; such heterogeneity was first observed in cultured human SCLC (hSCLC) cell lines[8], of which more than 70% show classical neuroendocrine (NE) features and are hence termed the classic subtype, whereas the remaining variant subtype exhibits relatively lower expression of NE markers. Recent studies revealed that the classic subtype is driven by high ASCL1 expression (hence renamed as SCLC-A), whereas the variant subtype is driven by high NEUROD1 expression (renamed as SCLC-N), both broadly defined as NE subtypes, contrasting with two non-NE subtypes further identified as SCLC-P and SCLC-Y[9] (Fig. 1a).

Regardless of the molecular subtypes, hSCLC demonstrates almost universal inactivating mutations of both tumour suppressor genes *TP53* and *RB1* (ref. 10), based on which several genetically engineered mouse models (GEMMs) of SCLC have been developed: the PR (*Trp53*$^{-/-}$ and

*Rb1*$^{-/-}$)[11], PRP130 (*Trp53*$^{-/-}$, *Rb1*$^{-/-}$ and *Rbl2*$^{-/-}$)[12] and PRPTEN (*Trp53*$^{-/-}$, *Rb1*$^{-/-}$ and *Pten*$^{-/-}$)[13] models represent the classic SCLC; the PRM model (*Trp53*$^{-/-}$, *Rb1*$^{-/-}$ and *Myc*$^{T58A}$) recapitulates the variant SCLC[14] (Fig. 1a). Studies in the classic GEMMs and GEMM-derived cell lines revealed that both NE (mainly corresponds to the SCLC-A subtype) and non-NE (comparable to the SCLC-Y subtype[15]) subpopulations could arise from the same tumour and even the same clone[4]. These studies provided the first experimental evidence of functional intratumoural heterogeneity (ITH) in SCLC. NE cells are the predominant metastatic cell type[5], whereas non-NE cells are thought to have a supportive role: co-culture with non-NE cells[4] or conditioned medium from non-NE cells[5] both promote NE cell proliferation, invasion and metastasis. Fibroblast growth factor 2 (FGF2) secreted by non-NE cells has been shown to mediate cooperativity[16], but additional mechanisms remain unknown. In hSCLC, single-cell RNA sequencing demonstrated an association between increased ITH and poorer prognosis and treatment resistance[17]; yet, experimental validation of functional ITH in human samples is lacking.

[1]Cancer Neuroscience Laboratory, Francis Crick Institute, London, UK. [2]Koch Institute of Integrative Cancer Research and Department of Biology, Massachusetts Institute of Technology, Cambridge, MA, USA. [3]Department of Pathology, National Taiwan University Hospital, Taipei, Taiwan. [4]Hamon Center for Therapeutic Oncology Research, University of Texas Southwestern Medical Center, Dallas, TX, USA. [5]Department of Internal Medicine and Simmons Comprehensive Cancer Center, University of Texas, Southwestern Medical Center, Dallas, TX, USA. [6]Tumour Cell Biology Laboratory, Francis Crick Institute, London, UK. [7]Department of Physiology, Development and Neuroscience, University of Cambridge, Cambridge, UK. [8]Division of Thoracic Surgery, Department of Surgery, National Taiwan University Hospital and National Taiwan University College of Medicine, Taipei, Taiwan. [9]Dana-Farber Cancer Institute, Boston, MA, USA. [10]Present address: Charles River Discovery Services, Portishead, UK. [11]These authors contributed equally: Paola Peinado, Marco Stazi, Claudio Ballabio. ✉e-mail: leanne.li@crick.ac.uk

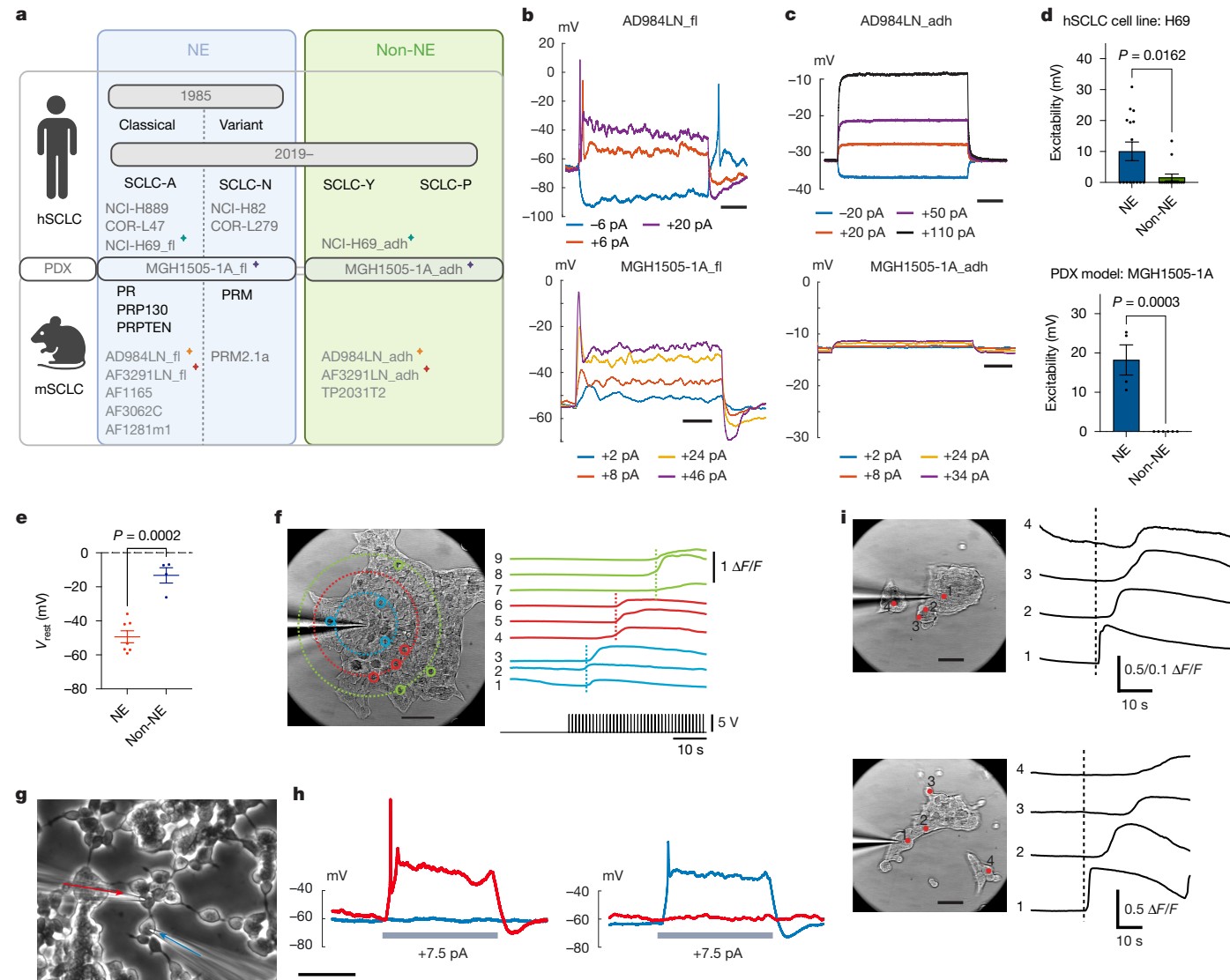

**Fig. 1 | Electrically active NE cells generate spontaneous and evoked calcium waves. a**, SCLC classification and models used in this study. Matched-colour stars depict paired cell lines/PDX models originally derived from the same tumours. **b,c**, Patch-clamp recordings in paired NE (**b**) and non-NE (**c**) mSCLC cell lines (top) or PDX models (bottom). Voltage responses to current steps of several amplitudes, as indicated. Graphs shown representative of $n = 9$ (MGH1505-1A_fl), 6 (MGH1505-1A_adh), 30 (AD984LN_fl) and 6 (AD984LN_adh) cells. **d**, Excitability (Methods) of paired NE and non-NE cells originally derived from the same hSCLC parental line (NCI-H69) and PDX model (MGH1505-1A). NCI-H69: $n = 15$ NE and 14 non-NE cells; MGH1505-1A: $n = 4$ NE and 6 non-NE cells. **e**, Resting membrane potentials ($V_{rest}$) of SCLC cells immediately following breakthrough into the whole-cell mode in current clamp. The average $V_{rest}$ values for each cell line are indicated as individual points; $n = 7$ NE cell lines and 4 non-NE cell lines (Extended Data Fig. 2j). **f**, Evoked, propagating calcium waves in GCaMP6m-expressing NE cells (AD984LN_fl). A patch pipette filled with KCl solution was pressed against a cell membrane within a cluster (left), and a train of extracellular voltage-stimulating pulses was applied, initiating a slowly propagating wave of intracellular calcium (right). Representative of $n = 7$ separate cultures. **g,h**, Simultaneous, paired patch-clamp recording in adjacent NE cells (**g**) did not detect synaptic-dependent currents or gap junction coupling (**h**). Representative of $n = 6$ pairs of AF3062C cells and 4 pairs of AF1165 cells. **i**, Propagating waves can be initiated by extracellular voltage-stimulating pulses given through patch pipette to the first cell (1) and traverse cell-free regions in the culture. Representative of $n = 5$ separate cultures. Mean ± s.e.m. shown in all graphs. Two-tailed unpaired $t$-test applied in **d** and **e**. Scale bars, 200 ms (**b,c**), 50 μm (**f,i**), 40 μm (**g**), 500 ms (**h**). Schematic in **a** was created using BioRender (https://biorender.com).

The predominant cells of origin of SCLC are thought to be the rare pulmonary neuroendocrine cells (PNECs), which cluster to form neuroendocrine bodies (NEBs)[11,18]. Interestingly, PNECs are electrically active[19,20]; electrophysiological studies also demonstrated excitability in a handful of hSCLC cells[21–23]. Whether the SCLC heterogeneity is also reflected in their electrophysiological properties has not been explored. Notably, neuronal features are proposed to underlie the metastatic potential of SCLC[22]: liver metastasis has been associated with the induction of a neuronal transcription programme in SCLC[24]; axon-like protrusions have been shown to promote SCLC invasion and metastasis[25]. However, it remains unknown whether the electrical activity per se directly impacts cancer progression. We performed a detailed electrophysiological characterization of a panel of human and mouse SCLC cell lines and patient-derived xenograft (PDX) models, together with metabolic and phenotypic analyses. We found that the electrical activity of NE cells directly drives their tumorigenic capacity; however, this requirement comes with a distinct metabolic vulnerability engendered by the increased ATP demand to maintain such an electrical activity. Finally, we found that non-NE cells provide metabolic support to NE cells to sustain their

electrical activity, analogous to the interplay between astrocytes and neurons[26].

## SCLC harbours electrically active NE cells

The crosstalk of NE and non-NE cells (both segregated from mouse SCLC (mSCLC) tumours according to their culture phenotypes: either forming floating aggregates, _fl, or being adherent, _adh, respectively) enhances the metastatic potential in classic SCLC GEMMs[4,5]. To explore the underlying mechanisms, we derived a panel of cell lines from GEMMs of classic SCLC, including pairs of NE and non-NE cells originally derived from the same tumour, and a pair of NE and non-NE lines from the same PDX model[27] (Fig. 1a and Extended Data Fig. 1). RNA sequencing analysis revealed distinct gene expression signatures segregating NE and non-NE cell lines (Extended Data Fig. 2a,b). Gene set enrichment analysis (GSEA) confirmed striking similarity with previous reports from mSCLC cells[5] (Extended Data Fig. 2c) and demonstrated that NE cells were enriched for a transcriptomic signature of synaptic signalling, whereas non-NE cells were enriched for an extracellular matrix signature (Extended Data Fig. 2d).

Despite the reported resemblance of NE cells to neurons and of non-NE cells to astrocytes at the transcriptomic level[5], whether such gene expression signatures translate into functional electrophysiological phenotypes has not been assessed. Patch-clamp electrophysiological recordings showed that NE cells, similar to neurons (Extended Data Fig. 2e), were excitable and capable of firing action potentials, unlike non-NE cells (and non-excitable astrocytes) (Fig. 1b–d and Extended Data Fig. 2f,g). In voltage-clamp recordings, only NE cells demonstrated prominent sodium and potassium voltage-gated channel currents necessary for action potential firing (Extended Data Fig. 2h,i). This voltage-gated sodium current mediating action potential generation in NE cells could be abolished by application of a highly specific, potent Na+ channel blocker, tetrodotoxin (TTX) (Extended Data Fig. 2i). The electrophysiological distinction between NE and non-NE cells was consistently observed across all mouse and human cell lines examined: regardless of the origin, all NE cell lines tested possessed more negative resting membrane potentials ($V_{rest}$), typical of neurons, whereas non-NE cell lines had a more depolarized $V_{rest}$, comparable to that reported for many other non-excitable cancer cells[28] (Fig. 1e and Extended Data Fig. 2j).

In neurons, membrane depolarization leads to calcium influx; therefore, calcium activity is often used as a surrogate marker of neuronal activity. Simultaneous patch-clamp recording paired with calcium imaging confirmed that large calcium transients in NE cells corresponded to individual action potentials (Extended Data Fig. 2k,l). Both spontaneous and evoked electrical activity and propagation of calcium waves in cultured NE cells could be observed by expression of a genetically encoded calcium reporter GCaMP6m[29] or non-invasive cell-attached patch-clamp recordings (Fig. 1f, Extended Data Fig. 2m–o and Supplementary Videos 1–3).

To assess whether the communication between NE cells was synapse-dependent, we performed paired patch-clamp recording by stimulating one NE cell and recording from an adjacent, physically contacting NE cell (Fig. 1g). Direct synaptic transmission would give rise to short-latency postsynaptic potentials and currents in the postsynaptic cell, but these were not observed (Fig. 1h). Notably, calcium waves could propagate across cell-free gaps (Fig. 1i); moreover, direct depolarization of NE cells through patch clamp increased membrane capacitance (Extended Data Fig. 2p), suggesting an increase of the plasma membrane area resulting from vesicular secretion (Extended Data Fig. 2q), consistent with the activity-dependent vesicular release of neurotransmitters typical of NE cells[30]. Collectively, these observations argue that diffusible factors are responsible for the propagation of calcium waves in SCLC NE cells, distinct from the mechanism of wave propagation mediated by tumour microtubes and gap junctions reported in glioma[31,32].

## Cholinergic signals alter SCLC activity

We took a candidate approach and focused on acetylcholine (ACh) as a possible diffusible factor that could trigger the electrical activity of NE cells because the cholinergic signalling pathway has been extensively characterized and shown to be crucially involved in SCLC tumorigenesis[33,34]. Both mouse and human SCLC NE cells express ionotropic, nicotinic cholinergic receptors (nAChR) (Fig. 2a and Extended Data Fig. 3a–c); a cholinergic agonist carbachol (CCh) elicited depolarization and widespread calcium activity in NE cell cultures (Fig. 2b, Extended Data Fig. 3d–f and Supplementary Video 4), and the evoked single-channel openings demonstrated amplitudes and lifetimes consistent with those reported for nAChRs (Fig. 2b).

Interestingly, ACh is the major neurotransmitter of cholinergic parasympathetic autonomic neurons, the axons of which comprise the pulmonary vagus nerves and innervate NEBs[35,36] (Fig. 2c,d, Extended Data Fig. 4a,b and Supplementary Video 5), but it is not known whether SCLC is similarly innervated. Strikingly, both the overall innervation (identified by a pan-neuronal marker β3-tubulin) of hyperproliferative NEB or early SCLC lesions (Fig. 2e,f and Extended Data Fig. 4c–e) and its proportion of ACh-secreting, cholinergic axons (identified by the vesicular ACh transporter (VAChT)) (Fig. 2g–i and Extended Data Fig. 4f–i) dramatically increased after tumour induction. The nerve axons were not only detected at the tumour periphery but were tightly woven into the core of the lesion, where we also observed structures typical of terminal boutons, which might point towards synapse formation at nerve endings (Fig. 2j,k, Extended Data Fig. 4j,k and Supplementary Video 6). Unexpectedly, in advanced SCLC tumours, both the innervation and the percentage of cholinergic nerve axons decreased (Fig. 2e–i, Extended Data Fig. 4c–i and Supplementary Video 7); concomitantly, β3-tubulin became highly expressed by cancer cells (Fig. 2l). Notably, SCLC cells synthesized and secreted ACh[33,34] (Extended Data Fig. 5). To further assess calcium activity in autochthonous SCLC tumours, we generated the PRP130-Salsa6f (tdTomato-V5-GCaMP6f)[37] mice, which expressed both the tdTomato lineage marker and calcium reporter GCaMP6f in SCLC cells (Fig. 2m). Indeed, ex vivo imaging on fresh lung slices revealed propagating calcium waves in SCLC tumours (Fig. 2n and Supplementary Video 8).

## High ATP needs of NE cells rely on OXPHOS

Studies in neurons have suggested that housekeeping activities account for only around 25% of ATP utilization in these excitable cells[38]. This ATP demand is further exacerbated when neurons fire action potentials, dissipating ionic gradients that require subsequent restoration[38] (Extended Data Fig. 6a–c). A major source of cellular ATP production is glucose catabolism[39], which can generate ATP through oxygen-independent glycolysis and oxygen-dependent oxidative phosphorylation (OXPHOS). Under aerobic conditions, pyruvate can enter the mitochondria to be further catabolized in the tricarboxylic acid cycle, which is coupled to the mitochondrial electron transport chain (ETC) for ATP production, a process collectively referred to as OXPHOS, which can generate up to 36 molecules of ATP per molecule of glucose (Fig. 3a). Notably, aerobic glycolysis, or the Warburg effect, is commonly observed in cancer cells[40], whereby much of the imported glucose molecules in cells are secreted as lactate even under aerobic conditions. It has been proposed that these actively proliferating cancer cells would prioritize anabolic pathways for biomass accumulation over catabolic pathways for ATP production, and, thus, are less ATP-dependent[41]. We hypothesized that the ATP demand in electrically active NE cells may lead to a distinct metabolic phenotype with a higher dependence on ATP-efficient OXPHOS compared to the majority of other cancer cells, which are non-excitable.

Indeed, NE cells were significantly less glycolytic than either their non-NE counterparts or other non-excitable adenocarcinoma lines

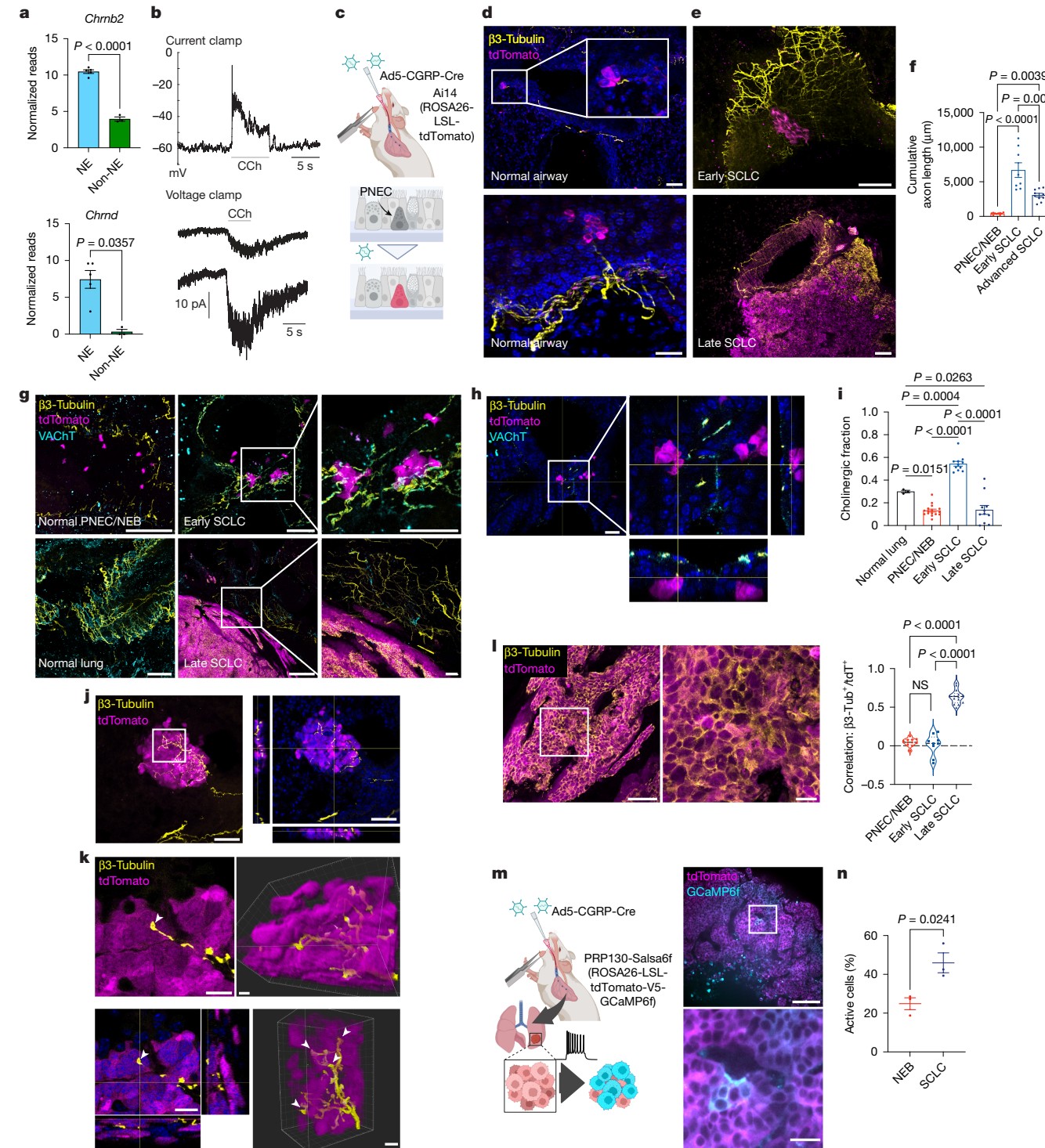

**Fig. 2 | Cholinergic signalling triggers depolarization and initiates calcium transients in NE cells. a**, nAChR subunit expressions in NE ($n = 5$) and non-NE ($n = 3$) cell lines. **b**, 100 μM CCh application to NE cells (AF1165). Top, depolarizing current-clamp responses, representative of $n = 10$ cells. Bottom, inward currents in two cells. Holding potential = −70 mV. Representative of $n = 5$ cells. **c–f**, In vivo labelling of calcitonin gene-related peptide (CGRP)-expressing cells (**c**). Representative images of PNEC/NEB ($n = 10$ from 3 mice) (**d**), early/small ($n = 7$ tumours per 3 mice) and advanced/large SCLC ($n = 10$ tumours per 4 mice) (**e**) with quantifications (**f**). **g–i**, Co-staining of β3-tubulin and VAChT in PNEC/NEB ($n = 16$ from 3 mice), normal lung ($n = 3$ mice) and early-stage ($n = 10$ tumours per 3 mice) and late-stage ($n = 10$ tumours per 4 mice) SCLC. Representative images (**g**) and orthogonal projection in early SCLC (**h**). Nuclei: DAPI (blue). Quantification of the percentage of VAChT+ fibres (**i**). **j,k**, Axons extending into

the core of early/small SCLC (**j**). Arrowheads indicate terminal buttons of nerve ending (**k**). Representative of $n = 10$ tumours per 3 mice. **l**, β3-Tubulin staining in an advanced SCLC tumour (left) and quantification of β3-tubulin+ in tdTomato+ PNEC/NEB ($n = 11$ from 3 mice), early ($n = 9$ tumours per 5 mice) and advanced ($n = 14$ tumours per 5 mice) SCLC cells (right). **m**, Ex vivo imaging of GCaMP6f-expressing mSCLC. **n**, Quantification of **m**. $n = 11$ PNECs/NEBs and 3 SCLC tumours (3 mice per group). Mean ± s.e.m. shown in all graphs. Two-tailed unpaired $t$-test applied in **a** (top) and **n**. Mann–Whitney test applied in **a** (bottom). Ordinary one-way analysis of variance (ANOVA) applied in **f**, **i** and **l**. Scale bars, 8 μm (**k** (right)), 10 μm (**l** (right), **k** (left)), 20 μm (**d**, **m** (bottom), **g** (right)), 30 μm (**h**), 50 μm (**d**, **e** (top), **l** (left), **j**), 100 μm (**e** (bottom), **g** (left and middle), **m** (top)). NS, not significant. Schematics in **c** and **m** were created using BioRender (https://biorender.com).

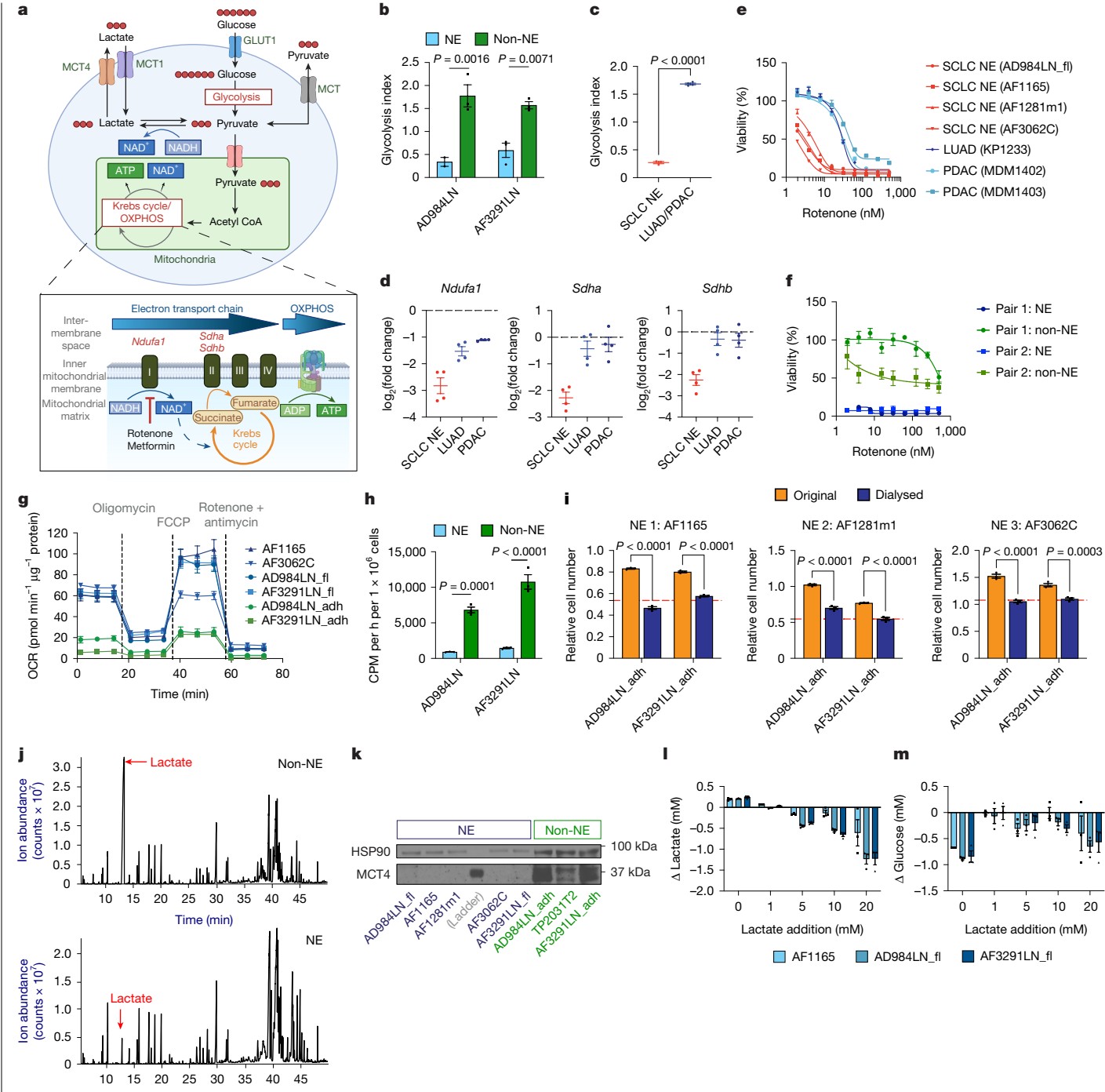

**Fig. 3 | High ATP demands of NE cells increase OXPHOS dependency.**
**a**, Glucose catabolism and OXPHOS. **b,c**, Glycolysis index (lactate secretion/glucose consumption) in paired SCLC NE and non-NE cell lines ($n = 2$ pairs; 3 replicates per cell line) (**b**) and in SCLC NE ($n = 4$) and LUAD/PDAC ($n = 4$) cell lines (**c**) (Extended Data Fig. 7a). Cells cultured in DMEM with 2% fetal bovine serum (FBS). **d**, Three genes encoding mitochondrial-related proteins (see Methods for the selection criteria) show SCLC-preferential vulnerabilities ($n = 4$ cell lines per group). **e,f**, Rotenone sensitivity (CellTiter-Glo (CTG) assay) of SCLC NE ($n = 4$) and LUAD/PDAC ($n = 3$) cell lines (**e**), as well as of paired SCLC NE and non-NE cell lines ($n = 2$ pairs; pair 1: AD984LN and pair 2: AF3291LN) (**f**) cultured in DMEM without pyruvate; $n = 3–4$ technical replicates per line. **g**, OCR in NE ($n = 4$) and non-NE ($n = 2$) SCLC cell lines; $n = 6$ technical replicates per line. **h**, Radioactive 2-deoxyglucose uptake assay ($n = 2$ pairs of SCLC NE and non-NE cell lines; 3 technical replicates per condition). **i**, Sulforhodamine B (SRB) assay of three independent NE cell lines in different conditioned media: NE conditioned media and two independent non-NE conditioned media with and without dialysis. Dotted red lines: cell number in NE conditioned media; $n = 3$ cell lines; 4 technical replicates per line. **j**, Representative gas chromatography–mass spectrometry results of conditioned media from paired NE and non-NE cell lines; $n = 3$ technical replicates per line. **k**, Western blot for MCT4 in NE ($n = 5$) and non-NE ($n = 3$) cell lines. **l,m**, Changes in lactate (**l**) and glucose (**m**) concentrations in media after incubation of SCLC NE cells with different concentrations of lactate; $n = 3$ technical replicates. Mean ± s.e.m. shown in all graphs. Two-way ANOVA; Sidak's multiple comparison test applied in **b**, **h** and **i**. Two-tailed unpaired $t$-test applied in **c**. Schematic in **a** was created using BioRender (https://biorender.com).

(Fig. 3b,c and Extended Data Figs. 6d and 7a). Coincidentally, SCLC NE-preferential vulnerability has been identified from a previous cross-cancer type CRISPR screen[42], in which three out of the eight top candidate genes encode ETC complex proteins: *Ndufa1*, *Sdha* and *Sdhb* (Fig. 3d), rendering ETC the most prominent hallmark of SCLC NE cell-preferential vulnerability (Extended Data Fig. 7b). On the other hand, SCLC NE cells were less sensitive to knockout of glycolysis pathway genes compared to lung adenocarcinoma (LUAD) and pancreatic ductal adenocarcinoma (PDAC) (Extended Data Fig. 7c). Pharmacological validation with a mitochondrial complex I inhibitor rotenone confirmed higher sensitivity in SCLC NE cells compared to non-excitable cells (Fig. 3e,f).

Notably, previous studies demonstrated that the sensitivity of adenocarcinoma cells to mitochondrial complex I inhibitors stems mainly from the disruption of NAD$^+$ regeneration[43] instead of ATP depletion. Further, it was proposed that the proliferation of some cancer cells is limited by NAD$^+$ availability more than ATP availability, such that complex I activity becomes relatively dispensable if NAD$^+$ can be regenerated through orthogonal pathways[41], including exogenously supplied pyruvate, which could regenerate NAD$^+$ through LDH-dependent lactate production[43] (Fig. 3a), as well as ectopic expression of the NADH oxidase from *Lactobacillus brevis* (LbNOX)[44]. Consistent with previous reports[41], pyruvate reduced the sensitivity of non-excitable cell lines to complex I inhibition; however, by sharp contrast, neither pyruvate supplement nor LbNOX expression rescued the high sensitivity of SCLC NE lines to rotenone (Extended Data Fig. 7d–g). These data suggest that the mechanism of sensitivity to ETC inhibition in SCLC NE cells was distinct from what was found in non-excitable cancer cells. Indeed, NE cells showed higher baseline and ATP production-coupled oxygen consumption rates (OCRs) compared to non-NE cells (Fig. 3g and Extended Data Fig. 7h–j), suggesting a higher ATP demand in NE cells.

## Metabolic shuttle supports NE cells

Surprisingly, the glucose uptake rate of NE cells was significantly lower compared to that of SCLC non-NE, LUAD and PDAC (LUAD/PDAC) cells (Fig. 3h and Extended Data Fig. 7k), which led us to query if other metabolites served as alternative fuels for ATP production in NE cells. Because non-NE conditioned medium was sufficient to promote NE cell malignancy[4,5,16] (Extended Data Fig. 7l–n), we assessed whether metabolite(s) from non-NE cells may underlie such cooperativity. Indeed, the pro-proliferation effect of non-NE conditioned medium was diminished by dialysis to remove metabolites (Fig. 3i). Mass spectrometry identified lactate and pyruvate to be differentially secreted in the conditioned medium (Fig. 3j and Extended Data Fig. 7o). In the central nervous system, monocarboxylate transporter 4 (MCT4, encoded by *Slc16a3*), a major protein involved in lactate export, is exclusively expressed by astrocytes[45], and the astrocyte-to-neuron lactate shuttle (ANLS) has been proposed to have a major role in neuronal energy supply[6], especially during elevated activity[26]. In analogy to the ANLS hypothesis, the astrocyte-like HES1$^+$ non-NE cells[5] expressed high MCT4 (Fig. 3k), whereas the neuron-like NE cell lines expressed MCT1 (Extended Data Fig. 7p), another monocarboxylate transporter proposed to be important for lactate import. NE cells started to consume lactate when cultured with concentrations of as low as 5 mM exogenously supplied lactate (Fig. 3l), which was accompanied by decreased glucose consumption (Fig. 3m). By contrast, non-NE cells continued to secrete lactate even when incubated with 20 mM lactate (Extended Data Fig. 7q).

## Non-NE cells fuel NE electrical activity

We hypothesized that metabolite shuttle sustains the high ATP-demanding electrical activity in NE cells (Fig. 4a). In neurons, the ATP demand for electrical activity could be broadly classified into two major categories: activity-dependent (including action potential firing and synaptic transmission) and $V_{rest}$ maintenance[38] (Extended Data Fig. 6c). Animals accommodate changes in energy supply during starvation by turning on various energy-saving adaptations in the brain and reducing non-essential neural functions, including ion channel conductance, spontaneous neuronal spiking and long-term memory formation[46]. Therefore, changes in the electrophysiological properties could serve as surrogate markers of the cellular energetic status in electrically active cells.

We first examined the activity changes by calcium imaging. Indeed, co-culturing NE cells with non-NE cells and non-NE conditioned medium increased the calcium activity of NE cells (Fig. 4b,c and Extended Data Fig. 8a,b), which could be suppressed by an MCT4 inhibitor, diclofenac (Fig. 4d and Extended Data Fig. 8c). Next, patch-clamp recordings were performed to characterize the effect of metabolites on the $V_{rest}$ of NE cells, which is predominantly determined by ionic conductances ($g_{in}$) involving Na$^+$ and K$^+$ ions. A biophysical model can estimate the energetic cost of maintaining $V_{rest}$ (Extended Data Figs. 6a and 8d and Methods). NE cells were cultured in a serum-free solution (starvation), with 5 mM glucose, 10 mM lactate or 2 mM pyruvate for different durations. NE cells trended towards decreased $g_{in}$ while maintaining their $V_{rest}$ in response to short-term starvation (less than 2 h), but the cells became depolarized after overnight starvation, even when cultured with glucose. Neither of these phenotypes was observed in cells cultured with either lactate or pyruvate (Fig. 4e and Extended Data Fig. 8e,f). Applying the biophysical model, the cellular ATP status could be inferred from the energetic cost of maintaining the $V_{rest}$ at their measured $g_{in}$, under the assumption of maintenance of normal sodium and potassium ionic gradients. As expected, starvation resulted in a gradual decline in the estimated cellular ATP cost over time; the trend was less obvious when cells were cultured with glucose, but lactate alone was sufficient to maintain the high-ATP-demanding cell status throughout the experiments (Fig. 4f). Importantly, overnight incubation with an MCT1 inhibitor SR-13800 led to depolarized $V_{rest}$ and decreased excitability in NE cells in the presence of lactate (Fig. 4g,h).

We further assessed the metabolite shuttle hypothesis in vivo. To specifically label NE cells, we identified a neural lineage marker SOX1 (ref. 47) to be selectively expressed in murine NE cell lines, unlike HES1, which is a known marker of non-NE cells[5] (Fig. 4i). High *SOX1* expression was also characteristic of the SCLC-A subtype in human lines (Extended Data Fig. 8g). In PDX models[27], *SOX1* expression correlated with the NE scores (Extended Data Fig. 8h), in contrast to the non-NE marker MCT4 (encoded by *SLC16A3*) (Extended Data Fig. 8i). ITH has been shown to gradually appear during mSCLC progression[48]. Indeed, in the PRP130 GEMM, we also found that both SOX1 (marking NE cells) and MCT4 (marking non-NE cells) were completely absent in PNEC and NEB (PNEC/NEB) but appeared in mSCLC tumours (Fig. 4j), and they were expressed by completely distinct subpopulations in both mSCLC and hSCLC tumours (Fig. 4k and Extended Data Fig. 8j). Ex vivo calcium imaging (Fig. 2m) revealed that diclofenac indeed reduced calcium transients in PRP130-Salsa6f SCLC tumours (Fig. 4l).

## Electrical activity boosts SCLC progress

Interestingly, despite having the same driver mutations, in sharp contrast to NE cells, non-NE cells fail to establish liver metastases even in immunocompromised animals[4] (Extended Data Fig. 9a), which motivated us to investigate the link between electrical activity and metastatic potential.

In neurons, action potential firing leads to calcium influx, activating various calcium-sensitive downstream signalling pathways. Frequently, cAMP response element-binding protein (CREB) phosphorylation and increased expression of FOS are used as markers of

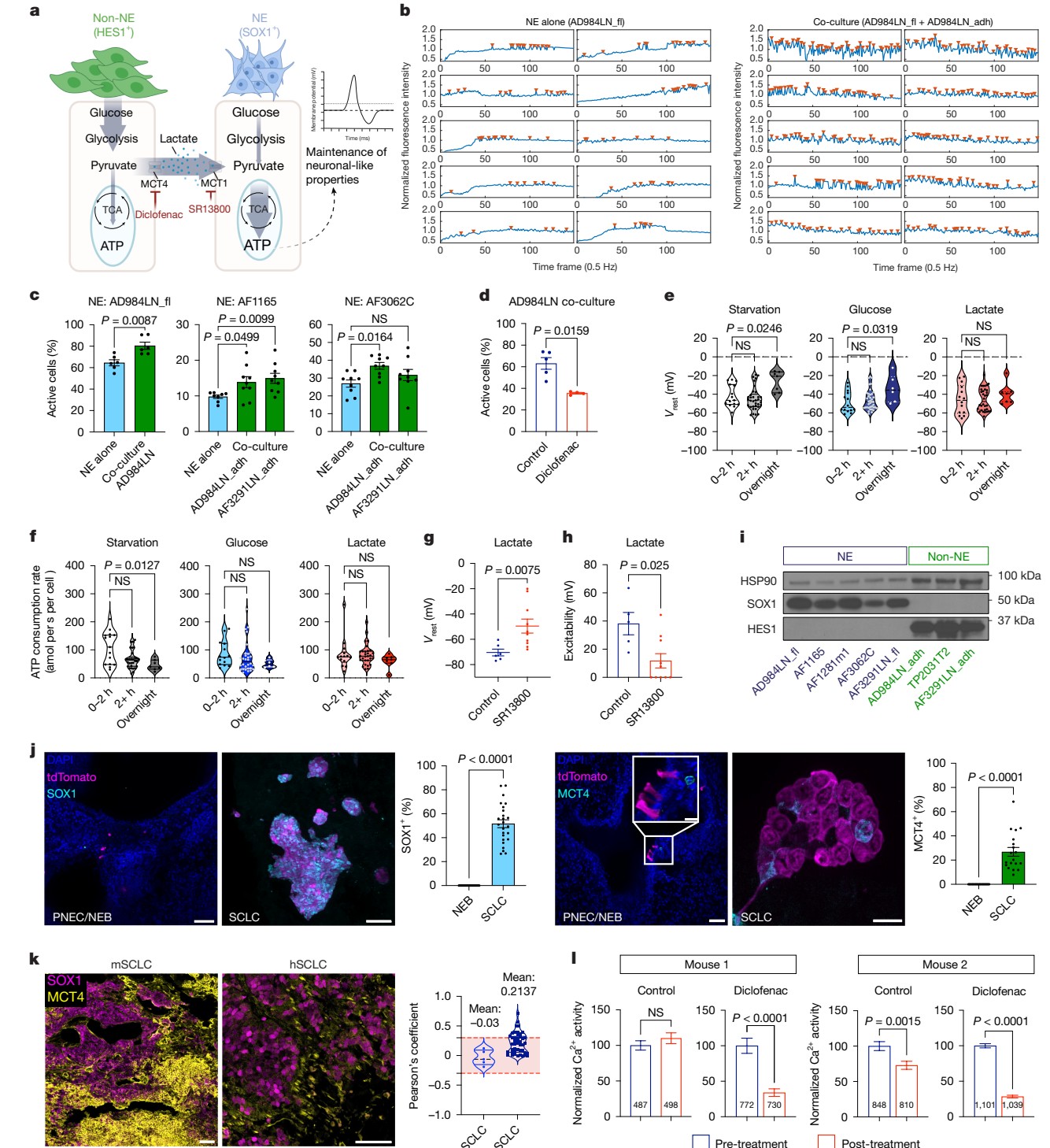

**Fig. 4 | Metabolite support from non-NE cells sustains the ATP demand of NE cells. a**, Proposed metabolite shuttle model. **b**–**d**, Calcium imaging of GCaMP6m⁺ NE cells, cultured alone or with non-NE cells. Calcium signal traces from the ten most active NE cells in each condition (arrowheads: calcium spikes) (**b**) and quantification of the percentage of NE cells with active calcium signalling (**c**) and when treated with diclofenac (0.5 mM) (**d**); $n = 6$–9 fields of view examined over 3 independent experiments, and each dot represents a field of view. **e**–**h**, Patch-clamp recording of NE cells (AF1165). Resting membrane potential ($V_{rest}$) (**e**) and predicted ATP consumption/demand (**f**) in different culture conditions: starvation ($n = 15$, 31 and 7 cells per time point); with 5 mM glucose ($n = 14$, 30 and 6 cells per time point); with 10 mM lactate ($n = 14$, 29 and 7 cells per time point). $V_{rest}$ (**g**) and excitability (**h**) when incubated overnight in lactate alone (**g**, $n = 5$ cells; **h**, $n = 6$ cells) or with SR-13800 (5 μM) ($n = 10$ cells). **i**, Western blot of HES1 and SOX1 in SCLC cell lines. **j**, Representative images and quantifications of SOX1 ($n = 25$ PNEC or SCLC, respectively; 6 mice per group) and MCT4 ($n = 19$ PNEC/SCLC; 6 mice per group) staining in mPNEC and small mSCLC tumours. **k**, Representative staining of SOX1 and MCT4 in mSCLC (PRP130) and hSCLC tumours; $n = 5$ mice (1 tumour per mouse) and 44 patients (1 tumour per patient). Pearson's coefficient from Extended Data Fig. 8j. Red area: negligible correlation. **l**, Calcium transient quantification in lung slices from PRP130-Salsa6f mice treated with either control (culture medium) or diclofenac (0.5 mM); $n$ cells per slice reported in graph from two independent imaging sessions. Mean ± s.e.m. shown in all graphs. Kruskal–Wallis test with Dunn's multiple comparison test applied in **c** (middle, right), **e** and **f**. Two-tailed Mann–Whitney test applied in **c** (left), **d**, **g**, **h** and **j**. Two-tailed unpaired $t$-test applied in **l**. Scale bars, 50 μm (**j** (SOX1 and MCT4 (left)), **k**), 10 μm (**j** (MCT4, right)). Schematic in **a** was created using BioRender (https://biorender.com).

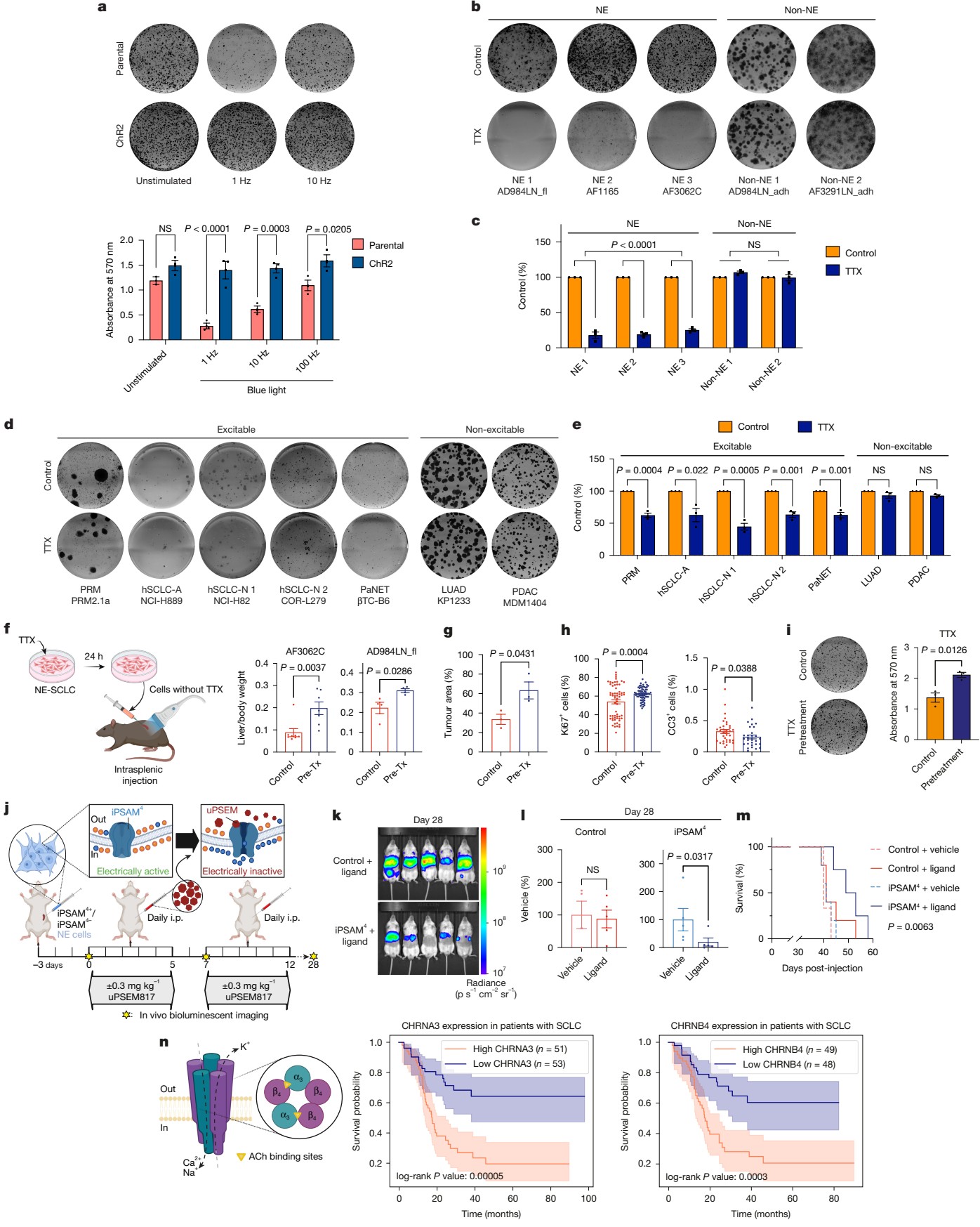

**Fig. 5 |** See next page for caption.

**Fig. 5 | Electrical activity promotes SCLC progression. a–e**, Colony formation assays. **a**, Representative images after blue light stimulation in parental or ChR2+ NE cells (AF3062C) (top) and their quantification (bottom). **b–e**, TTX treatment (1 µM) in mSCLC NE ($n$ = 3) and non-NE ($n$ = 2) cell lines (**b,c**) and other excitable ($n$ = 5) and non-excitable ($n$ = 2) cancer cell lines (**d,e**). Representative images (**b,d**) and quantification (**c,e**). **f–h**, Liver metastasis assays. **f**, Experimental design (left) and normalized liver weight. NE cells were either untreated or pretreated (pre-Tx) with 1 µM TTX. AF3062C: $n$ = 8 (untreated) and 7 (pre-Tx) mice. AD984LN_fl: $n$ = 4 mice per group. **g,h**, Quantification of tumour area (**g**), representative Ki67 (**h**, left) and cleaved caspase 3 staining (**h**, right) of the same liver lobes from the control ($n$ = 35 tumours per 4 mice) and pre-Tx ($n$ = 29 tumours per 3 mice). **i**, Colony formation in AF3062C NE cells either untreated (control) or pretreated for 24 h with TTX before seeding.

Representative images (left) with quantification (right). **j–m**, Chemogenetic suppression of NE electrical activity in liver metastasis assay. Mice were transplanted with AD984LN_fl cells ± iPSAM4, treated with vehicle or uPSEM817 ($n$ = 5 mice per group) (**j**). Bioluminescent imaging (**k,l**) and survival analysis (**m**). i.p., intraperitoneal. **n**, $(\alpha 3)_2 (\beta 4)_3$ nAChR subunits (left) are encoded by *CHRNA3* and *CHRNB4*; their impact on prognosis is shown in the Kaplan–Meier curves (right) ($n$ = 104 patients with SCLC from a published cohort[56]); 95% confidence intervals shown. For all colony formation assays, $n$ = 3 technical replicates, repeated at least twice per cell line in $n$ = 2–3 cell lines. Mean ± s.e.m. shown in all graphs. Two-way ANOVA and Sidak's multiple comparison tests applied in **a** and **c**. Two-tailed unpaired *t*-test applied in **e** and **g–i**. Two-tailed Mann–Whitney test applied in **f** and **l**. log-rank test applied in **m** and **n**. Schematics in **f**, **j** and **n** were created using BioRender (https://biorender.com).

---

neuronal activity (Extended Data Fig. 9b); these pathways are also known to promote neuronal survival[49]. Notably, CREB activity has been shown to maintain NE signature and promote SCLC progression in vivo[50], and *FOS* is a known oncogene that has been shown to promote cancer stemness[51]. Although both have been shown to promote cancer progression, including SCLC, it remains unexplored whether they can be induced by electrical activity in cancer cells, as occurs in neurons.

We first sought to increase the electrical activity of NE cells by optogenetics, using cation-permeable channelrhodopsin 2 (ChR2) (ref. 52). Blue light exposure induces conformational changes of ChR2, leading to cation influx and membrane depolarization, which further results in action potential firing only in excitable cells (Extended Data Fig. 9c–f); both CREB phosphorylation and FOS expression were increased after blue light exposure in ChR2+ NE cells (Extended Data Fig. 9g). Unexpectedly, growth reduction or even cell death was observed in parental NE lines after blue light stimulation alone (Extended Data Fig. 9h–j), which has been reported to be cytotoxic due to the induction of oxidative stress and upregulation of autophagy[53] (Extended Data Fig. 9k,l). Strikingly, ChR2 expression rescued these detrimental effects in NE cells (Fig. 5a and Extended Data Fig. 9m). By contrast, neither blue light exposure nor ChR2 expression had any effect on non-NE cells (Extended Data Fig. 9n,o).

Next, TTX was used to suppress action potential firing. Strikingly, TTX treatment did not alter NE cell viability but significantly hindered their long-term tumorigenic potential (Fig. 5b,c and Extended Data Fig. 9p). In an expanded cell line panel with distinct excitability, including different SCLC subtypes and different cancer types, TTX suppressed the long-term tumorigenic potential in all electrically active cells tested, regardless of their cancer types and tissues of origin; by stark contrast, non-excitable cells were unaffected by TTX treatment (Fig. 5d,e).

We further investigated whether the electrical activity of NE cells could impact their metastatic ability in vivo. TTX is a potent neuronal toxin and could be lethal in vivo; therefore, we pretreated cultured NE cells with TTX before intrasplenic transplantation (Fig. 5f). To our surprise, TTX pretreatment accelerated tumour progression in vivo (Fig. 5f–h and Extended Data Fig. 10a–c). It is well established in neurons that removal of TTX leads to rebound hyperactivity and increased neuronal firing frequency, a phenomenon known as homeostatic plasticity, which serves to maintain the average neuronal activity at a relatively stable long-term level[54]. Indeed, acute TTX washout in NE cells led to a dramatic induction of p-CREB (Extended Data Fig. 10d) and an overnight incubation with TTX, followed by washout before seeding increased colony formation (Fig. 5i), in contrast to the suppressive effect of continuous TTX treatment (Fig. 5b,c). To continuously suppress the electrical activity of NE cells in vivo, we used a chemogenetic method that uses ligand-activated ion channels called pharmacologically selective actuator module (PSAM4) (ref. 55). PSAM4-GlyR (inhibitory PSAM4 (iPSAM4)) mediates Cl− influx when bound to its ligand and, therefore, suppresses neuronal activity (Fig. 5j). Consistent with our hypothesis that electrical activity directly regulates the

metastatic ability of NE cells, iPSAM4 suppressed the progression of liver metastasis and prolonged survival (Fig. 5k–m and Extended Data Fig. 10e).

Finally, we investigated whether electrical activity could impact the prognosis of patients with SCLC. We began by assessing the levels of p-CREB as a surrogate marker for electrical activity in tumour samples through immunohistochemistry analysis (Extended Data Fig. 10f–h). In the autochthonous PRP130 animals, p-CREB was highly expressed in SCLC compared to cells in the normal adjacent airway (Extended Data Fig. 10f); p-CREB was also expressed in the PDX models (Extended Data Fig. 10g) and showed a trend towards higher expression in advanced hSCLC tumours (Extended Data Fig. 10h). The percentage of SOX1+ cells increased in advanced-stage hSCLC tumours, whereas that of MCT4+ cells remained unchanged (Extended Data Fig. 10i). Reanalysis of a published dataset[56] revealed high levels of SOX1 and CREB in hSCLC tumours compared to normal adjacent tissue (NAT), as well as a correlation between SOX1 and CREB in SCLC tumours but not in NAT (Extended Data Fig. 10j,k). Moreover, high expression of nAChR subunits, especially both subunits forming the $(\alpha 3)_2 (\beta 4)_3$ nAChR subtype (Fig. 5n and Extended Data Fig. 10l), but not the metabotropic muscarinic AChR (Extended Data Fig. 10m), was significantly associated with worse overall survival. Notably, both the IHC and survival analysis in hSCLC were subtype-agnostic, implicating a broader role of electric activity in promoting clinical SCLC progression. Our findings suggest that electrical activity directly activates the long-term tumorigenic potential and metastatic capability of SCLC NE cells. While cholinergic innervation may have an important role during the tumour initiation stage, NE cells become more neuronal as SCLC progresses, and non-NE cells metabolically support NE cells to promote their electrical activity, favouring a self-propagating loop of autocrine signalling and increased aggressiveness.

## Discussion

Neuronal receptors, ion channels and their related signalling pathways have been shown to impact cancer progression for decades; in addition, neuron-like features have been associated with a more aggressive phenotype in various cancers[57–59], even when these cancer cells, unlike neurons, do not fire action potentials. How bona fide neuron-like electrical activity in cancer cells impacts cancer progression remains an open question. We used TTX, optogenetics and chemogenetics to directly interrogate the role of electrical activity in cancer cells and provided functional evidence that action potential firing directly promotes SCLC aggressiveness. We also identified nAChR as an ion channel that participates in the initiation and propagation of the electrical activity in SCLC in response to ACh, although other neurotransmitters, including ATP, might also be important players. Notably, SCLC is almost invariably associated with smoking in patients[7], and tobacco nicotine is another potent agonist for nAChR. In addition, calcium activity in PNECs has recently been shown to be activated by various mechanical

and chemical stimuli[20], many of which become pronounced within the tumour microenvironment. Therefore, as SCLC progresses, multiple mechanisms may be co-opted to stimulate and sustain the intrinsic electrical activity of SCLC. Here we identified a metabolic support from non-NE cells to sustain the electrical activity of NE cells. Whether similar cooperativity exists between subpopulations with different electrophysiological profiles and promotes the progression of other NE cancers awaits future investigation. Alternatively, the metabolic support could also come from stromal cells within the tumour micro-environment. The ANLS theory led us to focus on lactate, although other metabolites, including pyruvate, may also have important roles in the cooperativity.

SCLC is disseminated at diagnosis, and this advanced presentation obscures key events in early tumorigenesis owing to a paucity of adequate clinical samples. Our data from different stages of mSCLC progression suggested that cholinergic innervation might be crucial for SCLC initiation but becomes dispensable in fully formed SCLC. This hypothesis is supported by evidence from an independent study, which shows that vagotomy before tumour initiation in the PRP130 model significantly suppressed SCLC development, whereas when performed at a late stage, it demonstrated an opposite trend to promote SCLC progression[60]. Concomitantly, increased neuronal markers, including SOX1 and β3-tubulin, as well as functional electrical activity are observed in advanced SCLC. These data collectively imply a switch from dependency on external stimuli to cancer-intrinsic amplification of malignancy.

Finally, our observation that electrical activity directly promotes long-term tumorigenic potential might be generally applicable to other electrically active cells. In the future, it will be interesting to determine whether non-excitable cancer cells also become excitable after acquiring an NE transcriptional signature and how neuronal mechanisms contribute to cancer progression in non-excitable cells. Nevertheless, SCLC may serve as a prototype for highly metastatic cancers. Deeper understanding of SCLC biology and what underlies its unusual aggressiveness might yield important insights beyond lung cancers or NE cancers, leading to new therapeutic interventions for highly aggressive cancers of diverse aetiologies.

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

## Methods

### Animal studies

$F_1$ mice used for intrasplenic injections of SCLC cells were obtained by crossing 129S6/SvEvTac males to C57BL/6J females imported, respectively, from Taconic (Germantown, NY, USA) and The Jackson Laboratory (JAX; Bar Harbor, ME, USA). Both male and female mice between 9 and 19 weeks of age were used, and age-, litter- and sex-matched mice were randomly assigned to experimental groups. NOD-SCID mice (Prkdc[scid]; JAX strain 001303) used for intrasplenic injection SCLC cells were obtained from The Jackson Laboratory. Cells expressing iPSAM[4] were injected in males between 7 and 11 weeks of age. Age- and litter-matched mice were randomly assigned to experimental groups. Paired NE and non-NE SCLC cells were injected in 8- to 12-week-old animals. Age-, litter- and sex-matched mice were randomly assigned to experimental groups. The animals were anaesthetized with a gas mixture of oxygen-enriched air and 3–3.5% isoflurane (Zoetis UK Ltd.), and an ultrasound-guided (Vevo 3100; FUJIFILM VisualSonics Inc.) transdermal intrasplenic injection of SCLC cells (50 µl of a $10^6$ cell per millilitre suspension in Dulbecco's PBS (DPBS)) was performed, as described previously[42]. For all intrasplenic injections of cancer cells with different treatments, the experiment was conducted by a researcher who was blind to the experimental groups.

For the transplantation of SCLC NE cells after TTX pretreatment, cells were treated with 1 µM TTX for 24 h before the injection. Liver was sampled 25 days (AF3062C) or 34 days (AD984LN_fl) after injection. Liver weight was normalized on the baseline body weight before injection. After weighing, the livers were fixed in 10% neutral buffered formalin (NBF) for 24–48 h.

Autochthonous SCLC tumours were obtained from mice harbouring $Trp53^{fl/fl}$, $Rb1^{fl/fl}$, $Rbl2^{fl/fl}$ and $Gt(ROSA)26Sor^{tm14(CAG-tdTomato)Hze}$ alleles (PRP130 model, as described previously[12]) or $Trp53^{fl/fl}$, $Rb1^{fl/fl}$, $Rbl2^{fl/fl}$ and $Gt(ROSA)26Sor^{tm1.1(CAG-tdTomato/GCaMP6f)Mdcah/J}$ alleles (PRP130-Salsa6f model). The PRP130 animals were on a mixed 129S5/C57BL/6J background. SCLC tumour formation was induced by intratracheal administration of Ad5-CGRP-Cre adenoviruses ($2 \times 10^8$ pfu per animal; VVC-Berns-1160; University of Iowa Vector Core; plasmid origin: A. Berns and K. Sutherland) according to the method described previously[61]. Tumour growth was monitored by regular computed tomography (CT) scans (Quantum GX2; PerkinElmer), starting from 5 months after virus administration. Animals that reached the humane end point (development of moderate breathing signs or weight loss below 15% of the maximum weight), typically in 7–10 months, were killed by an overdose of anaesthetic and processed for precision-cut lung slices or perfused transcardially with 20 ml of PBS containing 20 U ml⁻¹ of heparin (H3400; Sigma-Aldrich) and 20 ml of 10% NBF. Dissected lungs were incubated in 10% NBF for 24–48 h and then stored in a 0.02% solution of sodium azide in PBS at +4 °C until use.

SCLC tumour samples from the PRM model were obtained from mice harbouring $Trp53^{fl/fl}$, $Rb1^{fl/fl}$ and $Igs2^{tm1(CAG-Myc*T58A/luc)Wrey}$ alleles (JAX strain 029971) on a C57BL/6J background. SCLC tumour formation was induced by intratracheal administration of Ad5-CGRP-Cre adenoviruses ($1 \times 10^8$ pfu per animal; VVC-Berns-1160) at 11 weeks of age. Tumour growth was monitored by regular computed tomography scans (Quantum GX2), starting from 46 days after virus administration. Upon reaching humane end point at day 89 following virus administration, the animals were culled by cervical dislocation for tumour collection.

Pulmonary NE cells were labelled by intratracheal administration of Ad5-CGRP-Cre adenoviruses ($2 \times 10^8$ pfu per animal; VVC-Berns-1160) in 8- to 10-week-old $Gt(ROSA)26Sor^{tm14(CAG-tdTomato)Hze}$ homozygous mice (Ai14 allele; JAX strain 007908). Lungs were collected 1 week after adenovirus instillation, either for precision-cut lung slices or for immunofluorescence after transcardial perfusion.

The mice were group-housed (up to five mice per cage) at a specific pathogen-free facility at The Francis Crick Institute in individually ventilated cages (GM500; Tecniplast) at an ambient temperature of 22 ± 2 °C, relative humidity of 55 ± 10% and standard 12-h light/12-h dark cycles. The animals received standard rodent food (2018 Teklad Global; ENVIGO) and water ad libitum. Humane end points (moderate clinical signs, clinical signs of suffering/distress or loss of more than 15% body weight) for animals in tumour studies were not exceeded in any of the experiments. All procedures were conducted in accordance with the UK Animals (Scientific Procedures) Act 1986, approved by the Institutional Animal Welfare and Ethical Review Body (The Francis Crick Institute PPL Review Committee), conducted under the authority of the UK Home Office approved project licence PP4103600 and approved by the Massachusetts Institute of Technology (MIT) Institutional Animal Care and Use Committee.

### Cell culture

**Mouse SCLC lines.** The following cell lines were derived from GEMMs, as described previously[42]. In particular, AF1165 was derived from a primary tumour of a PR; $Rosa26^{LSL-Tom/+}$ mouse. AF3062C was derived from liver metastases of a male PR; $Rosa26^{LSL-Tom/+}$ mouse. AF1281m1 was derived from a relapse tumour after chemotherapy in a PR; $Rosa26^{LSL-Tom/+}$ mouse. TP2031T2 was derived from a primary PR tumour. AD984LN_fl/AD984LN_adh were derived from the same lymph node metastasis in a PR mouse[62]. AF3291LN_fl/AF3291LN_adh were also derived from the same lymph node metastasis in a PRPTEN mouse.

The PRM2.1a cell line was derived in-house from a primary tumour in a $Trp53^{fl/fl}$; Rb1$^{fl/fl}$; Myc$^{LSL-T58A/+}$ (PRM) mouse, as described previously[63]. Briefly, the primary tumour was dissected, cut into small pieces and incubated for 30 min at 37 °C in 6 ml digestion solution (10% TrypLE (12605010; Gibco), 1 mg ml⁻¹ of Collagenase IV (17104019; Gibco) and 1 mg ml⁻¹ of Dispase II (D4693; Sigma-Aldrich) in Hanks' balanced salt solution (HBSS)). The digestion reaction was quenched by adding 4 ml of ice-cold quenching medium (10% FBS (11320033; Gibco) and 18.75 µg ml⁻¹ of DNase I (DN25; Sigma-Aldrich) in DMEM). The cell suspension was passed several times through an 18G needle before being filtered with a 100-µm strainer. Cells were centrifuged at 800$g$ for 5 min, resuspended in 1× RBC Lysis Buffer (420301; BioLegend) and incubated for 3 min at 37 °C. After washing with ice-cold PBS, the cells were plated in a complete culture medium on tissue-treated culture vessels, selecting for cells growing as floating aggregates.

NE cells were maintained in DMEM-F12 with GlutaMAX (10565018; Gibco) with 1× NEAA (11140050; Gibco), 10% FBS (11320033; Gibco) and 100 U ml⁻¹ of penicillin–streptomycin (15140122; Gibco). NE cells were grown as floating aggregates or were allowed to adhere to culture vessels coated with 50 µg ml⁻¹ of growth-factor-reduced Matrigel (356231; Corning) or Cultrex BME (3432-010-01; Bio-Techne) in HBSS. NE SCLC cells were also grown 3D-embedded in a Matrigel drop for calcium imaging experiments. Non-NE cells were cultured in DMEM (10-013-CV; Corning) with 10% FBS, 2 mM GlutaMAX (35050038; Thermo Fisher Scientific) and 100 U ml⁻¹ of penicillin–streptomycin.

**Human SCLC lines.** NCI-H889, NCI-H82, COR-L47, COR-L279 (purchased from ATCC), NCI-H69_fl and NCI-H69_adh (a gift from J. Minna) cell lines were cultured in DMEM-F12 with GlutaMAX, 1× NEAA, 10% FBS and 100 U ml⁻¹ of penicillin–streptomycin.

**Mouse LUAD lines.** KP1233 and KP1234 were derived from the primary tumours of KP mice ($Kras^{LSL-G12D/+}$ and $Trp53^{fl/fl}$), as described previously[42] and cultured in DMEM (10-013-CV; Corning) with 10% FBS and 100 U ml⁻¹ of penicillin–streptomycin.

**Mouse pancreatic lines.** MDM1402 was previously derived from a primary tumour of a female KPC mouse ($Kras^{LSL-G12D/+}$, $Trp53^{LSL-R172H/+}$, $Rosa26^{LSL-Tom/+}$ and $Pdx1-cre^+$); MDM1403 was previously derived from a primary tumour of a male KPC mouse ($Kras^{LSL-G12D/+}$, $Trp53^{fl/+}$,

$Rosa26^{LSL-Tom/+}$ and $Pdx1$-$cre^+$)[42] and cultured in DMEM (10-013-CV; Corning) with 10% FBS and 100 U ml[−1] of penicillin–streptomycin.

Mouse βTC-B6 PaNET cell line was previously derived from a RIP1-Tag2 mouse model and cultured in DMEM (10-013-CV; Corning) with 10% FBS and 100 U ml[−1] of penicillin–streptomycin[1,64].

**Other cell lines.** HEK293T cells were supplied by the Cell Services facility at The Francis Crick Institute and cultured in DMEM (11995073; Gibco) with 10% FBS and 100 U ml[−1] of penicillin–streptomycin.

All the cell lines were regularly cultured under standard conditions (humidified 5% $CO_2$ atmosphere at 37 °C) and were routinely tested for mycoplasma contamination and authenticated by short tandem repeat (STR) profiling. For all conditioned medium experiments and metabolic assays comparing across different types of cancer cells, the same culture medium was used, mostly pyruvate-free DMEM (10-017-CV; Corning) with 2% or 10% dialysed FBS unless specified otherwise.

## MGH1505-1A cell line derivation
The PDX model MGH1505-1A was established from circulating tumour cells isolated from patient MG1505, a 59-year-old man with relapsed SCLC[27]. Two cell lines with distinct morphologies were derived from this PDX model over 260 days as follows (Extended Data Fig. 1). A fragment of an MGH1505-1A xenograft was implanted in the right flank of an NSG-GFP mouse (JAX strain 021937), in which all mouse cells express GFP. When the xenograft reached 1,500 mm[3], the NSG-GFP mouse was euthanized, and the xenograft was resected and manually dissociated in ice-cold PBS. Live-cell clusters were isolated through serial gravity sedimentation at room temperature in 15-ml conical tubes, with periodic evaluation of supernatant to identify fractions containing the highest ratio of live-cell clusters to cell debris. Red blood cells were lysed using ACK lysis buffer (A1049201; Gibco). The enriched cell clusters were grown in a modified HITES medium (DMEM-F12 with GlutaMAX, 5% FBS, 1× NEAA, 1× ITS-G supplement (Gibco, 41400045), 10 nM hydrocortisone (07925; STEMCELL Technologies), 10 nM β-estradiol (E2758; Sigma-Aldrich) and 100 U ml[−1] of penicillin–streptomycin), and the culture was monitored two to three times weekly. Adherent GFP[+] mouse fibroblasts began to proliferate within 3 days of dissociation, and the culture growth pattern was dynamic over the first 8 weeks, with initial suspension clusters transitioning to a mixture of tightly and loosely adherent tumour cells interspersed with GFP[+] murine cells with fibroblast morphology (Extended Data Fig. 1b). To deplete the murine cells, we leveraged the absence of functional *RB1* and consequent lack of CDK4/6 dependence that characterizes most SCLC, including MGH1505-1A. We treated the MGH1505-1A culture with the CDK4/6 inhibitor palbociclib (10 μM) (508548; Thermo Fisher Scientific) for 1 week, resulting in significant depletion of GFP[+] cells and a mixed culture of floating tumour clusters and adherent tumour cells. The adherent and suspension components were separated and cultured independently without palbociclib; however, residual GFP[+] cells were observed in both cultures after 4 weeks (Extended Data Fig. 1c,d), prompting an additional 4 weeks of palbociclib treatment. During this treatment, suspension clusters were dissociated by means of gentle pipetting, and adherent cells were dissociated using TrypLE digestion twice weekly to expose GFP[+] cells to the drug. Following palbociclib treatment, the suspension (MGH1505-1A_fl) and adherent (MGH1505-1A_adh) cultures were grown at low density and screened manually for GFP[+] cells two to three times weekly to confirm complete depletion of mouse cells (Extended Data Fig. 1e,f). The total culture time of 260 days exceeded 50 population doublings to establish both as cell lines with consistent and stable morphologies. The origins of both cell lines from patient MGH1505 were confirmed by STR profile comparison with patient germline genomic DNA.

## Molecular cloning and cell line engineering
**ChR2-expressing cell lines.** ChR2 coding sequence was PCR amplified from FCK-ChR2-GFP (Addgene plasmid 15814, a gift from E. Boyden)[65] and cloned into a piggyBac transfer plasmid under the constitutive CAG promoter in frame with a P2A-PuroR cassette[66] to generate the plasmid pPB CAG ChR2-P2A-PuroR. SCLC, LUAD and PDAC mouse cell lines were reverse-transfected using Lipofectamine 3000 (Invitrogen) or TransIT-LT1 (Mirus Bio) with pPB CAG ChR2-P2A-PuroR and a plasmid encoding for a hyperactive form of the piggyBac transposase (pCMV HAhyPBase, a gift from the Wellcome Sanger Institute) at a ratio 4:1. Stable ChR2-expressing cells were established after puromycin selection.

**GCaMP6m-expressing cell lines.** GCaMP6m coding sequence was PCR amplified from pGP-CMV-GCaMP6m (Addgene plasmid 40754, a gift from D. Kim and GENIE Project)[29] and cloned into a lentiviral transfer plasmid under the constitutive EFS promoter in frame with a P2A-PuroR cassette to generate the plasmid pLenti EFS GCaMP6m-P2A-PuroR. Lentiviral particles were generated and used to transduce mouse SCLC NE cell lines, as described below.

**LbNOX-expressing cells.** FLAG-tagged LbNOX coding sequence was PCR amplified from pUC57 LbNOX (Addgene plasmid 75285, a gift from V. Mootha)[44] and cloned into an all-in-one doxycycline inducible lentiviral vector to generate the pTL LbNOX-FLAG plasmid. The LbNOX-FLAG sequence was cloned under a tight TRE promoter in the pTL lentiviral backbone, which also contains a cassette expressing PuroR-P2A-rtTA Advanced under a constitutive EF1a promoter. Lentiviral particles were generated from pTL LbNOX-FLAG and used to transduce AD984LN_fl and AF3062C cells, as described below.

**iPSAM[4]-expressing cell lines.** The coding sequence of the inhibitory chemogenetic receptor PSAM[4]-GlyR (iPSAM[4]) was PCR amplified from pCAG PSAM[4] GlyR IRES EGFP (Addgene plasmid 119739, a gift from S. Sternson)[55] and cloned in frame with a cassette containing the T2A linker, firefly luciferase coding sequence linked through P2A to PuroR. Gibson assembly was used to generate the piggyBac transfer plasmid pPB CAG PSAM[4]-GlyR T2A Luc P2A PuroR. A control plasmid (pPB CAG Luc P2A PuroR) was assembled by cloning firefly luciferase coding sequence linked through P2A to PuroR in the same piggyBac backbone. AD984LN_fl cells were reverse transfected with either the iPSAM[4]-containing or control piggyBac transfer plasmid (together with pCMV HAhyPBase; 4:1 ratio) with TransIT-LT1 (Mirus Bio). Engineered cells were established after puromycin selection.

## Generation of lentiviral particles, titration and transduction
HEK293T cells were co-transfected with lentiviral packaging plasmids psPAX2 (Addgene plasmid 12260, a gift from D. Trono) and pMD2.G (Addgene plasmid 12259, a gift from D. Trono) and the appropriate transfer plasmid using the calcium phosphate method. Supernatants were harvested 60 h after transfection, filtered using 0.45-μm filters and stored at −80 °C. The viral titre was estimated using a lentiviral standard of known titre (IU ml[−1]) and the SYBR Green I-based PCR-enhanced reverse transcriptase assay (SG-PERT) method[67]. SCLC NE cells were transduced by resuspending them in a complete culture medium supplemented with the appropriate lentivirus (multiplicity of infection (MOI) = 10) and 10 μg ml[−1] of polybrene (Sigma-Aldrich, TR-1003-G). Cells–virus suspensions were plated in a coated well of a six-well plate and spun at 1,200g for 2 h at 30 °C. The medium was changed 24 h after transduction.

## In vivo chemogenetic treatment, bioluminescence imaging and survival analysis
Mice transplanted with iPSAM[4]-expressing or control cells were injected intraperitoneally with 0.3 mg kg[−1] of uPSEM817 tartrate (6866; Bio-Techne) in sterile saline or vehicle alone (daily for 5 days a week), starting from 3 days after cell transplantation. On the same day, tumour burden was monitored by in vivo bioluminescence imaging and followed up weekly. The animals were imaged 10 min after intraperitoneal

administration of 150 mg kg$^{-1}$ of D-luciferin (1-360223-200; Regis Technologies). The animals were anaesthetized with 2% isoflurane, and the bioluminescent signal was captured using the IVIS Spectrum system (PerkinElmer). Bioluminescent signals were quantified using the Living Image software (v.4.8; PerkinElmer): a 7-cm$^2$ square region of interest (ROI) was drawn on each mouse abdominal area, and the average radiance (p s$^{-1}$ cm$^{-2}$ sr$^{-1}$) was calculated. Background signal was removed from each measurement by subtracting the average value in a similar ROI placed on an empty area within the same field of view. Survival analysis was performed by calculating the lifespan after cell transplantation in days of every mouse in each experimental group. Mice that died due to causes unrelated to the study were censored in the analysis. Data were displayed using the Kaplan–Meier format, and the statistical significance of the results was tested using the log-rank (Mantel–Cox) test.

## Patch-clamp electrophysiology

Patch-clamp recordings were carried out by standard methods[68]. Cells were plated after passaging onto 35-mm tissue-culture-treated plastic Petri dishes and cultured for 1–4 days before recording. For NE cell lines, dishes were pre-coated with a 5% solution of Matrigel for 20–30 min to allow adhesion. Cells were visualized with phase-contrast microscopy (Olympus IX71) using 10× or 20× objectives.

Current-clamp and voltage-clamp recordings were carried out with an Axon MultiClamp 700B amplifier (Molecular Devices), and 16-bit waveforms were generated and sampled at 10–20 kHz using an X Series data acquisition interface (National Instruments), following low-pass analogue filtering (four-pole Bessel) at a 4-kHz cutoff and analysed with custom software in R or MATLAB. Before recording, the culture medium was exchanged for Ringer solution composed of 140 mM NaCl, 4 mM KCl, 1.4 mM CaCl$_2$, 1 mM MgCl$_2$ and 10 mM HEPES, balanced to pH 7.4 with NaOH. In most recordings, 5 mM glucose was further added but was omitted or replaced by lactate (10 mM) in some recordings, as described in the text.

Recordings were established using 1.5-mm-outer-diameter borosilicate glass capillary pipettes (Harvard Apparatus), fire-polished at the tip and filled with an intracellular perfusion solution composed of either 105 mM potassium gluconate, 30 mM KCl, 10 mM HEPES, 1 mM EGTA, 4 mM ATP·Mg, 0.3 mM GTP and 10 mM phosphocreatine·Na$_2$, balanced to pH 7.3 with KOH, or 125 mM potassium gluconate, 10 mM HEPES, 4 mM ATP·Mg, 0.3 mM GTP and 10 mM phosphocreatine·Na$_2$, balanced to pH 7.3 with KOH. Recordings were carried out at room temperature (23–25 °C).

Membrane potentials were adjusted for prenulling of a calculated liquid junction potential of 10 mV before seal formation, with the open pipette immersed in the bath solution. Pipettes had resistances of 5–10 MΩ. To estimate the unperturbed resting membrane potential, before significant dialysis of the intracellular compartment by pipette solution, current-clamp recordings were made immediately following the rupture of a cell-attached patch to establish whole-cell recording mode. This was usually carried out in current-clamp mode to allow time-resolved recording of the zero-current potential during rupture, and the initial value within 0–3 s was measured. If rupture was carried out in voltage clamp, the potential was measured in the current clamp within approximately 4 s. Cell input conductance and capacitance were evaluated from fits of the current-clamp responses to small (2–4 pA), hyperpolarizing step current stimuli. For voltage-clamp recordings, the built-in circuitry of the MultiClamp 700B was used to compensate for pipette and whole-cell capacitance and series resistance (80%).

To measure excitability, the overshoot or maximum positive difference between the peak voltage and steady-state depolarized voltage in a run was determined. If the cell shows a rebound spike, this is always the largest and the one used for the measurement. Excitability is defined as the maximum positive difference between the peak voltage and steady-state depolarized voltage in a sequence of progressively larger current step responses. Excitability is zero if there is no clear, consistent overshoot in more than one sweep in a run.

For pre-incubation in conditions of different energy supplies, the culture medium was exchanged for Ringer solution containing 0–10 mM of glucose or 5 mM lactate for periods from less than 2 h up to more than 16 h (overnight), and dishes were incubated at room temperature in a humidified chamber.

## Biophysical modelling of ATP consumption required for electrical activity

To estimate the energetic cost of maintaining the negative resting potential required to enable action potential generation, a simple model was used, which assumes that the membrane conductance at rest is composed solely of sodium-selective and potassium-selective fractions, which is reasonable because these contribute to the dominant electrochemical gradients at the resting potential. The model solves for the electrogenic sodium pump current, which equals the sum of inward sodium and outward potassium fluxes through the resting conductance of the membrane at steady state[69].

$$I_{pump} = g_{Na}(E_{Na} - V) + g_K(E_K - V)$$

where $V$ is the resting membrane potential, and and are the equivalent sodium-selective and potassium-selective fractions, respectively, of the total input leak conductance: $g_{in} = g_{Na} + g_K = 1/R_{in}$, which was measured from the response to small hyperpolarizing current steps in current-clamp mode. Nernst equilibrium potentials and were assumed to equal +53.8 and −108.4 mV, respectively, as calculated from the compositions of pipette and Ringer solutions, because in NE cells, there was typically a shift of only a few millivolts in the resting potential from the initial breakthrough potential, implying that the sodium and potassium contents of the unperturbed cytoplasm were close to those of the pipette solution.

Because the sodium pump transports three Na$^+$ ions out and two K$^+$ ions in for each ATP molecule hydrolysed,

$$I_{pump} = g_{Na}(E_{Na} - V)/3$$

From this, it can be shown that the inward sodium current at rest is given by

$$g_{Na}(E_{Na} - V) = \frac{3g_{in}(E_{Na} - V)(V - E_K)}{V + 2E_{Na} - 3E_K}$$

and the ATP consumption rate at rest for the cell is

$$\text{ATP rate} = \frac{g_{in}(E_{Na} - V)(V - E_K)}{F(V + 2E_{Na} - 3E_K)}$$

where $F$ is Faraday's constant (equations (3) and (4) from ref. 69). Thus, by measuring and $V$ and assuming the values of and $E_K$, one can estimate the energetic cost of maintaining the resting potential.

To estimate the cost of each action potential, the difference between resting potential and the peak of the action potential (assumed to be a typical value of 60 mV for AF1165 cells) is multiplied by the membrane capacitance (assumed 20 pF, a typical value for AF1165 cells), as estimated from the measured passive time constant of the membrane, to give the amount of sodium charge required to depolarize the membrane during each action potential. This gives an estimate of the ATP required of $2.5 \times 10^6$ ATPs per action potential, which is a conservative lower bound because it assumes complete efficiency of the action potential (non-overlap in time of sodium and potassium voltage-gated currents[70]).

To estimate the cost of vesicular release, we assumed a typical value of 20 F increase in the plasma membrane capacitance, which

was measured using the phase shift of current during voltage clamp to a 1-kHz sinusoidal command voltage, following a brief depolarizing pulse[71,72]. Assuming a specific membrane capacitance of $1\ \mu F\ cm^{-2}$, this translates to the release of about 400 vesicles (40-nm diameter). We have no specific information about the energetics of packaging of the vesicles in these cells, but using the rough estimate[73] of 23,400 ATPs per 40-nm vesicle in neurons yields a cost of $5 \times 10^7$ ATPs per action potential for the vesicular release.

These costs can be compared to reported measurements of ATP production in a panel of epithelial cancers[74], which showed an upper limit of around 20 pmol $\mu g^{-1}$ of protein per minute. Assuming a dry weight fraction of 25% and that each cell has a volume of 1 pl (equivalent to a cube with sides of 10 $\mu$m) yields 4,000 cells per microgram of protein. This equates to an ATP production rate of $5 \times 10^7$ ATPs per cell per second.

### Blue light stimulation

For blue light stimulation, we used Bluecell, a device with in-house built hardware and software, as detailed in a previous study[75]. This device operates with an array of blue LED strips (460 nm), as well as temperature sensors and cooling fans that prevent overheating the cell culture plates upon prolonged blue light exposure. The LED array is controlled by a Raspberry Pi microcomputer, which receives a light sequence that was previously programmed using the BLUECELL.ijm file provided by J.-P. Vincent's laboratory. The file was run in ImageJ2 (v.2.9.0), and different illumination sequences were designed. All the illumination sequences had a blue light intensity of 10 mW $cm^{-2}$ with different frequency patterns (1, 10 or 100 Hz) maintained during a range of time points (1, 5 or 10 min). Specifically, for the 1-Hz stimulation sequence, an additional step of 5 s without blue light exposure was set up in every cycle of stimulation to prevent exhaustion of the cells.

### Drug and metabolite treatments

The following drugs were used: tetrodotoxin citrate (TTX; ab120055; Abcam), rotenone (R8875; Sigma-Aldrich), oligomycin (O4876; Sigma-Aldrich), FCCP (HY-100410; MedChemExpress), antimycin A (A8674; Sigma-Aldrich), SR-13800 (5431; Tocris Bioscience), diclofenac sodium salt (D6899; Sigma-Aldrich) and CCh (C4382; Sigma-Aldrich). The metabolites used for different metabolic assays were sodium pyruvate (P2256-25g; Sigma-Aldrich) and sodium lactate (L7022-5G; Sigma-Aldrich). Concentrations were specified in each of the experiments. For LbNOX induction, cells were treated with 1 $\mu$g $ml^{-1}$ of doxycycline hyclate (D9891; Sigma-Aldrich).

### CellTiter-Glo assay and dose–response curves

Cells were plated in 96-well assay plates (3903; Corning) (3,000–6,000 cells per well for NE SCLC cells in wells pre-coated with Matrigel/HBSS solution; 500 cells per well for non-NE SCLC/LUAD/PDAC cells) in a regular culture medium. The next day, the cells were switched to DMEM with 10% dialysed FBS (F0392; Sigma-Aldrich) containing the drugs of interest. After 3 days of culture or the respective drug treatments, the number of viable cells per well was measured using the CellTiter-Glo Luminescent Cell Viability Assay (G7570; Promega). The results were normalized to control and untreated samples. All assays were performed in four technical replicates per cell line tested. For dose–response curves performed with LbNOX-expressing cells, they were cultured with 1 $\mu$g $ml^{-1}$ doxycycline since the seeding day.

### Conditioned medium collection and dialysis

To collect conditioned medium, non-NE cells were seeded in 15-cm plates at $5 \times 10^6$ cells per plate, whereas NE cells were seeded in Matrigel-coated 10-cm plates at $10^7$ cells per plate. Seeding was done in the respective regular culture medium. After 24 h of seeding, the medium was changed (20 ml per 15-cm plate and 10 ml per 10-cm plate) to pyruvate-free DMEM (10-017-CV; Corning) with 100 U $ml^{-1}$ of penicillin–streptomycin and 2% of commercially dialysed serum (F0392; Sigma-Aldrich). Another 24 h after medium change (48 h after seeding), the medium was collected and filtered through a 0.22-$\mu$m filter. The medium was either stored at 4 °C for use within a week or aliquoted and stored at −80 °C. The conditioned medium was subject to equilibrium dialysis using 3.5K MWCO cassettes (66330; Thermo Fisher Scientific). The conditioned medium was injected into cassettes and allowed to dialyse against a fresh medium at a volumetric ratio of 1:100 overnight at 4 °C. This process was repeated twice per round of dialysis for an expected dilution ratio of 1:10,000 of all molecules smaller than 3.5K MW. For calcium imaging experiments, FluoroBrite DMEM (A1896701; Thermo Fisher Scientific) was used in the protocol instead of DMEM (10-017-CV; Corning) and subsequently collected.

### Radioactive 2-deoxyglucose uptake assay

The day before the experiments, cells were seeded in six-well plates in a regular culture medium (DMEM; 10-013-CV; Corning; with 10% FBS and 100 U $ml^{-1}$ of penicillin–streptomycin), with two plates per cell line. On the day of the experiment, cells were changed to a 1 ml fresh medium per well followed by incubation in a 37 °C incubator. After 2 h, one plate per cell line was taken out, with 100 $\mu$l of radiolabelled 2-deoxyglucose spiked into each well (100 $\mu$l of 11 $\mu$Ci $ml^{-1}$ [$^3$H]-2DG, for a final concentration of 1 $\mu$Ci $ml^{-1}$). The plates were gently rocked for mixture and incubated for exactly 15 min at room temperature. The medium was then aspirated, and the plates were placed on ice and washed four times with ice-cold PBS (5 ml per well). The cells were trypsinized with 500 $\mu$l trypsin/EDTA and mixed well using a 1-ml pipette. Then, 400 $\mu$l of cells per trypsin mixture per sample was transferred into scintillation vials, and scintillation buffer was added. The vials were swirled until the solution turned clear, and the samples were ready for radioactivity measurement. Cell numbers in the other plate were counted in parallel.

### Lactate titration experiment and metabolite quantifications

NE cells were seeded at $10^6$ cells per well in 2 ml regular culture medium in six-well plates the day before the experiment. Mock plates were medium-only controls and treated the same way as other wells. The next day, wells were washed with 5 ml PBS twice before replacing with a fresh medium containing different concentrations of lactate (DMEM; 10-017-CV; Corning; with 10% dialysed serum or 2% dialysed serum for the paired NE/non-NE cell experiments). The next day, the medium was collected after gentle rocking of the plates, centrifuged at 1,000 rpm for 5 min, and then 1 ml of supernatant was collected. The glucose and lactate concentrations were measured in a YSI Bioanalyzer (Yellow Springs Instruments). The degree of secretion/uptake in these metabolites was determined by the difference between concentrations from the control and the cell-containing medium.

### Sulforhodamine B assay for proliferation

The sulforhodamine B (SRB) assay was performed, as described previously[76]. For the blue light stimulation experiment, between $6 \times 10^4$ and $4 \times 10^5$ mSCLC-NE, mSCLC-non-NE, LUAD and PDAC cells were seeded in six-well plates with 1 ml of medium per well. During the following 3 days, the cells were either daily stimulated with 1 or 10 Hz blue light (1, 5 or 10 min), or treated with 1 $\mu$M TTX or with the conditioned medium. The day after seeding, one of the plates was used as the $t_0$ time point of the experiment and was fixed by adding 500 $\mu$l of 10% trichloroacetic acid to each well. After 3 days, all plates were also fixed with 500 $\mu$l of 10% trichloroacetic acid per well and kept in the cold room for at least 1 h. All plates (including the $t_0$ plate) were then washed three times with distilled water (dH$_2$O) and stained with a solution of 0.04% SRB (prepared in 1% acetic acid) for at least 30 min at room temperature. Next, the dye was removed, and the wells were washed three times with 1% acetic acid. After the last wash, the remaining liquid was aspirated completely. The plates were placed with the lids off in an incubator at 37 °C

for 10–15 min until dried. Finally, 1 ml of 10 mM Tris (pH 10.5) was added to each well to solubilize the dye. The plates were shaken for 5–10 min at room temperature. Then, 100 µl of each well was transferred to a 96-well plate, and absorbance was read at 510 nm in the TECAN Infinite M1000 microplate reader. The doubling time per day was calculated with the following formula, where $X_n$ and $X_0$ are the absorbance values at the end and beginning ($t_0$) of the experiment, respectively. For each experiment, three technical replicates were included.

$$\text{Doubling time per day} = \left(\frac{\ln(X_n/X_0)}{\ln(2)}\right)/3$$

## Colony formation assay of mouse cell lines

Clonogenicity was evaluated by seeding 1,000–10,000 cells of mSCLC-NE, mSCLC-non-NE, LUAD and PDAC cells and 30,000 cells of PaNET cells per well in six-well plates. After 10–14 days, colonies were stained for 15 min with a solution of 0.1% crystal violet, 1% formaldehyde, 1% methanol in 1× PBS. After extensive washing and drying, the staining reagent was resolubilized in 1% acetic acid, and absorbance was measured at 570 nm with a TECAN Infinite M1000 microplate reader as an indirect measure of cell number.

For the experiments with blue light stimulation, cells were seeded for colony formation assay, and the next day, they were stimulated with blue light for 10 min (1, 10 or 100 Hz), including an unstimulated condition in each of the experiments. For the colony formation assays with TTX, two experimental set-ups were designed. In one of them, the cells were seeded in six-well plates, as described above. The next day, the medium was replaced by either a fresh medium or medium with 1 µM TTX. The plates were grown under regular conditions for 10 days before the staining of the colonies. In the other approach, before seeding the colony assays, the cells were pretreated with 1 µM TTX for 24 h. The following day, both untreated and treated cells were washed and replated for colony formation in a regular culture medium without TTX and grown for 10 days.

## Colony formation assay of hSCLC cell lines

Soft-agar colony formation assays were used to assess the clonogenicity of hSCLC cell lines, as described in a previous study[77]. In brief, a 1.2% low-melting-point agarose (16520-100; Invitrogen) solution was mixed with an equal volume of 2× medium, which contained 26.74 g l$^{-1}$ DMEM powder (50-013-PB; Corning), 2.4 g l$^{-1}$ sodium bicarbonate, 1 mM sodium pyruvate, 20% FBS (11320033; Gibco) and 2× NEAA (11140050; Gibco). Then, 1.5 ml of the agarose-medium mixture was layered onto the bottom of each well in six-well plates and left at room temperature to solidify for 30 min. For the top layer, each of the hSCLC cell lines was resuspended in 2× medium to reach a final density of 15,000 cells per well. Next, these cell suspensions were mixed with 0.6% agarose solution at a 1:1 ratio and added to the top of each well (1.5 ml). After a 30-min incubation at room temperature, 1 ml per well of complete culture medium (with or without 1 µM TTX) was added, and the plates were incubated under regular growth conditions for 3–4 weeks. Every 2 days, a fresh medium (with or without 1 µM TTX) was added to the wells to prevent evaporation. At the end of the experiment, the colonies were stained by adding 200 µl of nitroblue tetrazolium chloride solution (1 mg ml$^{-1}$; N6876; Sigma-Aldrich) per well and incubating the plates overnight at 37 °C.

## Western blot

Total protein was extracted from cells using cell lysis RIPA buffer, supplemented with a phosphatase and protease inhibitor cocktail (78440; Thermo Fisher Scientific); 30 µg of protein lysate was resolved by SDS–polyacrylamide gel electrophoresis (4–15%) and transferred to polyvinylidene fluoride (PVDF) membranes, which were blocked with either 5% BSA in TBS-T or 5% non-fat milk in PBS-T and incubated overnight with the indicated antibody. The primary antibodies used were the following: anti-HES1 (11988S; 1:1,000; Cell Signaling Technology), anti-MCT4 (sc-376140; 1:100; Santa Cruz Biotechnology), anti-MCT1 (AB1286-I; 1:1,000; Sigma-Aldrich), anti-SOX1 (4194S; 1:1,000; Cell Signaling Technology), anti-c-FOS (ab190289; 1:500; Abcam), anti-GPX4 (ab125066; 1:1,000; Abcam), anti-4-HNE (ab46545; 1:1,000; Abcam), anti-LCB3 (43566; 1:1,000; Cell Signaling Technology), anti-alpha-tubulin-HRP (ab40742; 1:3,000; Abcam), anti-Hsp90 (610418; 1:3,000; BD Biosciences), anti-phospho-Ser133-CREB (9198; 1:500; Cell Signaling Technology) and anti-FLAG tag (2368; 1:1,000; Cell Signaling Technology). If needed, the membranes were then incubated with HRP-conjugated anti-rabbit (ab205718; 1:3,000; Abcam), anti-mouse (G-21040; 1:3,000; Invitrogen) or anti-chicken (A16054; 1:3,000; Invitrogen) secondary antibodies. The target protein bands were visualized using ECL Prime Western Blotting Detection Reagents (RPN2232; Amersham) and ChemiDoc XRS+ System (Bio-Rad). For gel source data, see Supplementary Fig. 1.

## Immunofluorescence

A single fixed lung lobe was cryoprotected with sucrose (30% sucrose in PBS, overnight), embedded in optimal cutting temperature compound and slowly frozen in dry ice. Human SCLC tissue array LC703a was bought from US Biomax. Samples were cryo-sliced into 100-µm-thick sections using a Leica CM3050 S cryostat. Slices were collected on SUPERFROST PLUS adhesion microscope slides (J1800AMNZ; Epredia), washed in PBS for 30 min at room temperature and quenched in 0.24% NH$_4$Cl PBS for 10 min. Slides were permeabilized and saturated in a blocking solution (15% donkey serum, 0.20% glycine, 2% BSA, 0.25% gelatine and 0.5% Triton X-100 in PBS) for 2 h at room temperature and then incubated for 24 h in the blocking solution at 4 °C with the following primary antibodies: anti-MCT4 (sc-376140; 1:100; Santa Cruz Biotechnology), anti-SOX1 (AF3369; 1:200; Bio-Techne), anti-VAChT (139105; 1:100; Synaptic Systems), anti-tdTomato (AB8181-200; 1:200; SICGEN) and anti-β3-tubulin (ab52623; 1:250; Abcam). The day after, samples were washed three times in PBS for 10 min each and incubated in PBS for 1 h at room temperature with the following fluorescent conjugated secondary antibodies: donkey anti-mouse Alexa Fluor (AF) 488 (A21202; 1:250; Invitrogen), donkey anti-rabbit AF 488 (A-21206; 1:250; Invitrogen), donkey anti-goat AF 568 (A-11057; 1:250; Invitrogen), donkey anti-goat AF 647 (A32849; 1:250; Invitrogen) and goat anti-guinea pig AF 647 (A21450; 1:250; Invitrogen). The samples were then washed three times in PBS for 10 min and incubated with DAPI (D1306; 1 µg ml$^{-1}$; Thermo Fisher Scientific) in PBS for 5 min. The slides were mounted using the Dako fluorescence mounting medium (S3023; Agilent Technologies) and Epredia 22 × 50-mm coverslips. Images were captured with a Leica SP8 FALCON inverted confocal microscope with white light laser (470–670 nm), equipped with HC PL APO 20× NA 0.75 CB2, HC PL APO 40× NA 1.30 CS2 and HC PL APO 63× NA 1.40 CS2 oil immersion objectives and HyD detectors. The laser excitation line, power intensity and emission range were chosen according to each fluorophore to minimize bleed-through. Data were collected with LAS X software. Data were analysed using ImageJ software, and JACoP plugin was used to perform co-localization analysis and calculate Pearson's coefficient. To calculate the cumulative axon length in proximity to the tumour, an ROI was designed 300 µm radially to the tumour, and NeuronJ plugin was used. The axon density score was calculated as the ratio between the cumulative axon length and diameter of the tumour. The number of MCT4- and SOX1-positive cells at the TMA was calculated using QuPath 0.5.0, and 3D visualization, rendering and videos have been generated using the Imaris software.

## Immunohistochemistry

For immunohistochemistry analysis, tissue samples were fixed in 10% NBF, dehydrated, embedded in paraffin wax and sectioned at 4 µm using a Leica RM2235 microtome. Slides were dewaxed in xylene twice

for 5 min and rehydrated with 100% industrial methylated spirit (IMS) twice for 5 min, followed by 70% industrial methylated spirit for 5 min and dH$_2$O for 5 min. The samples were transferred to a citrate-based antigen retrieval solution (H-3300; Vector Laboratories), microwaved at 900 W for 8 min, cooled to 50 °C, boiled again for 3 min and finally cooled to room temperature. The slides were incubated in 1.6% H$_2$O$_2$ in PBS for 10 min, washed for 5 min in dH$_2$O, incubated in 1% BSA for 1 h at room temperature and incubated ON at 4 °C with the following primary antibodies: anti-phospho-Ser133-CREB (9198; 1:400; Cell Signaling Technology), anti-Ki67 (ab15580; 1:500; Abcam) and anti-cleaved Caspase 3 (9579S; 1:250; Cell Signaling Technology). The samples were washed three times in PBS-T for 5 min. For the secondary antibody (anti-rabbit polymer) and 3,3′-diaminobenzidine, a BOND Polymer Refine Detection kit (DS9800; Leica) was used. Briefly, the samples were incubated with a Polymer detection system reagent for 30 min at room temperature and washed in PBS-T three times for 5 min each. 3,3′-Diaminobenzidine chromogen solution was applied, washed in dH$_2$O to terminate the reaction and counterstained in Sakura Tissue-Tek Prisma. Images were captured with an Olympus VS200 slide scanner. The function 'positive cell detection' in QuPath 0.5.0 was used to quantify Ki-67 and cleaved Caspase 3-positive cells in liver sections.

## Gas chromatography–mass spectrometry

NE cells were seeded at 10$^6$ cells per well, and non-NE cells were seeded at 250,000 cells per well in six-well plates in 2 ml regular DMEM per well. Six wells per cell line were seeded for two time points and three technical triplicates per time point. The controls were DMEM in mock plates without cells. The next day, the plates were washed with 5 ml PBS twice for each well, including the mock plates, and then replaced with exactly 2 ml of the fresh medium (10-017-CV with 10% dialysed FBS; Corning) using P1000 pipettes. In the meantime, three wells per cell line were counted for cell number at T0. After 24 h, 10 µl of medium was collected for gas chromatography–mass spectrometry, and cell counts were obtained for T24. Growth rates of NE cells and non-NE cells were established by fitting an exponential growth equation to the initial and final cell counts, and integration of this equation was performed to determine the average cell per unit time over the course of this experiment (area under the curve). The collected medium was stored at −80 °C or proceeded directly to the extraction of polar metabolites. High-performance liquid chromatography (HPLC)-grade methanol containing a norvaline (Sigma) internal standard at a concentration of 10 µg ml$^{-1}$ was added to samples in 1.5-ml Eppendorf tubes to reach a final concentration of 80% methanol, and the mixture was then vortexed for 10 min at 4 °C then centrifuged at the maximum rpm for 10 min at 4 °C. The supernatant was collected and transferred to new Eppendorf tubes and dried under inert nitrogen gas. Dried samples were derivatized to form methoxime-tBDMS derivatives by initial incubation with 16 µl MOX reagent (2% methoxyamine hydrochloride in pyridine, Thermo Scientific) at 37 °C for 90 min, followed by addition of 20 µl N-tert-butyldimethylsilyl-N-methyltrifluoroacetamide (MTBSTFA) with 1% tert-butyldimethylchlorosilane (t-BDMCS) (Regis Technologies) and incubation at 60 °C for 30 min. The samples were then centrifuged at maximum speed for 10 min at 4 °C. Then, 16 µl of supernatant was transferred to a gas chromatography–mass spectrometry vial. Gas chromatography–mass spectrometry analysis was performed using an Agilent 6890 GC equipped with a 30 m DB-35MS capillary column connected to an Agilent 5975B MS operating under electron impact ionization at 70 eV. One microlitre of sample was injected in splitless mode at 270 °C, using helium as the carrier gas at a flow rate of 1 ml min$^{-1}$. The gas chromatography oven temperature was held at 100 °C for 3 min and increased to 300 °C at 3.5 °C min$^{-1}$. The mass spectrometer source and quadrupole were held at 230 °C, and the detector was run in scanning mode, recording ion abundance in the range of 100–605 $m/z$. Annotation of lactate and pyruvate peaks were determined by reference to comparison to lactate and pyruvate standards derivatized using the same procedure described above. The metabolite secretion rates were calculated by dividing the total number of extraction ions per metabolite by the average cell per unit time.

## Seahorse measurement of OCR

An Agilent Seahorse Bioscience Extracellular Flux Analyzer was used to measure OCRs. Cells were seeded in Agilent Seahorse XF96 cell culture microplate (TC-treated; 102416-100; Agilent Technologies) for experiments, and the wells were coated with Cultrex BME (3432-010-01; Bio-Techne) in HBSS. SCLC NE cells were seeded at 10$^5$ cells per well, whereas non-NE cells were seeded at 10$^4$ cells per well in 100 µl appropriate regular culture medium. The next day, the medium was replaced by 180 µl per well Seahorse XF DMEM Medium (103575-100; Agilent Technologies) supplemented with 10 mM D-glucose, 1 mM pyruvate and 2 mM L-glutamine. The plates were incubated at 37 °C for 1 h without CO$_2$. The XF Cell Mito Stress Test protocol was carried out after injecting oligomycin (1.5 µM, final concentration), FCCP (1.5 µM, final concentration) and rotenone–antimycin A (0.5 µM, final concentration). After running the assay, the protein content of each well was measured with the Pierce BCA Protein Assay Kit (23225; Thermo Fisher Scientific). The total milligrams of protein were used for normalization. Data analysis was performed using the online software Seahorse Analytics (v.1.0.0-699) to calculate the basal respiration rates, ATP-coupled respiration and coupling efficiency.

## In vitro calcium imaging

Cal-520, AM (ab171868; Abcam)-loaded SCLC NE cells, GCaMP6m-expressing mSCLC-NE cells alone or GCaMP6m-expressing mSCLC-NE and non-NE cells in a ratio of 4:1 were seeded in 35-mm glass-bottom dishes (81218-200; ibidi) coated with Matrigel (356231; Merck) and cultured overnight in FluoroBrite DMEM (A1896701; Thermo Fisher Scientific) supplemented with GlutaMAX (35050038; Thermo Fisher Scientific) and penicillin–streptomycin (15140122; Thermo Fisher Scientific) with 2% dialysed FBS (26400044; Thermo Fisher Scientific). They were imaged on the next day using an Olympus IX73 inverted epifluorescence microscope equipped with a pe-300 Ultra standard fluorescence illumination system with TTL trigger control and a Prime BSI Express sCMOS Camera 4.2 MP with a 95% quantum efficiency. A 40× Plan-Neofluar objective (0.75NA) was used to acquire images of cells that were stimulated with an LED light source using the green channel (excitation at 470 ± 20 nm) and a GFP filter set, and controlled by the Micro-Manager software (v.2.0.0). Recordings were made at 0.5 frames per second (Hz) for 5 min. The exposure time was 200 ms. For experiments, including the MCT inhibitor treatments, SR-13800 (5 µM) and diclofenac (0.5 mM) were added to the cell culture medium and incubated for 5 min before imaging.

Calcium imaging of 3D-cultured NE SCLC cells was performed using a custom-built upright light sheet fluorescence microscope at the Cambridge Advanced Imaging Centre.

## Ex vivo calcium imaging

Precision-cut lung slices were obtained from mice using a protocol adapted from a previous study[78]. A solution of 2% low-melting-point agarose (16520-100; Invitrogen) in HBSS was used for lung inflation. For imaging of tumour cells expressing Salsa6f reporter, the lung lobes harbouring the tumours were isolated and cut transversely at 300 µm using an automated vibratome (Leica VT1200S) in ice-cold HBSS/HEPES buffer.

The slices were placed in a 12-well plate in serum-free DMEM (21063029; Gibco) supplemented with 1% penicillin–streptomycin (15140122; Thermo Fisher Scientific) and incubated at 37 °C with 5% CO$_2$ for 30 min before mounting for imaging. The slices that were used for imaging of NEBs were incubated with Oregon Green 488 BAPTA-1-AM (O6807; Invitrogen) for 30 min before imaging.

The slices were mounted between two thin layers of a low-melting-point agarose gel, solidified at room temperature in a 24-well glass-bottom imaging plate with 1.5 cover glass (P24-1.5H-N; Cellvis). They were imaged using an Olympus CSU-W1 SoRa spinning disk confocal microscope with an environmental chamber (at 37 °C, 5% $CO_2$). A 30× silicon immersion objective (1.05 NA) and 488 and 561 nm laser excitation wavelengths were used for image acquisition. Recordings were made at 0.5 frames per second for 5 min for each field of view.

For experiments including treatment with the MCT4 inhibitor diclofenac (0.5 mM), time lapses were recorded from the same field of view before treatment with the drug and after 5-min incubation with the drug.

## Calcium imaging analysis
For all calcium imaging experiments, individual fields of view were analysed in ImageJ (v.1.54f). The TrackMate plugin and Cellpose detector pretrained models cyto and cyto2 were used for automated segmentation of cells and tracking during the time lapse recorded for each field of view. Only cells that have been segmented and tracked at all time points of the time lapse were included in the subsequent calcium peak analysis. The mean intensity fluorescence values were obtained for each segmented cell, and the background measured in each field of view was subtracted from individual values recorded at each time point. The individual fluorescence intensity traces for each segmented cell were generated using a custom MATLAB script[79], and peak metrics were generated using the findpeaks function. The fluorescence intensity at each time point was normalized to the median of all fluorescence intensity values for a particular segmented cell. A cutoff of 0.1 was used for peak prominence. Only cells with at least one peak with a minimum peak prominence of 0.1 were considered active cells.

## RNA sequencing
Total RNA from mSCLC-NE/non-NE cell lines was isolated using the miRNeasy Kit (QIAGEN). RNA libraries were prepared for sequencing with the Illumina TruSeq kit following the manufacturer's instructions. Illumina HiSeq 2000 50-nt single-ended reads were mapped to the UCSC mm9 mouse genome build (http://genome.ucsc.edu/) using RSEM[80] (v.1.2.12) and bowtie (v.1.0.1) with default options. Raw estimated expression counts were upper-quartile normalized to a count of 1,000 (ref. 81). Given the complexity of the dataset in terms of a mixture of different biological backgrounds, a high-resolution signature discovery approach was used to characterize global gene expression profiles. Independent component analysis, an unsupervised blind source separation technique, was used on this discrete count-based expression dataset to elucidate statistically independent and biologically relevant signatures, as detailed previously[1]. All RNA sequencing analyses were conducted in the R Statistical Programming language (http://www.r-project.org/). GSEA was carried out using the pre-ranked mode with default settings[82]. Heat maps were generated using the Heatplus package in R.

## Analysis of published datasets
To further investigate the presence of vulnerabilities of SCLC related to the characteristic metabolic requirements observed in our experiments, we used previously published CRISPR screens performed with SCLC, LUAD and PDAC cells[42,83]. The criteria chosen to identify specific vulnerabilities of SCLC were median $\log_2$ fold change (L2FC) < −2 in SCLC and median L2FC in SCLC minus the median L2FC of LUAD/PDAC < −1.5. Eight genes of the 4,915 genes within the library fitted these criteria. These eight genes were uploaded into Enrichr to perform a GSEA, selecting the 'GO Biological Process 2023' dataset as a refs. 84–86.

To extend our observations to other datasets generated with PDXs of SCLC, we analysed an RNA sequencing dataset from a cohort of 51 PDXs[27]. Specifically, we performed a Pearson correlation analysis using their normalized RNA sequencing data in fragments per kilobase of transcript per million mapped reads and the NE scores they assigned to these PDXs. Moreover, we divided these PDXs into two groups according to their NE score: NE with NE score greater than 0.8 and non-NE with NE score less than 0.2. *SOX1* and *SLC16A3* (encoding MCT4) mRNA levels were analysed in these two groups. The normality of the data was assessed using the Shapiro–Wilk test, and data transformations and statistical tests were chosen accordingly, as described in figure legends.

Additionally, we analysed the RNA sequencing and proteomic datasets of a cohort of 112 patients with SCLC (with paired tumour and NAT samples) recently published[56]. Further details about data acquisition, normalization and clinical information can be found in the original paper. For differential expression analyses between tumours and paired NATs, the Wilcoxon matched-pair signed-rank test was applied. For correlation analyses, the Spearman test was used. For survival analyses, we split the patients into two groups ('high' or 'low') based on whether their normalized and $\log_2$-transformed expression was above or below the median for each of the genes of interest. We plotted Kaplan–Meier curves for 'high' and 'low' patients and compared both groups using a univariate log-rank test.

## Statistics and reproducibility
Unless stated otherwise, all statistical analyses were performed in GraphPad Prism using the recommended tests and post hoc tests from the software. No data have been excluded from the analyses. The number of replicates for each experiment is reported in the corresponding figure legend and methods. Western blotting was performed with at least two biological replicates. No statistical methods were used to calculate the sample size. Sample sizes were chosen based on preliminary experiments aimed at capturing biological effects in line with similar research in the field.

## Reporting summary
Further information on research design is available in the Nature Portfolio Reporting Summary linked to this article.

## Data availability
The raw and processed data of SCLC and PDAC cell lines from the RNA sequencing experiments are available in the NCBI Gene Expression Omnibus database (accession number: GSE270281). Previous RNA sequencing data of LUAD cell lines were deposited under the accession number GSE145945 (ref. 83). Expression data from the hSCLC cell lines were accessed from the Cancer Dependency Map (DepMap) portal (www.depmap.org), specifically from DepMap Public 23Q2 and the proteomic dataset. The values from the DepMap Public 23Q2 dataset were inferred from RNA sequencing data using the RSEM tool and are reported after $\log_2$ transformation, using a pseudo-count of 1; $\log_2$ (TPM + 1). The expression data and NE scores from the PDX models were provided by L. Girard. Source data are provided with this paper.

## Code availability
Bioinformatics analyses were performed using open-source software. MATLAB codes for calcium imaging analyses and biophysical modelling of ATP consumption are available at Figshare (https://doi.org/10.6084/m9.figshare.27630099)[87].

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

**Acknowledgements** We thank J. Sage, K. Vousden, A. Gould and C. Tsantoulas for insightful comments on the paper; K. Yee, J. Teixeira, C. Whittaker and J. MacRae for technical support; and J.-P. Vincent laboratory for Bluecell. We are grateful to The Francis Crick Institute's STPs: BRF, CALM, EHP, Metabolomics and Viral Vector Core. This study was supported by the HHMI, Ludwig Center for Molecular Oncology at MIT, Koch Institute Support (core) grant P30-CA14051 from the National Cancer Institute, CRUK Multidisciplinary Project DRCMDP-Jun22\100016, The Francis Crick Institute Core Grant CC2165 and UKRI Grant ERC-UKRI-2021-101041807-CancerNeuroscience. P.P. was supported by the MSCA-RESPIRE4 Fellowship (R4202305-01050), and C.B. was supported by an EMBO Postdoctoral Fellowship (ALTF 227-2022). B.J.D. was supported by CPRIT RR20007 and NIH 1K08CA237832. E.S. was supported by The Francis Crick Institute, which receives its core funding from Cancer Research UK (CC2040), the UK Medical Research Council (CC2040), Wellcome Trust (CC2040) and the European Research Council (ERC Advanced Grant CAN_ORGANISE, grant agreement no. 101019366). M.G.V.H. was supported by the MIT Center for Precision Cancer Medicine, Ludwig Center at MIT, Lustgarten Foundation and NCI (R35CA242379 and P30CA014051). L.L. was supported by the Swiss National Science Foundation Early and Advanced Postdoc.Mobility Fellowships and the Hope Funds Postdoctoral Fellowship. Schematics in Figs. 1a, 2c,m, 3a, 4a and 5f,j,n and Extended Data Figs. 1a, 2q and 9b,c were created using BioRender (https://biorender.com).

**Author contributions** L.L., Z.L., M.G.V.H. and T.J. conceived of and supervised the study. P.P., M.S., C.B., Z.L., M.V.K., H.P.C.R., M.G.V.H. and L.L. designed experiments. M.-S.H. performed the pathological analysis under the supervision of J.-S.C. S.P.C., V.S. and S.H. derived PDX models under the supervision of B.J.D. P.P. and A.B. performed bioinformatics analyses. M.-B.M. performed and analysed calcium imaging with help from A.L.M., supervised by E.S. Electrophysiology experiments were performed by J.C., L.C., M.V.K. and H.P.C.R., and analysed and supervised by H.P.C.R. H.P.C.R. generated the mathematical model and code for ATP estimation. All other experiments were performed by P.P., M.S., C.B., Z.L., C.I.C., Y.C., M.M., S.R.N., M.V.K. and L.L. The results were analysed by P.P., M.S., C.B., Z.L., M.V.K. and L.L. P.P., M.S., C.B., M.G.V.H., H.P.C.R. and L.L. wrote the paper with comments from all authors.

**Funding** Open Access funding provided by The Francis Crick Institute.

**Competing interests** Z.L. is currently an employee of Sesame Therapeutics and shareholder of Sesame Therapeutics and Tango Therapeutics. L.C. is currently an employee of AstraZeneca. B.J.D. consults for AstraZeneca, Sonata Therapeutics and Dialectic Therapeutics. J.-S.C. receives research funding from VolitionRX and AstraZeneca (Taiwan). M.G.V.H. discloses that he is a scientific advisor for Sage Therapeutics, Agios Pharmaceuticals, Auron Therapeutics, iTeos Therapeutics, Lime Therapeutics, Pretzel Therapeutics, MPM Capital and Droia Ventures. T.J. is a member of the Board of Directors of Amgen and Thermo Fisher Scientific. He is also a co-founder of Dragonfly Therapeutics and T2 Biosystems. T.J. serves on the Scientific Advisory Board of Dragonfly Therapeutics, SQZ Biotech and Skyhawk Therapeutics. None of these affiliations represent a conflict of interest with respect to the design or execution of this study or the interpretation of the data presented in this paper.

**Additional information**
**Correspondence and requests for materials** should be addressed to Leanne Li.

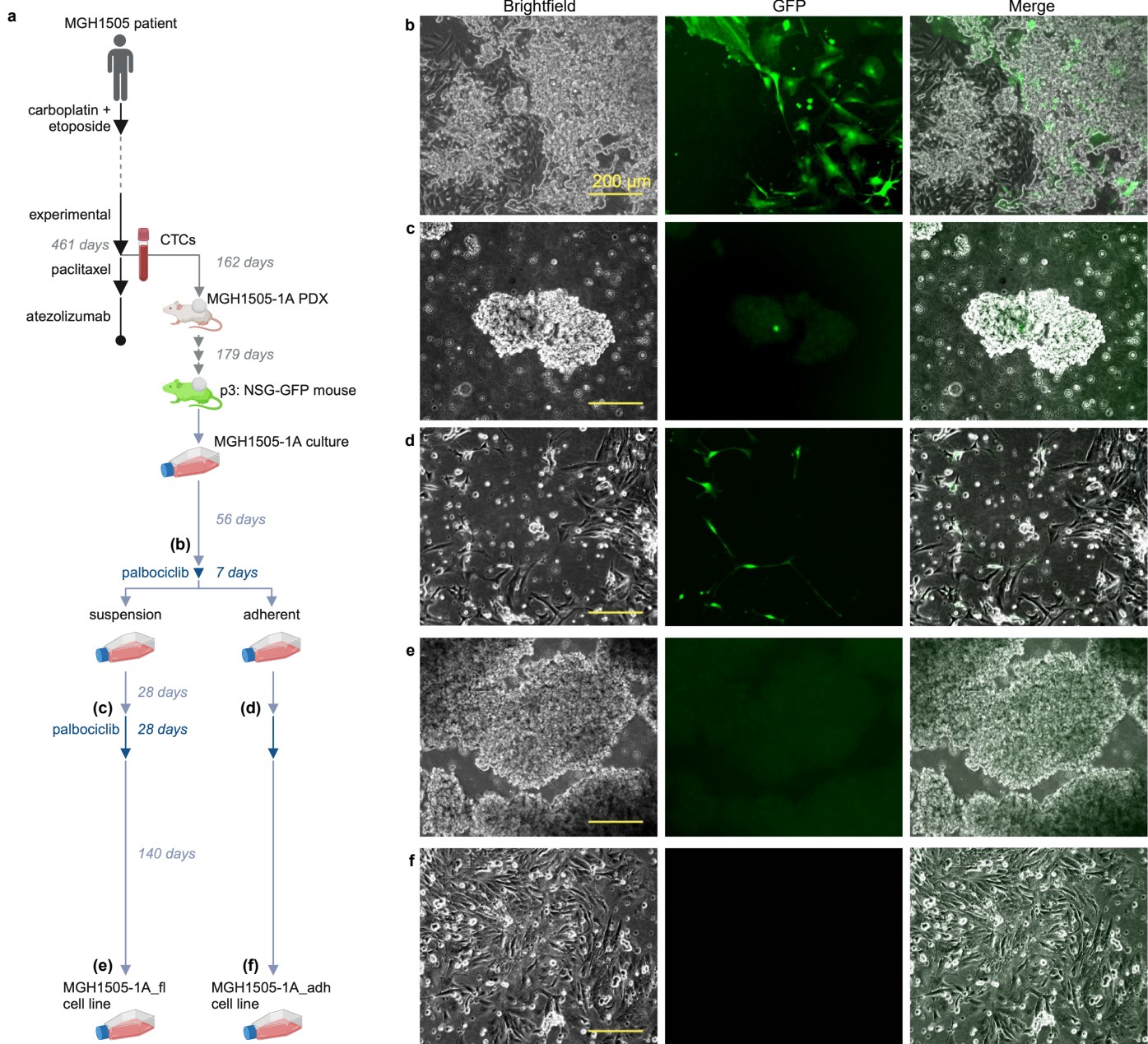

**Extended Data Fig. 1 | Generation of matched PDX-derived SCLC cell lines.**
(a) Schematic illustration of the workflow used to generate PDX-derived
SCLC cell lines with two distinct morphologies from the same tumor.
As previously described[27], patient MGH1505 was treated with standard first-
line chemotherapy with an excellent response, and at first relapse received an
experimental therapy to which he also responded. 461 days into his treatment
course, after his second relapse and just before the beginning of 3rd-line
paclitaxel, live circulating tumor cells (CTCs) were isolated from a routine
blood draw and injected into the right flank of an NSG mouse. After a prolonged
latency, a CTC-derived xenograft, MGH1505-1A, emerged and was resected
162 days after injection. Fragments of MGH1505-1A were passaged serially to
two NSG mice, with significantly shorter latencies (50–70 days). The third
passage was into an NSG-GFP mouse, which enabled identification of murine
stromal cells by GFP+ signal. After reaching a size of 1500 mm³, this xenograft
was manually dissociated for cell line derivation. Numbers next to the

arrows represent the latency times between two events (in gray) or the time
of palbociclib treatment (in blue). (b - f) Live images of cell cultures at
different time points during the cell line derivation process, as annotated in
the schematic in (a). GFP+ cells are murine, while GFP⁻ cells are human SCLC
cells. (b) Image of the culture 56 days after tumor dissociation showing a
mixed culture consisting of tightly and loosely adherent cells. Some tightly
adherent cells are GFP⁻, indicating that they are tumor cells. After this image
was taken, the culture was treated with palbociclib for 7 days to deplete the
GFP+ murine cells. (c, d) Separated suspension (c) and adherent (d) cultures
imaged 28 days after initial palbociclib treatment. Only a few GFP+ cells remain,
and these were fully depleted by palbociclib treatment for an additional 28
days. (e, f) Cell lines with two distinct morphologies, floating clusters (e) and
adherent (f), were generated from a single tumor and were free of any GFP+
murine cells. Scale bars: 200 μm. Schematic in a was created using BioRender
(https://biorender.com).

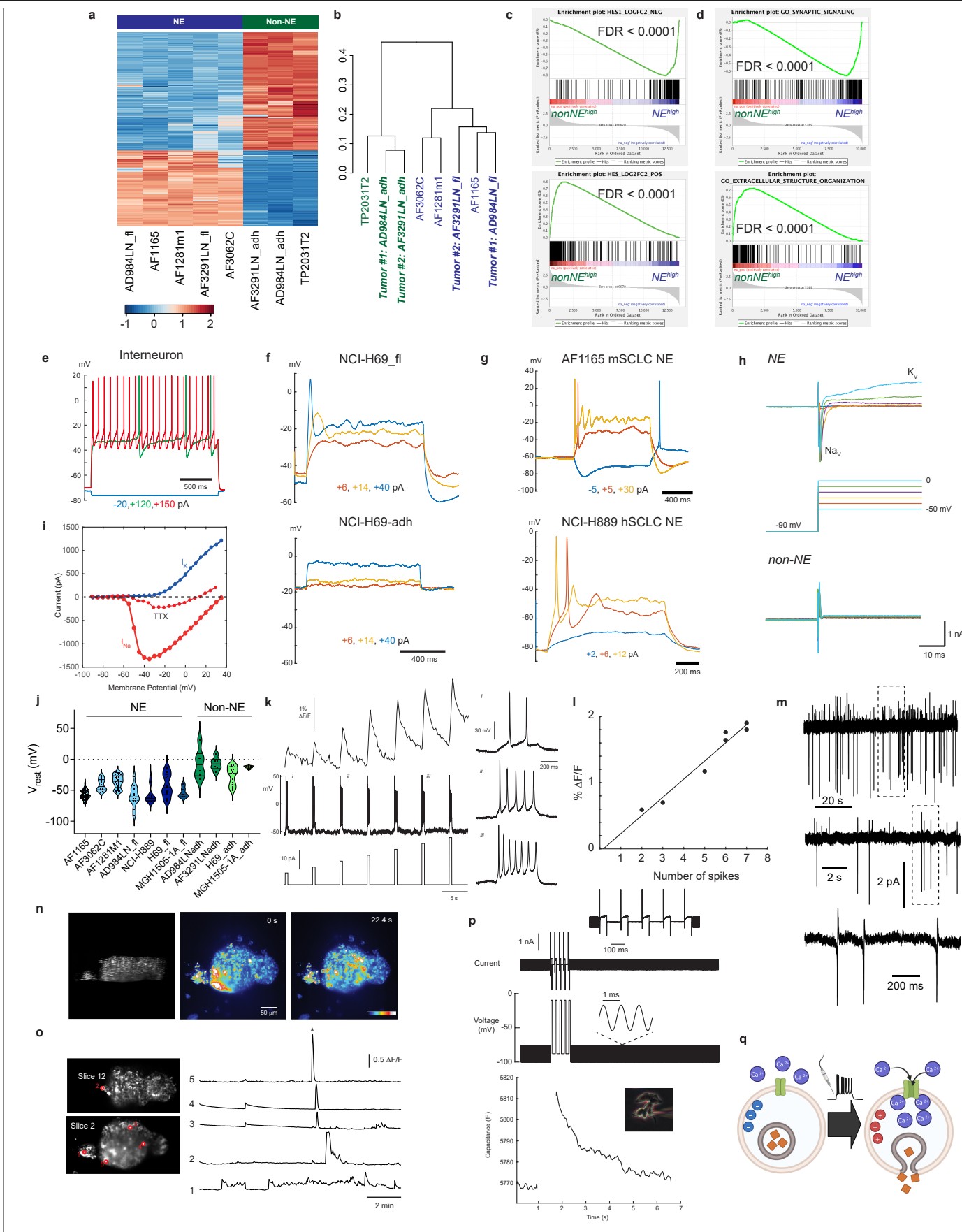

**Extended Data Fig. 2** | See next page for caption.

**Extended Data Fig. 2 | Transcriptional and electrophysiological characterization of NE and non-NE cells.** (**a**) Heatmap of differentially expressed genes in mSCLC cell lines (n = 5 NE cell lines and 3 non-NE cell lines). (**b**) Clustering analysis of NE and non-NE cell lines showing separate clustering, even when two pairs of NE and non-NE cell lines originally derived from the same tumors were compared. (**c, d**) GSEA of rank-ordered differential gene scores of non-NE versus NE mSCLC cell lines showing an enrichment of NE with Hes1$^{negative}$ mSCLC tumor cells (**c**, top) and GO_Synaptic_Signaling (**d**, top); and non-NE enrichment in Hes1$^{high}$ (**c**, bottom) and GO_Extracellular_Structure_Organization pathways (**d**, bottom). Benjamini-Hochberg method applied for correction of multiple testing. False Discovery Rate (FDR) shown. (**e**) Families of representative current-clamp recordings from a mouse cortical interneuron. (**f**) Families of representative current-clamp recordings from paired hSCLC NE (top) and non-NE (bottom) cell lines originally derived from the same parental cell line NCI-H69. Three different current step amplitudes as indicated. Illustrated voltage responses are representative of 20 (NCI-H69_fl) and 15 (NCI-H69_adh) cells. (**g**) Families of current-clamp recordings from mSCLC NE (top) and hSCLC NE (bottom) cell lines. Membrane potential responses to indicated current steps are shown. For each cell line, the illustrated voltage responses are representative of 20 cells patched. Note a rebound spike on the termination of the hyperpolarizing step for AF1165, commonly observed in some NE SCLC cells. (**h**) Voltage-clamp currents in response to a series of voltage steps (middle panel) show the presence of inward fast Na$^+$ and maintained outward K$^+$ currents in NE, but not in non-NE cells. Results were consistent in all mSCLC NE (n = 5) and non-NE (n = 2) cell lines tested. (**i**) Peak early and late currents during depolarizing voltage-clamp steps as a function of membrane potential in NE cells (AF1165). TTX: inward current during application of 1 µM tetrodotoxin. The results are representative of 3 independent NE lines: n = 3 (AF1165), 4 (AD984LN_fl), 5 (AF1281m1) cells. (**j**) Resting membrane potentials (V$_{rest}$) of SCLC cells immediately following breakthrough into the whole-cell mode in current-clamp. Individual lines are shown in the left panel: n = 71, 12, 26, 13, 6, 6, 8, 6 cells/line, from left to right, respectively. (**k**) Left: Calcium signal monitored by Cal-520 indicator (expressed as fluorescence increment relative to baseline, ΔF/F) in the cell body (top) while simultaneously recording membrane potential (middle) and injecting a series of increasing current steps (bottom). Right: Expanded-time view of the action potential responses indicated in the upper panel. Representative of recordings in 4 AD984LN_fl cells. (**l**) Linear relationship between the number of spikes in each response and the amplitude of the calcium increment relative to **k**. (**m**) Non-invasive cell-attached recording of spontaneous action potentials in an AD984LN_fl mSCLC NE cell. Middle and bottom panels: expanded time scale views as indicated by dashed rectangles above. In these extracellular recordings, capacitive currents reflecting action potential upstroke are plotted downwards and repolarization upwards. Spontaneous firing has been observed consistently in more than 10 cells per cell line and across at least three different NE lines examined. (**n, o**) Lightsheet microscope imaging of 3D-cultured GCaMP6m$^+$ AD984LN_fl cells. 12 planes, (side view shown at left), were imaged every 2.8 s (**n**). Scale bar: 50 µm. Representative of 12 tumoroids in 6 different 3D cultures. (**o**) Maximum intensity projections across all planes show the propagation of a spontaneous wave starting from the invasive outgrowth at the left, which shows ongoing rapid activity (location 1) (middle and right panels). (**p**) Membrane capacitance increases following a train of depolarizing voltage-clamp steps in NE cells as monitored by current phase shift in response to a 1 kHz sinusoidal voltage command signal, as illustrated in (**q**). Representative of recordings in n = 8 AF1165 cells. Schematic in **q** was created using BioRender (https://biorender.com).

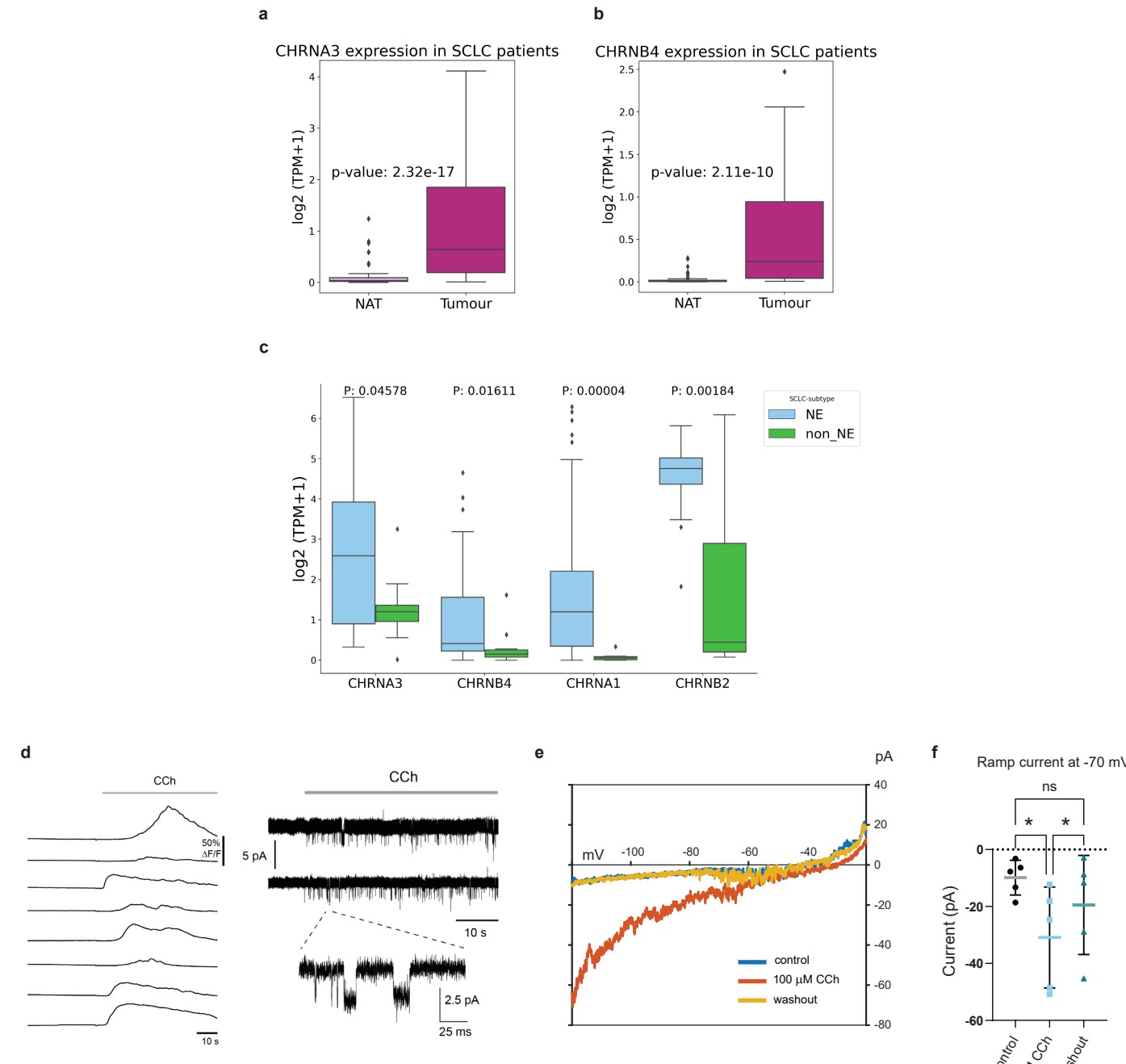

**Extended Data Fig. 3 | Cholinergic signaling in SCLC. (a, b)** Boxplots of normalised mRNA expression values of *CHRNA3* (**a**) and *CHRNAB4* (**b**) in tumour and normal adjacent tissue (NAT) samples from a previously published cohort of SCLC patients (n = 112)[56]. Wilcoxon matched-pairs signed rank two-sided test was applied. Boxplots show the median (line) and IQR (interquartile range), with whiskers no more than 1.5× IQR. (**c**) Boxplot of differentially expressed nAChR subunits between a group of n = 43 hSCLC NE (SCLC-A, SCLC-N) and n = 11 non-NE (SCLC-Y, SCLC-P) cell lines. Significance was evaluated with the two-tailed Mann-Whitney test. Boxplots show the median (line) and IQR (interquartile range), with whiskers no more than 1.5× IQR. (**d**) Left: Fluorescence responses in multiple cells in a field of view to carbachol superfusion (CCh, 40 µM).

Results representative of 5 cultures of AD984LN_fl-GCaMP6m cell line. Right: Membrane channel currents in low-noise whole-cell patch-clamp recordings activated by CCh (40 µM), showing channels with amplitudes and lifetimes characteristic of nAChR. Representative of 5 recordings from AF3062C NE cells. (**e**) Currents during a slow depolarizing ramp of membrane before, during and after application of CCh (100 µM). Representative of recordings from 5 cells of the AF1165 NE cell line. (**f**) Comparison of current level measured at −70 mV during ramps as in (**e**), for n = 5 cells shows significant activation of inward current by CCh. One-way ANOVA was performed to determine statistical significance, (left*) p = 0.0391, (right*) p = 0.0308, ns=non-significant. Mean ± SD shown.

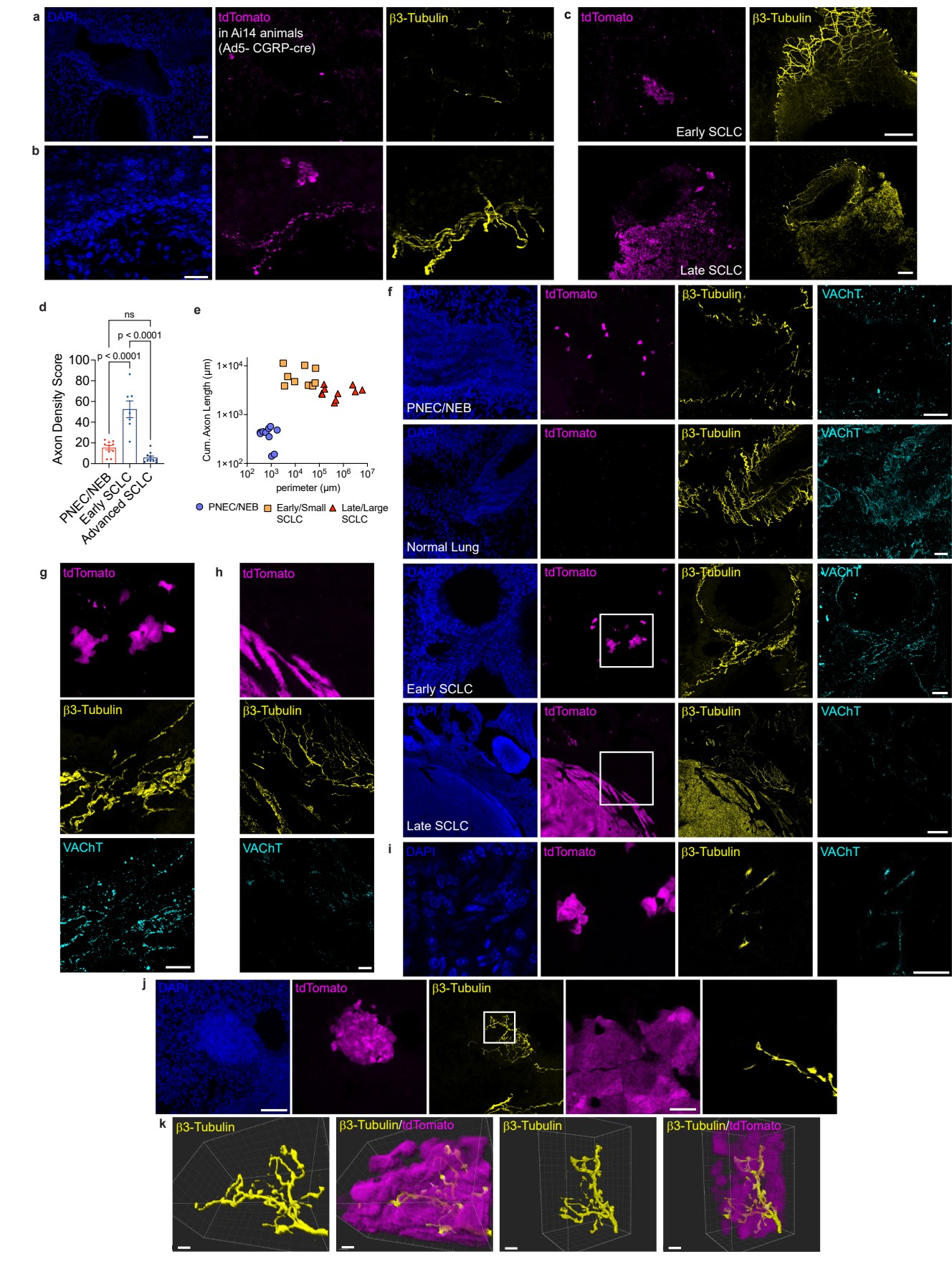

**Extended Data Fig. 4** | See next page for caption.

**Extended Data Fig. 4 | Single channel characterization of SCLC innervation.**
(**a**–**e**) Single channel images of immunostaining of PNEC (a, from Fig. 2d, upper panel; b, from Fig. 2d, lower panel) and SCLC (c, from Fig. 2e) innervation. (**a, b**) Normal pulmonary neuroendocrine cells marked by CGRP (*magenta*) are innervated by β3 tubulin positive neurons (*yellow*). (**b**) A fraction of these nerve axons was CGRP[+], suggesting their sensory origin and relative axon density quantification (**d**). Nuclei are stained by DAPI (*blue*). Scale bars: (**a**) 50 μm; (**b**) 20 μm; (**c**) 50 μm (top); 100 μm (bottom). (**e**) Correlation plot between cumulative axon length around the tdTomato[+] PNEC/SCLC and the perimeter of tdTomato[+] cells. Normal airways: n = 10 PNEC/NEB (3 mice); early SCLC: n = 7 tumors (3 mice); late SCLC n = 10 tumors (4 mice). (**f**) Single channel images of immunostaining from Fig. 2g, left and middle panels. Scale bars from top to bottom: 50, 50, 30, 100 μm. PNEC/NEB: n = 16 PNEC/NEB (3 mice); normal lung: n = 3 mice; early SCLC: n = 10 tumors (3 mice); late SCLC: n = 10 tumors (4 mice). (**g, h**) Single channel images of the immunostaining zoom-ins (right panels) from Fig. 2g (**g**, from the early/small SCLC; **h**, from the advanced/large SCLC). Scale bars: 20 μm (**g**), 50 μm (**h**). Early SCLC: n = 10 tumors (3 mice); late SCLC: n = 10 tumors (4 mice). (**i**) Single channel images of the immunostaining zoom-ins (right panels) from Fig. 2h. Scale bars: 30 μm. n = 10 tumors (3 mice). (**j**) Single channel images of immunostaining of early SCLC innervation from Fig. 2j. Scale bars: 50 μm (left panel), 10 μm (right panel). n = 10 tumors (3 mice). (**k**) 3D reconstruction of axons (β3-tubulin: *yellow*) inside a small SCLC (*magenta*). Scale bars: 8 μm. n = 10 tumors (3 mice). Mean ± SEM shown in all graphs. Ordinary one-way ANOVA applied in **d**. ns: not significant.

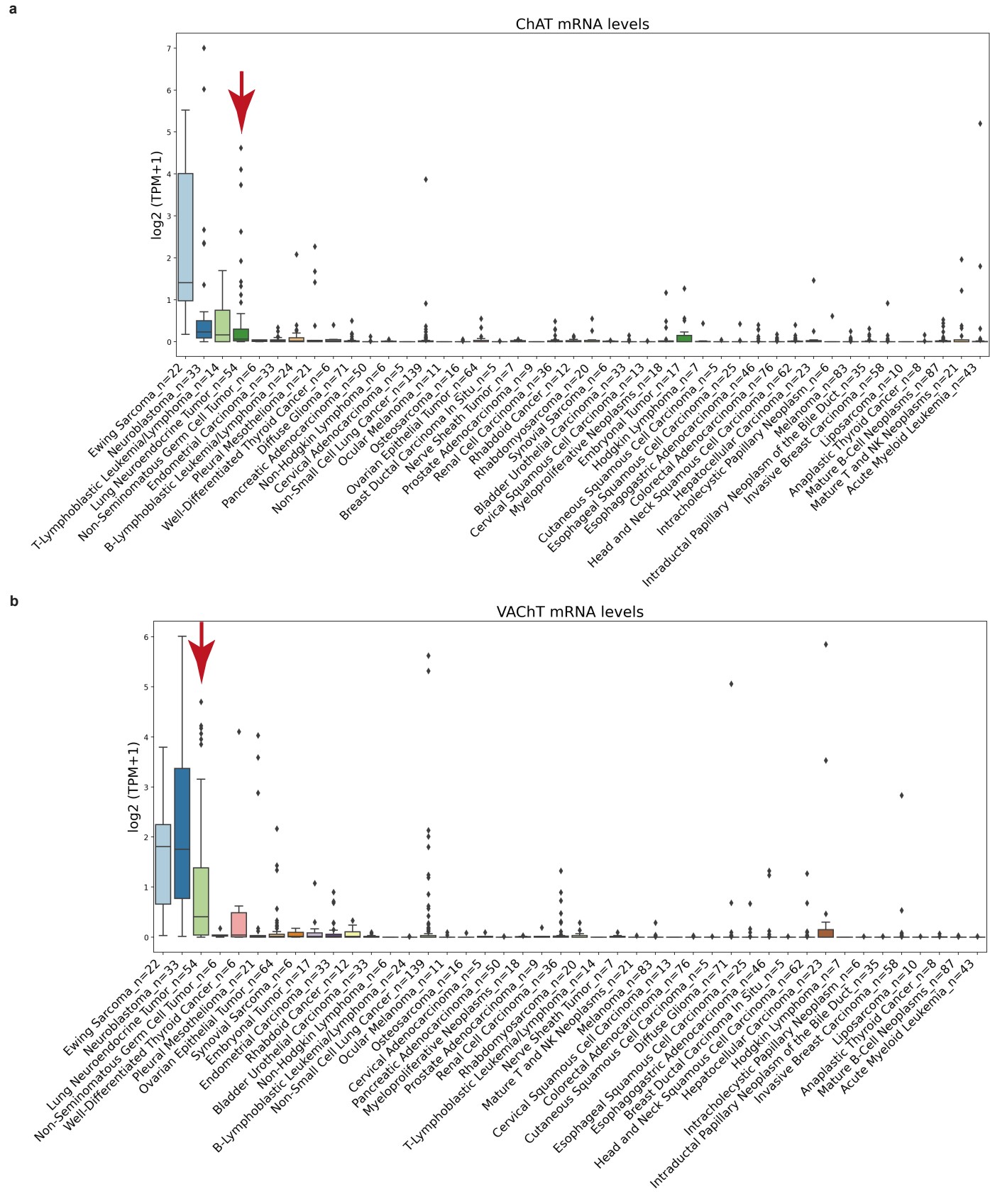

**Extended Data Fig. 5 | Expression levels of cholinergic markers across human cancers.** (a, b) Pan-cancer analysis of different cell lines from various human tumor types obtained from DepMap[88]. Human SCLC cell lines (shown as "Lung Neuroendocrine Tumor" pointed by a red arrow on the boxplots) are among the top 4 or top 3 of the highest CHAT- and SLC18A3-expressing cancer types, respectively. CHAT encodes choline acetyltransferase (ChAT), which is a key enzyme in acetylcholine synthesis; SLC18A3 encodes vesicular acetylcholine transporter (VAChT), which mediates acetylcholine storage in synaptic vesicles. Boxplots show the median (line) and IQR (interquartile range), with whiskers no more than 1.5× IQR. Number of cell lines per tumor type specified next to their corresponding labels.

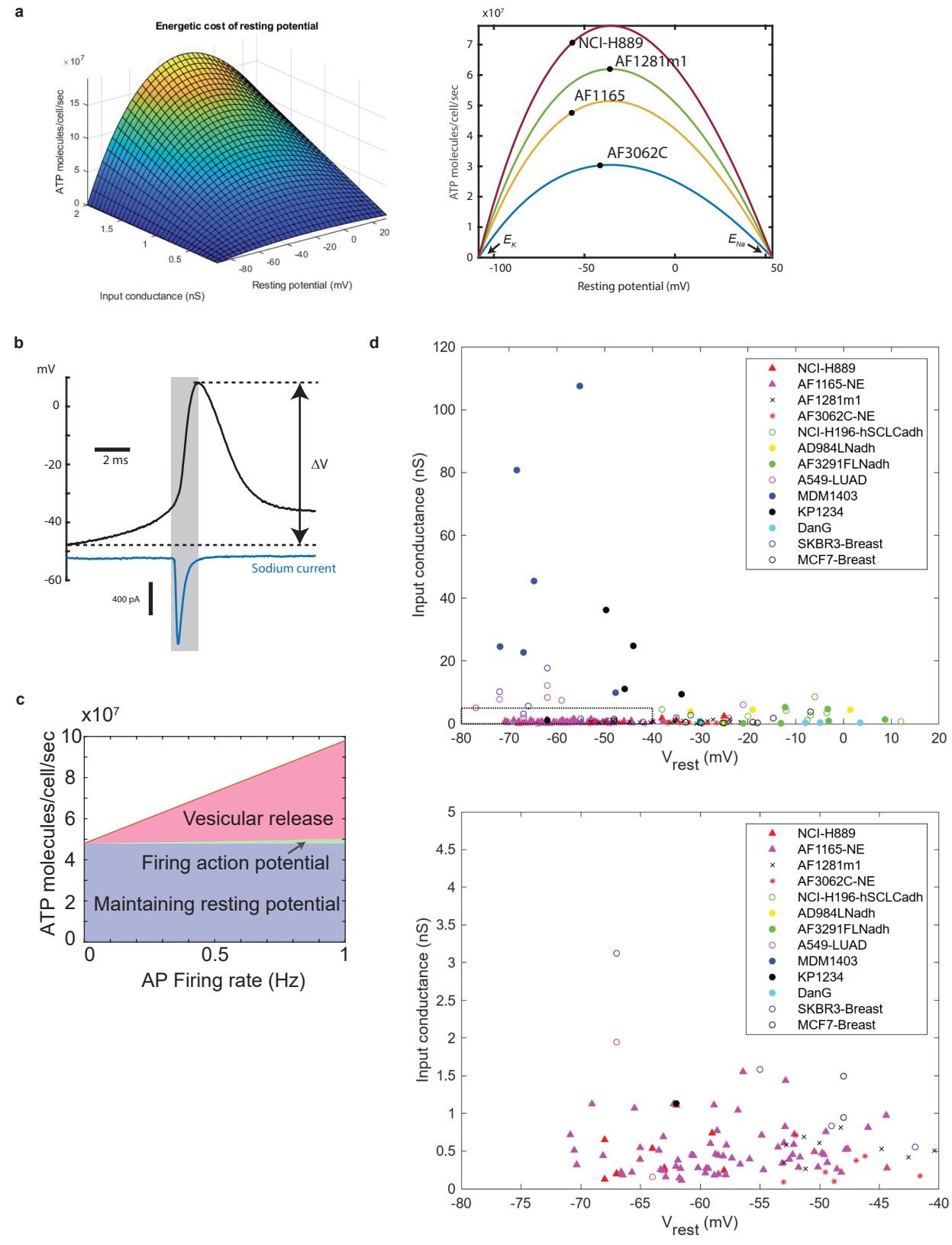

**Extended Data Fig. 6 | Analysis of the energy cost of maintaining the resting potential. (a)** Mathematical modeling of the energy cost of maintaining the resting potential. Average values of input conductance and resting potential ($V_{rest}$) for four different NE lines predict the relationship between membrane potential and ATP consumption (curves), and the specific amounts associated with the observed resting potentials (points). **(b)** Illustration of how the rising phase (gray bar) of the action potential in a NE SCLC cell is driven by sodium current: the resulting charge influx charges the membrane capacitance by an amount $\Delta V$, requiring a sodium charge influx of $C_m \Delta V$, ultimately requiring restoration by sodium pumping. **(c)** Indicative energy budget for electrical signaling, using mean parameters for AF1165 cells as an example. **(d)** Scatter plot of $V_{rest}$ with input conductance in a panel of mouse (n = 7) and human (n = 6) cell lines measured by patch clamp recording. Lower panel: Zoom in on the area within the dotted square in the left lower part of the upper panel.

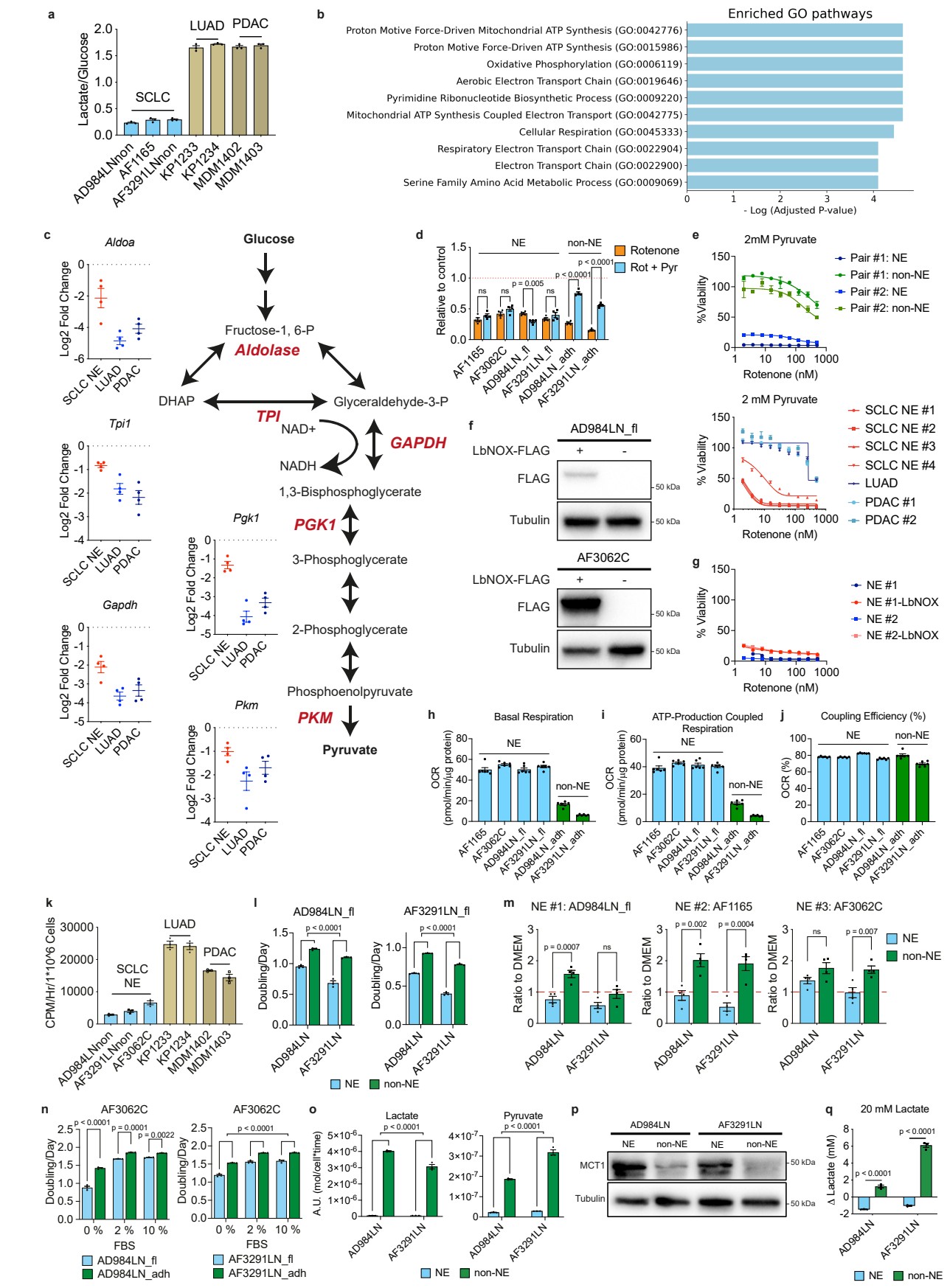

**Extended Data Fig. 7** | See next page for caption.

**Extended Data Fig. 7 | Metabolic characterization of NE and non-NE cells.**
(**a**) Glycolysis index in SCLC NE (n = 3), LUAD (n = 2) and PDAC (n = 2) cell lines, measured by the amount of lactate secretion relative to glucose consumed in DMEM with 2% FBS (n = 3 technical replicates/cell line). (**b**) Gene ontology (GO) enrichment with the top eight candidates from a previous CRISPR screening to identify SCLC preferential vulnerabilities[42]. Fisher's exact test and Benjamini-Hochberg method applied for correction of multiple testing (adjusted p-values shown) (**c**) CRISPR-screen results of genes involved in glycolysis[42,88]. Gene scores (log$_2$ fold change, L2FC) for the indicated genes for SCLC, LUAD and PDAC (n = 4 cell lines/cancer type). (**d**) Normalization of luminescence signals to the control condition (DMEM without rotenone and pyruvate: dotted red line). SCLC-NE (n = 4 cell lines, 4 technical replicates/cell line) and non-NE (n = 2 cell lines, 4 technical replicates/cell line) were treated with 5 or 50 nM rotenone for NE or non-NE, respectively (in the presence or absence of 2 mM pyruvate). Two-way ANOVA with Sidak's multiple comparisons test. (**e**) CTG assay in paired NE and non-NE SCLC cell lines (top), and in a panel of SCLC NE, LUAD and PDAC cell lines (bottom) cultured in DMEM with 2 mM pyruvate, both treated with different concentrations of rotenone (dose-response curves). Pair #1: AD984LN; pair #2: AF3291LN. NE #1-4: AD984LN_fl, AF1165, AF1281m1, AF3062C; LUAD: KP1233; PDAC #1, 2: MDM1402, MDM1403. n = 4 technical replicates/line. (**f**) WB of FLAG in AD984LN_fl and AF3062C cells engineered to express a FLAG-tagged version of LbNOX. (**g**) CTG assay in SCLC NE cell lines (NE #1: AD984LN_fl; NE #2: AF3062C) with or without the LbNOX construct and treated with different concentrations of rotenone (dose-response curves). n = 4 technical replicates/line. (**h-j**) Quantification of the basal (**h**), ATP-linked oxygen consumption rates (OCRs) (**i**), and coupling efficiency (**j**) of SCLC cell lines, being the last one the percentage of mitochondrial OCR dedicated to ATP synthesis. n = 6 technical replicates/cell line, 4 NE-SCLC cell lines and 2 non-NE cell lines. (**k**) Radioactive 2-deoxyglucose uptake in SCLC NE (n = 3), LUAD (n = 2) and PDAC (n = 2) cell lines (n = 3 technical replicates/condition). (**l**) SRB assay of mSCLC NE cell lines cultured in conditioned media from two pairs of NE and non-NE cells, originally derived from the same tumors (AD984LN and AF3291LN). n = 3 technical replicates/condition. Two-way ANOVA with Sidak's multiple comparisons test. (**m**) CTG assay of the 3 SCLC NE cell lines cultured in conditioned media from 2 pairs of NE and non-NE cells, originally derived from the same tumors (AD984LN and AF3291LN). The luminescence value in each condition normalized to those cultured in DMEM with 2% FBS (dotted red lines). n = 4 technical replicates/condition. Two-way ANOVA with Sidak's multiple comparisons test; ns: not significant. (**n**) SRB assay in the mSCLC NE cell line AF3062C cultured with conditioned media from 2 pairs of NE and non-NE cells, originally derived from the same tumors (AD984LN and AF3291LN) using different concentrations of FBS. n = 3 technical replicates/condition. Two-way ANOVA with Sidak's multiple comparisons test applied. (**o**) Lactate and pyruvate concentrations in 2 pairs of NE and non-NE cell lines were normalized to cell number across time (n = 3 technical replicates/cell line). Two-way ANOVA with Sidak's multiple comparisons test applied. (**p**) WB of MCT1 expression in paired NE and non-NE SCLC cell lines. (**q**) Changes in lactate concentration in media with 20 mM lactate of paired NE and non-NE SCLC cell lines (n = 3 technical replicates/line). Two-way ANOVA with Sidak's multiple comparisons test applied. Mean ± SEM shown in all graphs.

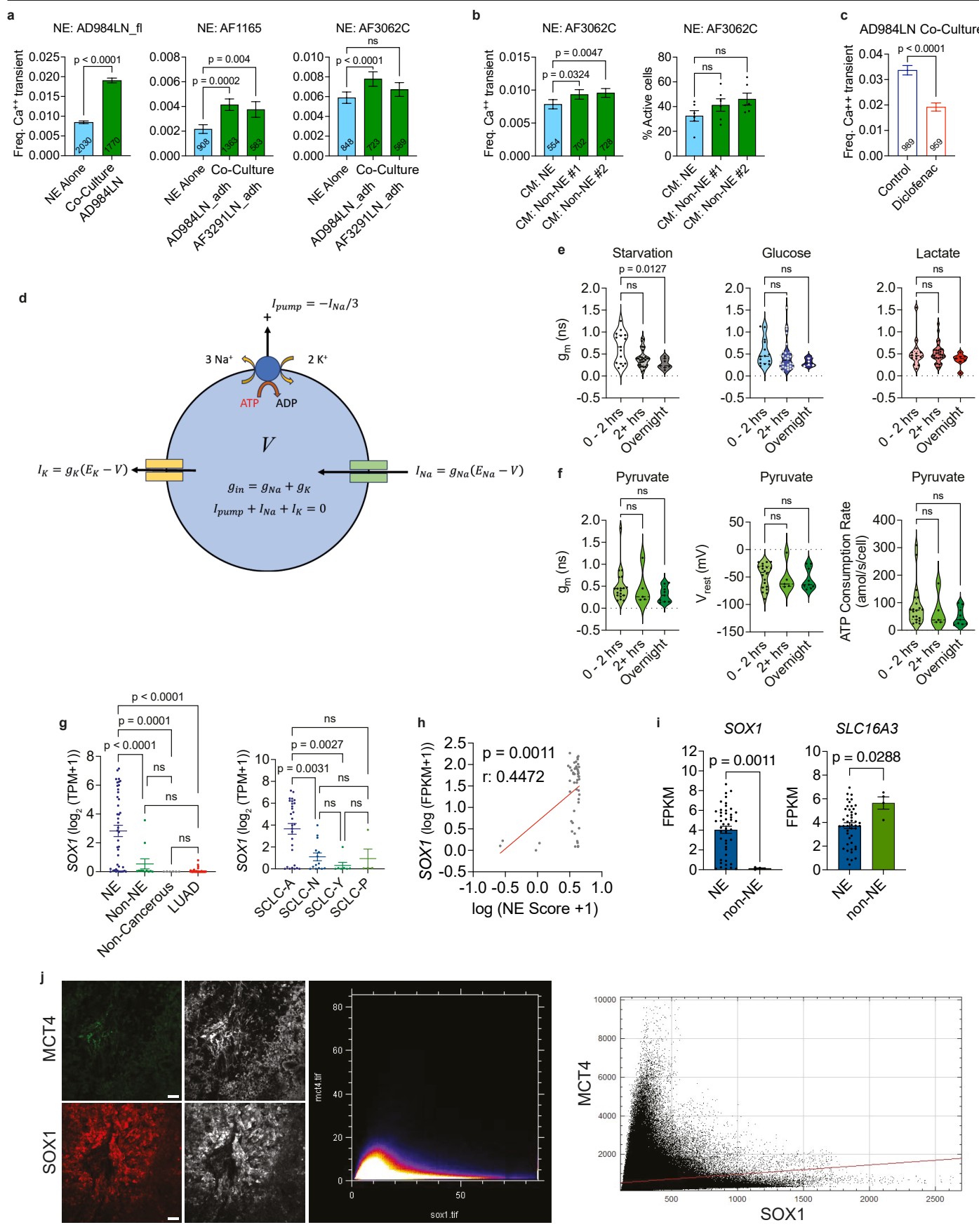

**Extended Data Fig. 8** | See next page for caption.

**Extended Data Fig. 8 | Metabolic influences on electrophysiological properties of SCLC.** (**a**) Frequency of calcium transients in GCaMP6m+ NE cells cultured either alone or with non-NE cell lines. n = number of cells reported in each column over 3 independent experiments. (**b**) Calcium peak frequency in NE cells (left) and fraction of NE cells with active calcium signaling ("peaks") (right) when GCaMP6m[+] NE cells (AF3062C) were cultured in conditioned media (CM) from NE or non-NE cells (non-NE cells: #1: AD984LN_adh, #2: AF3291LN_adh). Left panel: n = number of cells reported in each column over 3 independent experiments; right panel: n = 6 fields of view were analyzed for each condition, each dot represents percentage of active cells per field of view. Frequency of transients is normalized to the number of time frames acquired at 0.5 Hz. (**c**) Frequency of calcium transients in GCaMP6m[+] NE cells co-culture with AD984LN_fl after Diclofenac (0.5 mM) administration. n = number of cells reported in each column over 3 independent experiments. Frequency of transients is normalized to the number of time frames acquired at 0.5 Hz. (**d**) Principle of estimating the energetic cost of the resting potential, by balancing the Na pump current and the Na and K components of the resting leak current. See Methods for details. (**e**) Membrane conductance ($g_m$) in AF1165 NE cells in different conditions. Starvation: n = 15, 31, 7 cells/time point; 5 mM glucose: n = 14, 30, 6 cells/time point; 10 mM lactate: n = 14, 29, 7 cells/time point. (**f**) $g_m$, $V_{rest}$ and estimated ATP consumption in AF1165 cells incubated with pyruvate for different duration. n = 20, 5, 9 cells/time point. (**g**) Left: *SOX1* normalised mRNA counts of human SCLC cell lines (n = 54), human LUAD cell lines (n = 79), and non-cancerous human cell lines (n = 6) from the CCLE (Cancer Cell Line Encyclopedia) provided by the DepMap 23Q2 database. Right: *SOX1* mRNA expression levels across the different hSCLC subtypes (n = 54 cell lines). (**h**) Correlation analysis of *SOX1* mRNA levels and the NE score of previously published PDX models[27] (n = 50 PDX models, Pearson correlation. Two-tailed P-value shown). (**i**) *SOX1* and *SLC16A3* (encoding MCT4) normalised mRNA reads in human PDX cell lines previously published[1]. n = 46 of NE-like cell lines (with NE-Score >0.8), n = 4 of non-NE-like cell lines (with NE-Score <0.2). (**j**) Co-localization analysis between MCT4 and SOX1 in tumors from a GEMM of SCLC. The analysis was performed by ImageJ plugin JaCoP; the intensity of each pixel is plotted on x-axis (SOX1) or on y-axis (MCT4) (left panel). The Pearson correlation coefficient between these two events was calculated (red line, right panel). n = 5 mice (1 tumor/mouse). Scale bar = 20 µm. Mean ± SEM shown in all graphs. Two-tailed Mann-Whitney test applied in **a** ("AD984LN" panels), **c, i** (SOX1). Kruskal-Wallis test with Dunn's multiple comparisons test in **a** (other panels), **b, e, f**. Ordinary one-way ANOVA applied in **g**. Two-tailed unpaired t-test applied in **i** (SLC16A3). ns: not significant.

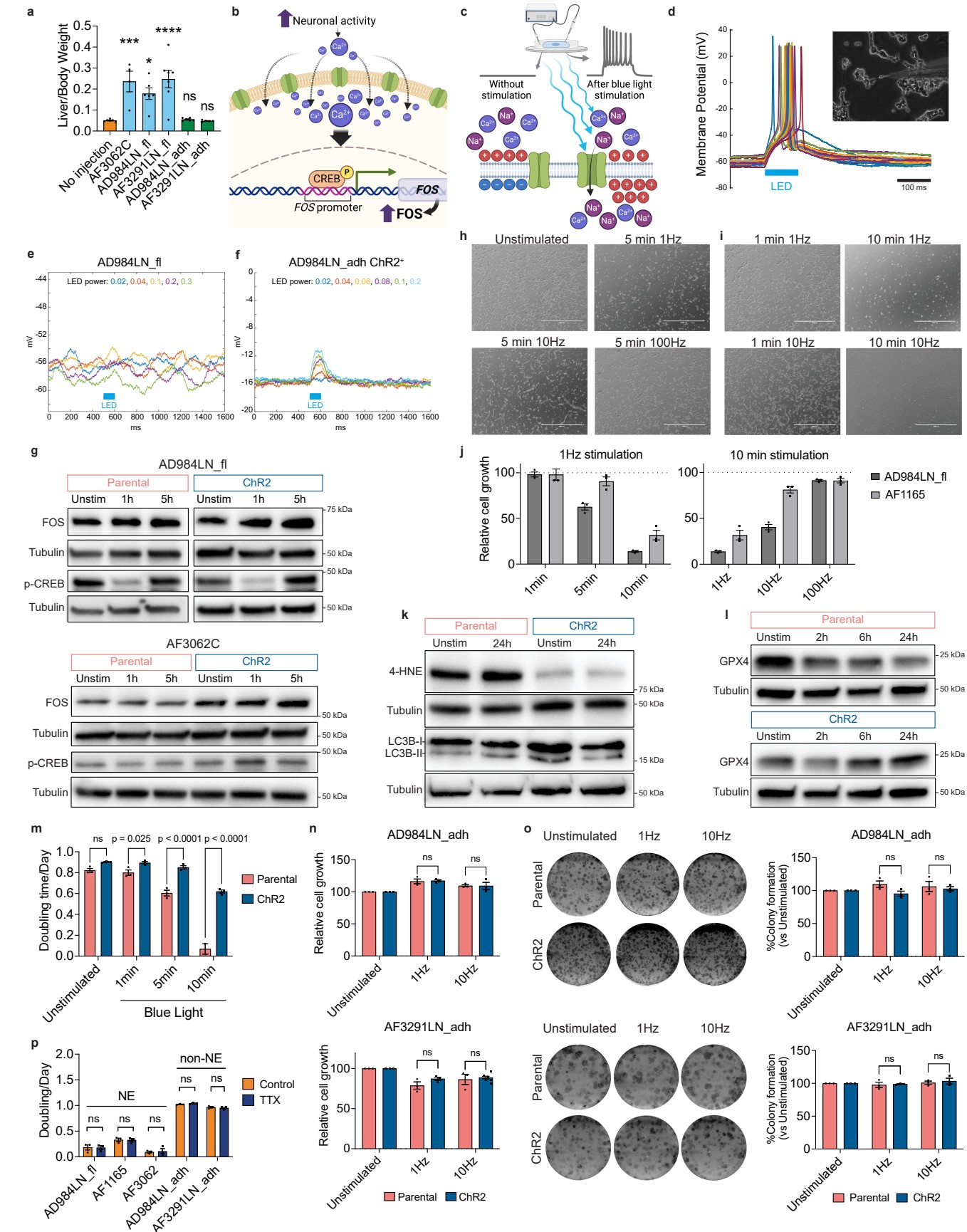

**Extended Data Fig. 9** | See next page for caption.

**Extended Data Fig. 9 | Effects of optogenetic stimulation on SCLC.**
(**a**) Normalized liver weight after liver metastasis assay. NOD-SCID mice were intrasplenically injected with either NE or non-NE cells. Liver weight was normalized to animal weight at injection and compared to non-injected controls. Ordinary one-way ANOVA, Dunnett's multiple comparisons test. *: p = 0.0104; ***: p = 0.0008; ****: p < 0.0001; ns: not significant. n = 6, 4, 6, 7, 6, 5 mice from left to right. (**b**) Induction of FOS expression and CREB phosphorylation are key downstream events following neuron activation and its subsequent calcium influx. (**c, d**) Illustration of optogenetics (**c**) and membrane potential responses recorded from ChR2-expressing SCLC NE cells (AD984LN_fl) in current-clamp mode upon blue light stimulation. Representative of recordings in n = 8 cells. A similar result was observed in ChR2$^+$ AF1165 as well (**d**). (**e, f**) Current-clamp membrane potential recording of ChR2$^-$, parental AD984LN_fl NE cells (representative of n = 3 cells) (**e**) and ChR2$^+$ AD984LN_adh non-NE cells (representative of n = 4 cells) (**f**) upon blue light stimulation. While depolarization was observed in the AD984LN_adh cells due to ChR2 expression, no action potential firing could be elicited. (**g**) Western blot of FOS and p-CREB at different time points after blue light LED stimulation (1 Hz for 10 min) of AD984LN_fl and AF3062C NE cells. (**h, i**) Representative bright field images of NE cells (AD984LN_fl) three days after blue light exposure at different frequencies for 5 min (**h**) and at different frequencies and durations (**i**) Scale bars: 1 mm. n = 3 technical replicates/condition. (**j**) Normalised proliferation rates measured by SRB assay in the mSCLC NE cell lines AD984LN_fl and AF1165 after different exposure times (left) or different pulse frequencies (right) of blue light. The dotted line represents the unstimulated condition. n = 3 technical replicates/cell line. (**k, l**) WB for markers of ferroptosis (the lipid peroxidation adduct: 4-HNE (**k**); reduction of GPX4 (**l**)) and autophagy (LC3B-I/II forms, (**k**)) in parental and ChR2-expressing AD984LN_fl NE lines, with and without blue light exposure. (**m**) SRB assay in AD984LN_fl cells (parental and ChR2) after 10Hz-blue light stimulation. n = 3 technical replicates/condition, also other two mSCLC-NE lines showed similar trends. Two-way ANOVA, Sidak's multiple comparison tests applied. (**n, o**) Blue light exposure had no effect on the ChR2$^+$ non-NE cells, both in the short-term proliferation assay with SRB (**n**) and in the long-term colony formation assay (**o**) n = 3 technical replicates/cell line/condition. Groups were compared with two-way ANOVA, Sidak's multiple comparison tests, ns: non-significant. (**p**) SRB assay in SCLC cells incubated with 1 μM TTX for three days (n = 3 technical replicates/condition). Two-way ANOVA, Sidak's multiple comparison tests applied. Mean ± SEM shown in all graphs. Schematics in **b** and **c** were created using BioRender (https://biorender.com).

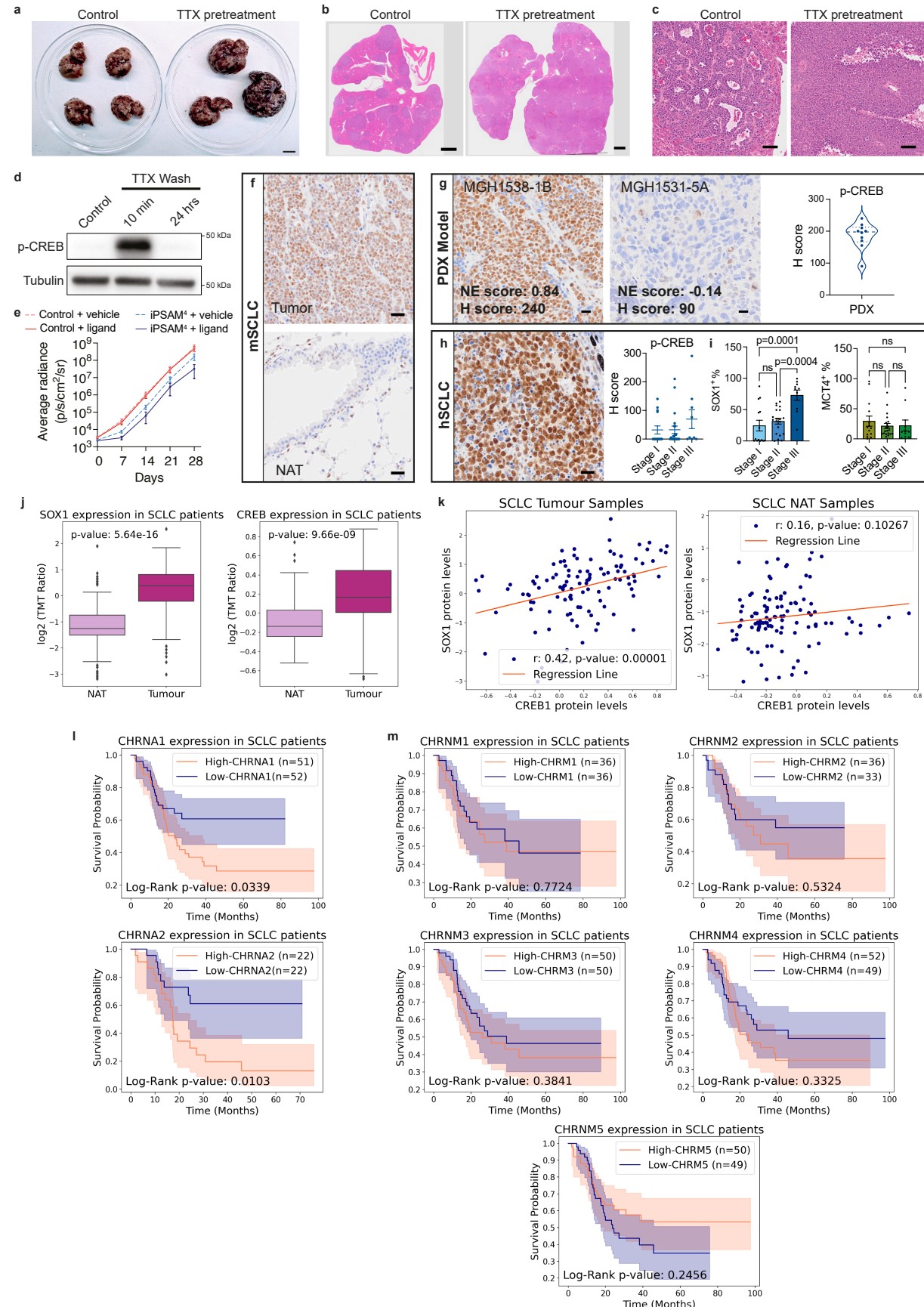

**Extended Data Fig. 10** | See next page for caption.

**Extended Data Fig. 10 | Evaluation of electrical activity in SCLC progression.**
(**a**) Representative images of livers dissected from animals intrasplenically injected with control and TTX-pretreated AF3062C cells relative to Fig. 5f. Scale bar: 1 cm. (**b, c**) Representative H&E images relative to Fig. 5g of livers from (**a**), the same lobes were used for comparison. Scale bars: 2 mm (**b**), 50 μm (**c**). Control (n = 35 tumors/4 mice) and Pre-Tx (n = 29 tumors/3 mice). (**d**) Western blot for p-CREB in control (untreated) and TTX washout (Pre-Tx) AF3062C NE cells. (**e**) Bioluminescence quantification of mice injected with AD984LN_fl cells±iPSAM[4] expression, treated with uPSEM817 or vehicle control (n = 5 mice/group). Mean ± SEM shown. (**f**) Immunohistochemistry (IHC) analysis of p-CREB in an mSCLC tumor from a classical PRP130 GEMM (top), and a large airway from its normal adjacent tissue (NAT) (bottom). Representative images of n = 5 mice ( > 20 tumors). Scale bars: 20 μm. (**g, h**) p-CREB IHC in PDX models (**g**) and clinical SCLC tumor samples (**h**); PDX models with the highest (240) and lowest (90) H scores (**g**) and hSCLC tumor with the highest H score (300) (**h**) were shown. Scale bars: 20 μm. n = 20 patients, 10 PDX models. (**g**, right)

Quantification of p-CREB IHC in PDX models (n = 10). (**i**) Percentage of SOX1[+] cells was increased in advanced-stage hSCLC tumors; percentage of MCT4[+] cells remained stable. Data are presented as the mean ± SEM, stage I: n = 14; stage II: n = 20; stage III: n = 10. Groups were compared by ordinary one-way ANOVA. P-values shown, ns= not significant. (**j**) Boxplots of normalised protein expression values of SOX1 (left) and CREB (right) in tumour and normal adjacent tissue (NAT) samples from a previously published cohort of SCLC patients (n = 112)[56]. Wilcoxon matched-pairs signed rank two-sided test was applied. Boxplots show the median (line) and IQR (interquartile range), with whiskers no more than 1.5× the IQR. (**k**) Correlation analysis of SOX1 and CREB protein levels in tumour (left) and NAT (right) samples from SCLC patients (n = 112). Spearman correlation (two-sided) applied, p-values shown. (**l, m**) Kaplan–Meier curves of a published cohort of SCLC patients[56] (n = 104) according to the mRNA expression of different nAChR (**l**) and mAChR (**m**) subunits. Log-rank test applied, 95% confidence intervals and p-values shown.

2023-10-17704D

# Reporting Summary

## Statistics

For all statistical analyses, confirm that the following items are present in the figure legend, table legend, main text, or Methods section.

| n/a | Confirmed | |
|---|---|---|
| ☐ | ☒ | The exact sample size (*n*) for each experimental group/condition, given as a discrete number and unit of measurement |
| ☐ | ☒ | A statement on whether measurements were taken from distinct samples or whether the same sample was measured repeatedly |
| ☐ | ☒ | The statistical test(s) used AND whether they are one- or two-sided<br>*Only common tests should be described solely by name; describe more complex techniques in the Methods section.* |
| ☒ | ☐ | A description of all covariates tested |
| ☐ | ☒ | A description of any assumptions or corrections, such as tests of normality and adjustment for multiple comparisons |
| ☐ | ☒ | A full description of the statistical parameters including central tendency (e.g. means) or other basic estimates (e.g. regression coefficient) AND variation (e.g. standard deviation) or associated estimates of uncertainty (e.g. confidence intervals) |
| ☐ | ☒ | For null hypothesis testing, the test statistic (e.g. *F*, *t*, *r*) with confidence intervals, effect sizes, degrees of freedom and *P* value noted<br>*Give P values as exact values whenever suitable.* |
| ☒ | ☐ | For Bayesian analysis, information on the choice of priors and Markov chain Monte Carlo settings |
| ☒ | ☐ | For hierarchical and complex designs, identification of the appropriate level for tests and full reporting of outcomes |
| ☐ | ☒ | Estimates of effect sizes (e.g. Cohen's *d*, Pearson's *r*), indicating how they were calculated |

*Our web collection on statistics for biologists contains articles on many of the points above.*

## Software and code

Policy information about availability of computer code

| | |
|---|---|
| Data collection | MicroManager open-source software (version 2.0.0) was used for acquisition of calcium imaging data.<br>For the RNA-seq dataset, Illumina HiSeq 2000 50-nt single-ended reads were mapped to the UCSC mm9 mouse genome build (http://genome.ucsc.edu/) using rsem v1.2.12 and bowtie v1.0.1 with default options.<br>For in vivo bioluminescent imaging, Living Image software 4.8 has been used. |
| Data analysis | -Biophysical modelling of ATP consumption: A custom script in Matlab R2022b Update! (9.13.0.2080170) was used following the model described in the M&M section and can be accessed at https://doi.org/10.6084/m9.figshare.27630099<br>-Calcium Imaging Analysis: ImageJ (version 1.54f) with TrackMate plugin and Cellpose detector was used for image segmentation, tracking of cells and outputing fluorescence intensity measurements for segmented cells at each time point analyzed.<br>A custom script in Matlab R2022b Update! (9.13.0.2080170) was used for finding calcium peaks and outputing peak frequency and peak prominence. The Matlab code for calcium imaging analyses can be accessed at https://doi.org/10.6084/m9.figshare.27630099<br>-RNA-seq analysis: Raw estimated expression counts were upper-quartile normalized to a count of 1000. Given the complexity of the dataset in terms of a mixture of different biological backgrounds, a high-resolution signature discovery approach was employed to characterize global gene expression profiles. Independent Component Analysis (ICA), an unsupervised blind source separation technique, was used on this discrete count-based expression dataset to elucidate statistically independent and biologically relevant signatures as detailed in the M&M section. All RNA-seq analyses were conducted in the R Statistical Programming language (v.3.6.0). Gene set enrichment analysis (GSEA) was carried out using the pre-ranked mode with default settings. Heatmaps were generated using the Heatplus package in R (v.2.26.0).<br>-Graph Pad Prism has been employed for statistical analysis (v. 10.3.0).<br>-In vivo bioluminescent imaging: Living Image software 4.8 has been used.<br>-Immunofluorescence analysis: ImageJ (version 1.54f) with neuronj plugin has been used for cumulative axon length quantification; JaCoP plugin has been employed to calculate the Pearson's correlation coefficient. Qupath 0.5.0 has been used for TMA and IHC quantification. |

Imaris has been used for 3D rendering and visualization (v. 10.1.1).
-Oxygen consumption rates and downstream analyses were performed using Seahorse Analytics (Version 1.0.0-699).

For manuscripts utilizing custom algorithms or software that are central to the research but not yet described in published literature, software must be made available to editors and reviewers. We strongly encourage code deposition in a community repository (e.g. GitHub). See the Nature Portfolio guidelines for submitting code & software for further information.

## Data

Policy information about availability of data

All manuscripts must include a data availability statement. This statement should provide the following information, where applicable:
- Accession codes, unique identifiers, or web links for publicly available datasets
- A description of any restrictions on data availability
- For clinical datasets or third party data, please ensure that the statement adheres to our policy

Raw and processed data of SCLC and PDAC cell lines from the RNA-seq experiments have been deposited in the NCBI Gene Expression Omnibus database (accession number: GSE270281). RNA-seq data of LUAD cell lines have been deposited under the accession code GSE14594584. Expression data from human SCLC cell lines was accessed from the Cancer Dependency Portal (DepMap) (www.depmap.org), specifically from the DepMap Public 23Q2 and the proteomic dataset. The values from this the DepMap Public 23Q2 dataset are inferred from RNA-seq data using the RSEM tool and are reported after log2 transformation, using a pseudo-count of 1; log2 (TPM+ 1). Clinical and expression data from a cohort of 112 SCLC patients were obtained from Liu et al, Cell 187, 184-203 e128 (2024). Expression data and NE Scores from patient-derived xenografts (PDX) models were provided by Dr. Benjamin J. Drapkin.

## Research involving human participants, their data, or biological material

Policy information about studies with human participants or human data. See also policy information about sex, gender (identity/presentation), and sexual orientation and race, ethnicity and racism.

| | |
|---|---|
| Reporting on sex and gender | NA |
| Reporting on race, ethnicity, or other socially relevant groupings | NA |
| Population characteristics | NA |
| Recruitment | NA |
| Ethics oversight | NA |

Note that full information on the approval of the study protocol must also be provided in the manuscript.

# Field-specific reporting

Please select the one below that is the best fit for your research. If you are not sure, read the appropriate sections before making your selection.

☒ Life sciences          ☐ Behavioural & social sciences          ☐ Ecological, evolutionary & environmental sciences

For a reference copy of the document with all sections, see nature.com/documents/nr-reporting-summary-flat.pdf

# Life sciences study design

All studies must disclose on these points even when the disclosure is negative.

| | |
|---|---|
| Sample size | No statistical methods were used to calculate sample size. Sample sizes were chosen based on preliminary experiments aiming to capture the biological effects in line with similar research in the field. |
| Data exclusions | No data have been excluded from the analysis. |
| Replication | The number of replicates for each experiment has been reported in the corresponding figure legend and methods. RNA-seq experiments were performed in 5 or 3 different cell lines for the mSCLC-NE or non-NE group, respectively. |
| Randomization | For cell culture works, samples were randomly assigned to the experimental groups. For in vivo work, age-, litter-, and sex-matched mice were randomly assigned to experimental groups. For imaging analyses, fields of view were randomly selected and when required, allocated to the different experimental groups based on the size of the tumor or the lack thereof. For electrophysiology experiments, cells were randomly selected for subsequent analyses. For human Tissue Microarray (TMA), the whole biopsy sections were analyzed and classified according to their histological stage as provided by the supplier. |
| Blinding | We ensured blinded conduct during data acquisition and the subsequent analysis. When the experimental setup allows it, such as in the intrasplenic injection of cancer cells with different treatments, the experiment was conducted by a researcher who was blind to the experimental groups. |

# Reporting for specific materials, systems and methods

We require information from authors about some types of materials, experimental systems and methods used in many studies. Here, indicate whether each material, system or method listed is relevant to your study. If you are not sure if a list item applies to your research, read the appropriate section before selecting a response.

## Materials & experimental systems

| n/a | Involved in the study |
|-----|----------------------|
| ☐ | ☒ Antibodies |
| ☐ | ☒ Eukaryotic cell lines |
| ☒ | ☐ Palaeontology and archaeology |
| ☐ | ☒ Animals and other organisms |
| ☒ | ☐ Clinical data |
| ☒ | ☐ Dual use research of concern |
| ☒ | ☐ Plants |

## Methods

| n/a | Involved in the study |
|-----|----------------------|
| ☒ | ☐ ChIP-seq |
| ☒ | ☐ Flow cytometry |
| ☒ | ☐ MRI-based neuroimaging |

## Antibodies

**Antibodies used**

Primary Antibodies (dilutions reported in Methods):
MCT4 (Santa Cruz, sc-376140)
SOX1 (Bio-Techne Ltd, AF3369)
VAChT (Synaptic System, 139105)
TdTomato (Sicgen, AB8181-200)
β3 tubulin (Abcam, ab52623)
Phospho-Ser133-CREB (Ser133) (Cell Signaling, 9198)
c-FOS (Abcam, ab190289)
alpha-Tubulin-HRP (Abcam, ab40742)
HES1 (Cell Signaling, 11988S)
MCT1 (Sigma, ab1286-I)
SOX1 (Cell Signaling, 4194)
GPX4 (Abcam, ab125066)
4-hne (Abcam, ab46545)
LC3B (Cell Signaling, 43566)
hsp90 (BD, 610418)
FLAG (Cell Signaling, 2368)
Ki67 (Abcam, ab15580)
Cleaved Caspase 3 (Cell Signaling, 9579)

Secondary antibodies (dilutions reported in Methods):
Goat anti-rabbit HRP-conjugated (Abcam, ab205718)
Goat anti-rmouse HRP-conjugated invitrogen g-21040
Goat anti-chicken HRP-conjugated invitrogen a16054
Donkey anti-mouse Alexa Fluor (AF) 488 (Invitrogen, A21202)
Donkey anti-rabbit AF 488 (Invitrogen, A-21206)
Donkey anti-goat AF 568 (Invitrogen, A-11057)
Donkey anti-goat AF 647 (Invitrogen, A32849)
Goat anti-guinea pig AF 647 (Invitrogen, A21450)

**Validation**

Primary antibodies:
-MCT4 (Santa Cruz, sc-376140) previously validated in GEMM models in the following manuscript (Qian Y et al. (2023). MCT4-dependent lactate secretion suppresses antitumor immunity in LKB1-deficient lung adenocarcinoma. Cancer Cell 41(7):1363-1380.e7. doi: 10.1016/j.ccell.2023.05.015)
-SOX1 (Bio-Techne Ltd, AF3369) previously validated in mouse cortical stem cells as stated by the manufacturers on their website (https://www.bio-techne.com/p/antibodies/human-mouse-rat-sox1-antibody_af3369)
-VAChT (Synaptic System, 139105) previously validated in mouse spinal cord section as stated by the manufacturers on their website (https://sysy.com/product/139105)
-TdTomato (Sicgen, AB8181-200) previously validated in 293HEK cells transfected with cDNA as stated by the manufacturers on their website (https://www.origene.com/catalog/antibodies/tag-antibodies/ab8181-200/tdtomato-goat-polyclonal-antibody)
-β3 tubulin (Abcam, ab52623) previously validated in human cerebellum tissue section as stated by the manufacturers on their website (https://www.abcam.com/products/primary-antibodies/beta-iii-tubulin-antibody-ep1569y-neuronal-marker-ab52623.html)
-Phospho-Ser133-CREB (Ser133) (Cell Signaling, 9198) previously validated in mouse lung section for IHC and in lysates from SK-N-MC cells for WB as stated by the manufacturers on their website (https://www.cellsignal.com/products/primary-antibodies/phospho-creb-ser133-87g3-rabbit-mab/9198)
-c-FOS (Abcam, ab190289) previously validated in HeLa cell lysate as stated by the manufacturers on their website (https://www.abcam.com/products/primary-antibodies/c-fos-antibody-bsa-free-ab190289.html)
-alpha-Tubulin-HRP (Abcam, ab40742) previously validated in HeLa cell Lysate, NIH/3T3 cell lysate and in brain (rat) tissue lysate as stated by the manufacturers on their website (https://www.abcam.com/products/primary-antibodies/hrp-alpha-tubulin-antibody-

dm1a-loading-control-ab40742.html)
-HES1 (Cell Signaling, 11988S) previously validated in mouse spinal cord section as stated by the manufacturers on their website (https://www.cellsignal.com/products/primary-antibodies/hes1-d6p2u-rabbit-mab/11988)
-MCT1 (Sigma, ab1286-I) previously validated by the manufacturers.
-GPX4 (Abcam, ab125066) previously validated in Human embryonic kidney epithelial cell, lung carcinoma epithelial cell whole cell lysate and mouse lysates by the manufacturers on their website (https://www.abcam.com/products/primary-antibodies/glutathione-peroxidase-4-antibody-epncir144-ab125066.html).
-4-hne (Abcam, ab46545) previously validated in frozen mouse cardiac tissue by the manufacturers on their website (https://www.abcam.com/products/primary-antibodies/4-hydroxynonenal-antibody-ab46545.html).
-LC3B (Cell Signaling, 43566) previously validated in HCT 116 and HCT 116 LC3B knockout cells by the manufacturers on their website (https://www.cellsignal.com/products/primary-antibodies/lc3b-e7x4s-xp-rabbit-mab/43566).
-Hsp90 (BD, 610418) previously validated by the manufacturers.
-FLAG (Cell Signaling, 2368) previously validated in COS or 293T cells by the manufacturers on their website (https://www.cellsignal.com/products/primary-antibodies/dykddddk-tag-antibody-binds-to-same-epitope-as-sigma-s-anti-flag-m2-antibody/2368)
-Ki67 (Abcam, ab15580) previously validated in mouse spleen formalin fixed paraffin embedded tissue section by the manufacturers on their website (https://www.abcam.com/products/primary-antibodies/ki67-antibody-ab15580.html).
-Cleaved Caspase 3 (Cell Signaling, 9579) previously validated in paraffin-embedded Jurkat cell pellets by the manufacturers on their website (https://www.cellsignal.com/products/primary-antibodies/cleaved-caspase-3-asp175-d3e9-rabbit-mab/9579).

# Eukaryotic cell lines

Policy information about cell lines and Sex and Gender in Research

| Cell line source(s) | Mouse SCLC (AD984LN_fl/adh; AF1165; AF3062C; AF1281m1; TP2031T2; AF3291LN_fl/adh; PRM2.1a), LUAD (KP1233, KP1234), PaNET (βTC-B6), and PDAC (MDM1402, MDM1403) cell lines were derived from GEMMs as previously described or as described in the method section. Human cell lines (HEK293T; NCI-H69_fl/adh; NCI-H889; NCI-H82; COR-L47; COR-L279) were either obtained from the Cell Services Core Facility depository or purchased from external sources including ATCC. PDX derived cell lines (MGH1505-1A) were obtained as described in the method section. |
|---|---|
| Authentication | Cell lines were authenticated by STR profiling. |
| Mycoplasma contamination | Cell lines were routinely tested for mycoplasma contamination. |
| Commonly misidentified lines (See ICLAC register) | No commonly misidentified cell lines were used in this study. |

# Animals and other research organisms

Policy information about studies involving animals; ARRIVE guidelines recommended for reporting animal research, and Sex and Gender in Research

| Laboratory animals | Mouse. F1 mice (9-19 weeks of age) were obtained by crossing 129S6/SvEv Tac males to C57Bl/6J females imported, respectively, from Taconic (Germantown, NY, USA) and Jackson Laboratories (Bar Harbor, ME, USA). NOD-SCID mice (Prkdcscid, JAX strain #001303, 7-12 weeks of age) were obtained from the Jackson laboratory. SCLC autochthonous mice (8-12 weeks old) harbor Trp53fl/fl, Rb1fl/fl, Rbl2fl/fl and Gt(ROSA)26Sortm14(CAG-tdTomato)Hze alleles (PRP130 model, described previously, see methods) or Trp53fl/fl, Rb1fl/fl, Rbl2fl/fl and Gt(ROSA)26Sortm1.1(CAG-tdTomato/GCaMP6f)Mdcah/J alleles (PRP130-Salsa6f model). PRP130 animals were on a mixed 129S5/C57Bl/6J background. PRM model (11 weeks old) was obtained from mice harbouring Trp53fl/fl, Rb1fl/fl, and Igs2tm1(CAG-Myc*T58A/luc)Wrey alleles (JAX strain #029971) on a C57Bl/6J background. 8–10-week-old Gt(ROSA)26Sortm14(CAG-tdTomato)Hze homozygous mice (Ai14 allele, JAX strain #007908) and NSG-GFP mice (JAX strain #021937) were also used. |
|---|---|
| Wild animals | No wild animals were used in this study |
| Reporting on sex | Age-, litter-, and sex-matched mice were randomly assigned to experimental groups. |
| Field-collected samples | No field-collected samples used in this study. |
| Ethics oversight | All procedures were conducted in accordance with the United Kingdom Animal (Scientific Procedures) Act 1986, approved by the Institutional Animal Welfare and Ethical Review Body (The Francis Crick Institute PPL Review Committee) and conducted under the authority of the UK Home Office approved Project License PP4103600, and approved by the Massachusetts Institute of Technology (MIT) Institutional Animal Care and Use Committee. |

Note that full information on the approval of the study protocol must also be provided in the manuscript.

## Plants

Seed stocks

NA

Novel plant genotypes

NA

Authentication

NA

