## [Peer Review File · Nature]

Intrinsic electrical activity drives small-cell lung cancer progression

Corresponding Author: Dr Leanne Li

Version 1:

Reviewer comments:

Referee #1

(Remarks to the Author)

In this manuscript, the authors describe a really fascinating story of an electrically active, neuronal-like, AP-generating SCLC cell subpopulation (NE cells) that collaborates with non-(active) NE cells to promote malignancy; in turn, the non-NE cells provide metabolic support for the NE cells to meet their high energy demand associated with this neuronal-like behavior (like astrocytes with neurons).

In general, the concept developed here is exciting, raising the possibilities of tumor cell-autonomous neuronal features to a new level, and linking the field of cancer neuroscience to cancer metabolism. The abstract and introduction read great and very convincing. The data provided however does often not live up to the expectations raised. It is not sufficiently experimentally addressed how exactly the NE and non-NE cells collaborate; whether this is really reflecting and impacting intra- (rather than inter-) tumor heterogeneity; how this influences important parameters of tumor biology, which are by and large missing (other than liver weight, see below, and some tumor cell number quantifications) – what about tumor burden throughout the organism, cancer cell proliferation, apoptosis, animal survival etc; and important control experiments are missing, too.

The following points should be addressed:

Major points:

The authors should discuss their findings more in the context of what is known about cancer cell intrinsic electrical activity in other cancers, including glioma. This appears particularly important, since the emerging fields of CNS and extracranial Cancer Neuroscience need to understand the similarities and potential differences better. In this respect I would suggest to discuss the preprint reporting action potentials recorded in a new population of glioma cells with a neuronal profile (Curry, et al. bioRxiv 2023); and also the work about neural-like networks of gliomas and potentially also extracranial cancer entities that also communicates with Ca transients and where neurodevelopmental mechanisms are hijacked, including for generation of an autonomous rhythmic activity (Osswald et al Nature 2015; Hausmann et al Nature 2023). Importantly, while they nicely demonstrate evidence for the former, the authors should provide experimental data whether those latter features of gliomas can be found in SCLC, too. ED Movie 1 could be a strong hint that they might. In any case, they should discuss the implications. Along these lines, the well-known axon-like protrusions of SCLC cells that the authors cite: do they resemble tumor microtubes (TMs) of incurable glioma types? Do these protrusions also interconnect SCLC cells to one communicating syncytium, potentially involving NE and non-NE cells and thus providing an explanation how activity (Ca transients?) generated in one cancer cell subpopulation can govern the entire tumor? Does an anatomical network allow electrical auto-stimulation of the entire tumor? And, as important, is this a biological mechanism how the metabolic symbiosis between NE and non-NE is achieved, complementing the paracrine/autocrine mechanism that is also plausibly supported by the data of Fig. 2,3? This could (and probably should) be experimentally addressed, too, since these questions have broad implications for Cancer Neuroscience and would significantly strengthen this manuscript.

To establish the difference of NE vs non-NE cells with respect to generating APs, the authors establish an experimental setup in which they compare a (limited) number of established cell lines that are more NE vs less (non-) NE. The exciting point of how these differences influence intratumoral heterogeneity would be made much more convincing if they sort and then electrophysiologically compare NE vs non-NE cells from the SAME patient (cell line), ideally by using primary patient-

derived cells. Similarly, this paradigm should wherever possible also be applied to other parts of this manuscript.

Similarly, it would be relevant to learn about differential effects of ChR expression in NE vs non-NE cells, preferably those coming from the same line/patient. Is optogenetically-induced cancer cell excitation in non-NE cells possible, and if (unexpectedly) so, can it transform non-NE cells to NE cells on a transcriptional and functional level (“rescue experiment”)? Otherwise this would be a good control experiment. In this context: Fig. 1h: adequate control experiments (light stimulation without ChR) are missing.

Of similar importance, blue light stimulation of ChR expressing cells is also simulating what happens on the postsynaptic side (induction of EPSCs). In general, are these SCLC cells showing any indications of EPSCs, spontaneous or after electrical stimulation (including e.g. of NE-cell only)? Is optogenetic stimulation inducing EPSC(like) currents and not only APs?

Last but not least, are there any synapses or synapse-like structures between cancer cells? Between NE cells, or – potentially even most interesting in light of the general concept of this study – between NE cells (presynaptic) and non-NE cells (postsynaptic)?

With respect to TTX, one important fundamental experiment is missing: does TTX influence AP generation in SCLC NE cells?

Another crucial experiment that is missing is the proof that TTX removal is indeed increasing hyperactivity/firing frequency in the SCLC cells, as shown for some nonmalignant neuronal cells.

The concept of particularly high energy demand of AP-firing SCLCs is intriguing. However, I am struggling to understand whether the authors claim that the OXPHOS pathway is particularly relevant here, or the lactate shuttle (“metabolic symbiosis”) mechanism. I am also missing data on differential relevance of glycolysis (another way of ATP production) between NE vs non-NE cells.

The authors claim that cholinergic signaling is the key driver of synaptic cell-cell communication between SCLC cells. The data to support this (other than cited previous work) is however very limited and seems to be limited to Fig. 4a – one single recording without statistics. It would be very important to substantially strengthen this part of the manuscript by providing much more robust data on the structural, molecular and electrophysiological level.

The data provided in Fig. 4 (and else) is not really convincingly supporting one key hypothesis (if not conclusion) of the study: that a cancer-autonomous electrical self-stimulation is making SCLC independent from direct neuronal input during cancer progression. In fact, everything cited in favor for this concept is no direct proof, rather circumstantial. One way to address this interesting hypothesis would be to investigate different metastatic sites: are liver metastasis (where the main tumor biology – findings are made) are particularly hypo-/non-innervated, which makes NE cells particularly liver metastasis-proficient – while other metastatic (or primary) sites are much better innervated, which makes the electrical activity of NE cells less important? This should really be addressed in mouse studies covering multiple sites of tumor growth, characterizing innervation patterns and at the same time recording tumor growth kinetics and finally overall animal survival.

Minor points:

Abstract: ...similar to astrocytes...: correct, but this applies to any other non-neuronal cell; the authors might want to stress this point that also implies an even wider relevance of their finding.

Ext Data Fig 1h, but also Fig. 1l and EDF 3e etc.: the quantification of “metastasis” is not convincing; only liver size and weight seems to be determined. EDF 1h: What is the color code of the graph?

Referee #2

(Remarks to the Author)

Small cell lung cancer is a highly aggressive form of lung cancer, characterised by a neuroendocrine phenotype. The precise molecular mechanisms that underlie its aggressive nature are largely unknown and elucidating such mechanism pertinent for designing improved therapeutic interventions for patients. Given that SCLC exhibits neuronal characteristics, Peinado and colleagues, explored whether, like neurons NE cells present within SCLC tumors, exhibit similar features and activities, with a focus on whether they can fire action potentials. The authors explore this question using the well-described NE and non-NE cellular populations present in SCLC tumors and demonstrate that indeed NE cells can fire action potentials and this activity requires a heightened ATP demand. Together, these processes promote tumor progression.

Major points:

1. The authors utilise NE and non-NE cell lines derived from PRp130 SCLC lung tumors (maintained on a mixed genetic background). B6129SF1/J mice are utilised for intrasplenic injection studies, which demonstrate that NE, but not non-NE tumor cells seed metastatic lesions in the livers of recipient mice (Extended data Fig. 1h). The authors attribute this in part to differences in the electrical activity of NE vs non-NE cells. However, given that non-NE tumor cells have been previously shown to express high levels of MHC-I, and fail to engraft tumors when transplanted into immune-competent recipient mice

(PMID: 33707236). The difference therefore seen in metastatic potential may not be solely attributed to differences in the electrical activity of the NE vs non-NE cells. To more directly address this: is a similar difference in metastatic potential seen following transplantation of NE and non-NE cells into immune-deficient recipient mice?

2. Related to the above, the authors show in Fig. 11, that inhibiting the action potential firing in SCLC NE cells using tetrodotoxin (TTX) leads to a suppression in the tumorigenic capacity of SCLC NE cells, what time-point was examined, as the liver burden seen following injection of A3062C cells in Extended Data Fig. 1h, was significantly higher than the control level seen in Fig. 11. Has this experiment been repeated with an additional independent NE cell line e.g., AF1165?

2. "non-NE" human SCLC cell lines were used to determine whether the differences in electrical activity were conserved in human tumor cells. Several of the human SCLC cell lines utilized for this analysis (original classification as YAP1-expressing SCLC subtype by PMID: 30926931), such as H841, SB-5, have recently been re-classified as SMARCA4-UT tumors (BioRxiv; <https://www.biorxiv.org/content/10.1101/2022.10.09.511505v1>) or were misclassified, and thus should not be used as a model of SCLC.

3. Expression of MPC1 and MPC2 is higher in NE cells, compared to their non-NE counterparts (Fig. 2g), however, this expression was variable with some NE lines, exhibiting low (MPC2) to barely detectable levels. Can the authors speculate on the heterogeneity observed and how that may impact/relates to the metabolic capacity of these tumor cell lines?

4. Some of the findings presented in this manuscript are leveraged from a prior CRISPR screen performed by several of the co-authors. However, in several places, additional context/details are warranted to support some of the findings presented. For example, and please mind my ignorance, but it is unclear how the genes, shown in Extended Data Fig. 6 are less sensitive to KO of glycolysis pathway genes.

5. In several of the studies, non-excitabile, PDAC and LUAD cell lines served as "control" samples. While non-NE cells, which the authors also show are non-excitabile are not used/shown and would serve as far superior controls in several of the experiments, and thus for consistency it would be favourable to include, where possible a side-by-side comparison of non-NE cell lines.

6. Quantification of the findings shown in Fig. 4f-k and Extended data Fig. 9 would increase the robustness of the findings. Particularly the expression of MCT4 which looks quite abundant in Fig. 4k, but how reproducible that that pattern of expression across a series of progressed lesions?

Minor points:

1. Light field images are provided of NE cells exposed to varying light pulse frequencies and duration (extended Fig. 2f-g) from which it is concluded that blue light simulation alone resulted in "...duration dependent growth reduction or even cell death". Quantitative evidence should be provided to support this apparent growth reduction and/or cell death, e.g., FACS analysis.

2. Images in Extended Data Fig. 3b do not capture the bottom of the well and could be improved.

3. References missing, NE cells as cells of origin of SCLC – PMID: 21665149 and PMID: 21822053.

Referee #3

(Remarks to the Author)

This paper identifies differences in intrinsic electrical activity between small cell lung cancer (SCLC) cells containing and lacking neuroendocrine (NE) features. The authors provide evidence that excitability is required for some aspects of tumorigenicity of NE cancer cells, including growth after implantation into the liver. They also demonstrate metabolic differences between NE and non-NE cells, particularly a reduced rate of lactate secretion and increased sensitivity to oxidative phosphorylation (OXPHOS) inhibitors in NE cells. The authors postulate that an increased dependence on OXPHOS helps NE cells meet the metabolic demands of generating action potentials. They show that unlike non-NE cells, NE cells can undergo net lactate uptake, and they propose a model akin to the astrocyte-neuron lactate shuttle (ANLS), in which glycolytic non-NE cells secrete lactate, which NE cells take up and oxidize to meet their energetic demands. The paper proposes an interesting mode of cooperativity between NE and non-NE cells within the same tumors. However, key aspects of the model are not supported by the data, and in other areas the descriptions of the experiments are inadequate. Specific comments:

1. It is unclear how the authors obtained the GEMM-derived NE and non-NE cell lines. Were these from independent clones from the same isogenic GEMMs?

2. Independent human SCLC cell lines with high or low NE scores are also assessed, and the NE score is correlated with electrical activity. But these experiments are susceptible to idiosyncrasies unrelated to the NE state. A better experiment with more relevance to the idea that ITH supports the growth of NE cells would be to select NE and non-NE cells from the same parental population, then compare electrical activity between the two sub-populations. In several human SCLC cell lines, straightforward methods have been described to enrich the NE and non-NE populations (see Pongor LS et al., *Cancer Discovery* 2023, PMID 36715552; Cai L et al., *Communications Biology* 2021, PMID 33750914).

3. For perspective/scale, can the authors show a typical current-clamp recording of neurons in Fig. 1?

4. SCLC NE cells express high levels of ATP synthetase subunits, but expressing ATP synthetase alone (i.e. without components of the ETC) would have little effect on ATP production. Did this proteomics experiment also reveal increased expression of other components of the ETC in NE vs non-NE cells? This is worth knowing, because in some contexts ATP synthetase has functions distinct from its ability to produce ATP.

5. In the experiments describing ATP synthetase and OXPHOS dependence, the authors seem to imply that NE cells have higher respiration/higher OXPHOS activity than non-NE cells, but this is never directly shown. It is conceivable that NE cells have enhanced dependence on OXPHOS without having higher OXPHOS rates. A direct comparison of OCR between NE and non-NE cells would close the loop on these experiments.

6. If true, it is important that OXPHOS dependence is unrelated to NAD⁺ recycling. But these experiments are superficial. What happens to the NAD⁺/NADH ratio in rotenone-treated NE cells with and without pyruvate? The argument would be

more compelling if the ratio changed when sensitivity to rotenone did not. Another way to exclude the role of NAD⁺ recycling would be to test the effect of LbNOX on rotenone-treated NE cells. This bacterial enzyme converts NADH to NAD⁺ independently of ETC complex I.

7. Key experiments related to the proposed ANLS-like mechanism are missing. First, is adding lactate to dialyzed serum sufficient to drive proliferation of NE cells, as in Fig. 3e? And how does adding pyruvate compare to adding lactate in these experiments? Second, if NE cells do not express MCT4, do they express MCT1 or some other monocarboxylate transporter to allow them to take up lactate? Third and related, the role of the lactate shuttle in NE cell excitability and growth is not tested. The key experiment would be to inactivate lactate import and test whether this suppresses some of the features of NE cells described in the paper (electrical activity; ATP levels; the ability of electrical activity to stimulate growth; etc).

Minor:

1. Line 261 says that NE cells have an increased "fraction of glucose utilized for OXPHOS," but this was not demonstrated. The NE cells secrete a lower fraction of glucose as lactate, but glucose has more metabolic fates than lactate and the TCA cycle.

2. The images in Fig. 3d and ED Fig 7f are difficult to interpret. The authors need a better way of arguing that SOX1 and HES1 are expressed in different cells than MCT4.

3. In Fig. 3g, please confirm in the legend that the lactate-supplemented medium also contains glucose.

Referee #4

(Remarks to the Author)

The manuscript by Peinado et al. addresses the relation between cellular heterogeneity and electrical activity in small-cell lung cancer (SCLC), and uncovers novel links to metabolic symbiosis between SCLC cells of neuroendocrine phenotype (NE) and non-neuroendocrine cells (non-NE). The study reports novel data that help clarify several open questions in the field of SCLC and particularly with respect to the role of electrical activity in driving tumor growth and progression, which they now report to be specific to NE cells. NE cells were reported to acquire a neuronal phenotype, while non-NE cells were reported to mimic astrocyte-like cells of the nervous system. Employing a number of novel techniques, including single cell sequencing-based cell classification, metabolic assays, electrophysiology and modelling of cell activity data, the study reports that NE and non-NE cells play non-redundant roles in tumor progression, whereby lactate derived from non-NE cells supports the energy metabolic demands of NE cells, which are high because electrical activity of NE cells requires large ATP consumption to maintain membrane potential. Tumor heterogeneity developing over the course of tumor growth was suggested to establish a self-sustaining system of symbiosis between NE and non-NE cells. A large majority of analyses are performed on cultured tumor cell lines and tumor cells classified via gene expression analyses.

The study uncovers several highly interesting principles and puts forward novel concepts. There is the promise of major conceptual advance, particularly if the data can be substantiated in a more in vivo setting. The study has the potential to make a major impact, not only on the field of SCLC, but also other types of cancers containing cells of NE origin. The following major and minor points can be considered.

Major points:

1. As such, it is known since several decades that both cell lines and tumors of the lung can show Ca²⁺ transients and electrogenesis. The authors have done a highly thorough and excellent job in electrophysiological analyses to uncover that NE type cells in SCLC (and other types of tumors, such as pancreatic neuroendocrine cells) but not non-NE cells possess the ability to generate electrical activity and calcium transients, leading to neuronal plasticity-like processes, indicating that this is a property that is specific to certain cells in a heterogenous environment. However, all of the work is based upon current injections in patch clamp analyses, which simulates an artificial situation where cells are driven to spike. Also, although spontaneous activity was confirmed as a phenomenon that is known previously, a major open question in the field remains unanswered as to what leads to natural electrical activity in cancer cells. Nowhere in the manuscript does it become why NE cells spike naturally, nor is activity analysed in the natural setting of a tumor. Which processes lead to natural depolarization of cancer cells? Which channels mediate electrical activity and why? A number of previous studies have reported expression of sodium, calcium and potassium channels in tumor cells, including SCLC, while others have contradicted a role for electrical signals driven by channel activity in tumor progression. While the beauty of this study is its allocation of electrical activity and the downstream sequelae to specific cell types within SCLC tumors, it needs to work out the mechanisms as to how such activity can come about in the first place in a real tumor setting (i.e. independent of the cell culture media employed etc.).

2. Another second concern is that SCLC cells are very broadly and somewhat crudely classified as NE and non-NE. All of the functional experiments imply that the non-NE population is a homogenous population that does not possess electrical activity, but is there to promote the growth of NE cells by providing lactate as a fuel. However, it is well established via single cell sequencing analyses both in cell lines and in vivo tumors that the non-NE population is actually very heterogenous. Based on gene signatures, such as ASCL1, NEUROD1, and POU class 2 homeobox 3 (POU2F3), several subclasses are known. Heterogeneity is also given by origin (PNEC, tuft cells, chemosensory epithelial cells etc.). A more mechanistic and refined analysis would be timely, rather than a general NE vs. non-NE classification.

3. The optogenetic experiments claiming a functional role for electrical activity in driving tumor progression in colony forming in vitro assays and organ cultures addressing hepatic metastases are not conclusive. A major caveat, the authors report, is given by toxicity induced by blue light LEDs and the authors rely on complicated explanations and indirect observations to

account for paradoxical observations. However, this should be easy to circumvent. In this era of multicolor optogenetics, it is easily feasible to use red-shifted variants of channelrhodopsin or even other substitutes for excitatory manipulations. In the same vein, the authors find results contrary to their expectations with the use of TTX to silence neuronal activity and attribute it to homeostatic plasticity. To substantiate the latter, the authors will need to provide a series of additional evidence in the vein of whether all of the molecular elements mediating classical homeostatic plasticity are actually present in SCLC NE cells. Moreover, elsewhere in the manuscript, TTX treatment does not evoke homeostatic plasticity, but gives expected results. Again, it is advisable to use additional tools to circumvent the technical issues that currently occlude unequivocal interpretations, e.g. any of the myriad of optogenetic or chemogenetic activity silencing tools. This is a critical point that needs to be adequately addressed to support functional claims.

4. Along the lines of point 1 and point 4 above, while the authors attempt to make a case for neuron-like activity in NE cells in driving tumor growth, there are no mechanistic insights given in to how this takes place. CREB phosphorylation is reported, but how this or other pathways lead to cancer progression is not clarified. This is a critical deficit, also in the light of several previous studies that have already speculated on the ability of electrical activity to promote tumor growth. Key mechanistic analyses are needed to extend beyond previous studies and enable novel advance.

5. The remaining part of the study then focusses on major energy demands inflicted on NE cells by electrical activity. Here, the authors have performed a series of excellent experiments to propose models and address the role of Oxphos and glucose-pyruvate-lactate metabolism and shuttling of lactate from non-NE to NE cells. The data are exciting and the model they propose is indeed intriguing. Here, two questions arise: (i) the authors refer to these interactions as 'metabolic symbiosis', which would imply that both types of cells benefit from each other. While the study reports on how non-NE cells may be providing lactate to vulnerable, energy-depleted NE cells, like astrocytes provide for neurons, it does not become evident what non-NE cells derive from NE cells. Why is this a mutually beneficial arrangement and how does it promote intratumor heterogeneity over time?

6. At the end of the manuscript, nicotinic cholinergic signaling is suddenly brought into picture, but the morphological analyses and in vitro experiments are sketchy at best. Potentially unrelated observations are strung together, and no evidence given in in vivo setting although the authors describe the availability of appropriate animal models. Based on presence of cholinergic innervation alone, major deductions are made. The reviewer would advise to either work this out in more detail or to drop this part. It is correlative at best, and does not give any functional or mechanistic insights.

7. A major concern is that most of the data are restricted to cell lines and in vitro experiments. Can key findings be substantiated in a more natural, in vivo setting? For example, human SCLC types vary between NE-high and NE-low. Can the authors use biopsies from heterogenous types of SCLC as a basis to test the differential roles they ascribe based on in vitro analyses to NE and non-NE cells?

8. Finally, there is some concern about the low number of data points in several figures. N numbers are given only occasionally and mostly entail n of 3 technical replicates. What about biological replicates? In many experiments, the data points shown by individual dots on bar graphs (which are often too difficult to spot or to clearly decipher) appear to be very few – 2 or 3. By using appropriate means to make transparent and clear figures, these critical pieces of information can be made more clearly visible. Moreover, small n numbers need to be accounted for and ideally, substantiated in a more reliable data cohort.

Other comments:

(i) By the same token, data shown in Fig. 4f-k deserves to be in extended figures at best. Particularly the stainings for β tubulinIII and ChAT in panel 4h, at the magnification shown, do not reveal any insights. Moreover, it is surprising to see that putative cholinergic nerves show 100% identity with a marker for sensory nerves. The use of the term 'neurons' in the legend is incorrect – these are axons or efferents, not somata. Because the data do not show any contacts between the axons and the putative tumor cells, one does not understand what this figure achieves.

(ii) Similarly, pCREB data and Sox/MCT4 data are purely qualitative. How many patients/mice show these findings? Do they stand up to statistical analyses?

Version 2:

Reviewer comments:

Referee #1

(Remarks to the Author)

The authors have responded well to my previous concerns. I apologize for a mis-wording that caused some confusion (..cholinergic signaling is the key driver of synaptic cell-cell communication: I did not mean "synaptic" here of course).

The manuscript has benefitted significantly from the additions of so many control experiments and methodologies which present a tour-de-force of Cancer Neuroscience – relevant technologies today. The authors also made changes that allowed to understand some of their concepts and conclusions better.

Having said that, the additions to my and the other reviewers' concerns increased the problem that this manuscript is overall a difficult read, which is partly due to the extensive number of experiments performed, but also by data interpretation, presentation, ordering, and conclusions made. This manuscript continues to look like many studies combined into one paper. For this reviewer, it is really challenging to delineate a clear path out of this situation. One possibility would be to focus on the cancer cell intrinsic aspect and leave the (putative, see below) relation to neuron-cancer interactions for a future study.

The EPSP-like potentials in NE cells without cell contacts are puzzling, and in contrast to the many studies describing those potentials in tumor cells with similarly high input resistance and still only recorded in co-cultures with neurons but not in monocultures. Either the authors provide extensive ephys data what is exactly going on here (which single inward channels open, etc.), or they need to delete Fig. 1i and EDF 2r. Anyway, those two do not contribute to the story in a relevant way.

It remains unclear what the tumor biological effect of the AP-generated intercellular Ca transients is. A diffusible factor is plausible, but do the authors suggest this is indeed Ach (this is at least how the story is presented)? I cannot see how, and evidence for this is missing. It is much more likely that these are extracellular Ca waves mediated by e.g. ATP/purinergic receptors, a well established mechanism of Ca wave propagation in unconnected cells. The authors should test this. In fact, later on in the manuscript, they are making first experiments into this direction (Extended Data F 5l,m), but only in the context of NE depolarization. Thus, I would also recommend to re-order the presentation of these datasets, too. Moreover, do the non-NE cells also respond to the activation from NE cells, another interesting mechanism how both cancer cell subpopulations interact?

Fig. 2a,b: why are no AP's recorded here in NE cells? Wouldn't this be expected after strong (like with CCh) Ach receptor stimulation?

To prove that "classical synapses" exists between sensory neurons and SCLC cells, the authors need to go beyond rabies virus tracing, which is not a definite proof of synaptic connections but can simply show spatial proximity. EM and electrophysiology of acute sections of the tumor tissue is needed. Otherwise (since this is not a key aspect of this manuscript) they would need to discuss this data accordingly and phrase it much more cautiously.

Fig. 2m: the authors should clarify better in the text that the beta3 tubulin staining is not showing neuronal structures here but is a tumor cell-specific staining pattern.

It is not clear why the authors assume that the Ca²⁺ activity measured by GCamp6 signals in fresh tumor-bearing lung slices is an indication of "spontaneous" (i.e. tumor-intrinsic) electrical activity. It could be that those are generated by neuronal input, synaptic or otherwise, by the remaining (if transected) nerve endings in the tissue, or even neuronal somata that can be newly formed in tumor environments (see the works of C. Magnon and others).

Along these lines, it would really strengthen one major conclusion of this manuscript if the early tumor growth-promoting influences of neuronal innervation (namely cholinergic) that is not relevant for later tumor growth would have been supported by any data stemming from interventional experiments: in vivo growth differences under early vs late vagotomy, for example, or pharmacological or genetic interference. All data provided in this direction is descriptive. As it is, this concept remains hypothetical (albeit an interesting and even plausible hypothesis). When no more data in this direction is provided, the authors should reconsider the last sentence of the abstract, for example.

Referee #2

(Remarks to the Author)

I take this opportunity to thank the authors for performing an extensive amount of additional experiments to address the concerns raised by myself and the other reviewers. These additional experiments and clarification have enhanced the impact of the work. Based on this, the paper is acceptable for publication.

Referee #3

(Remarks to the Author)

This is a strong revision and the authors have addressed my critiques.

Referee #4

(Remarks to the Author)

The authors have adequately responded to my questions and comments. This is a very interesting study that will have impact on the field.

Version 3:

Reviewer comments:

Referee #1

(Remarks to the Author)

The authors have again responded very well to my points and concerns. Although the manuscript remains a difficult read, that might be the best possible.

All in all, this is a strong manuscript that will definitely contribute significantly to our understanding of SCC pathobiology and will also contribute to the field of Cancer Neuroscience in a very meaningful way.

If have only three remaining points:

1. It is certainly not beyond the scope of the manuscript to test whether ATP or ACh or K⁺ ions are responsible for the intercellular Ca waves that are clearly conducted extracellularly. There are many drugs available that can easily be used to interfere with these three possibilities in in vitro experiments. Since this is not an unimportant point - how does the "tumor organism" of these two major cell types function? - I would still find it very nice data to have in the manuscript. However, if this is completely beyond the capabilities or would require many months for the authors (...?), this is not mandatory for manuscript acceptance in my view.
2. Mechanism of how both cancer cell subpopulations interact: I was more thinking into the direction of intercellular Ca waves, too. Any data on this?
3. please rephrase: ...of termin boutons, which might point towards synapse formation at nerve endings (or a similar cautious phrasing).

Response to Referees:

We thank the editor and the reviewers for your comments and feedback to improve this manuscript. In response to the critiques and suggestions of the reviewers, we have performed additional experiments and extensively revised the manuscript. This includes the addition of two new Main Figures (6 in total) and one new Extended Data Figure (10 in total). The manuscript now covers three major themes that are presented in the following order:

A. Electrophysiological characterization of SCLC cells and mechanism of calcium wave initiation and propagation (Figure 1, 2)

B. Excitable NE cells have unique metabolic phenotypes compared to non-excitable cells: they are highly dependent on an ability to do OXPHOS and benefit from metabolic support by non-NE cells to support their electrical activity (Figure 3, 4)

C. Electrical activity directly promotes SCLC progression both *in vitro* and *in vivo* (Figure 5, 6)

Major additional experimental data included in the revised manuscript is summarized below:

1. Electrophysiology:

- a. Electrophysiology analysis of the excitability and resting membrane potential in paired mouse and human NE/non-NE cell lines and PDX models
- b. Detailed characterization of ion channels expressed on NE cells
- c. Paired patch clamp recording to identify potential synaptic interaction between NE cells

2. Human data:

We have included a large amount of human data, including RNA and protein expression in hSCLC tumors and PDX models, as well as survival analysis in hSCLC patients

3. GEMM data:

- a. Extensive characterization and quantification of changes in SCLC protein expression and innervation patterns at different stages of SCLC progression
- b. *Ex vivo* imaging on fresh lung slices to demonstrate spontaneous calcium activity in autochthonous SCLC
- c. Validation that abolishing metabolic support by an MCT4 inhibitor suppressed SCLC electrical activity by *ex vivo* imaging

4. *In vivo*, experimental validation of electrical activity in SCLC progression:

Provided extensive *in vivo* validation of electrical activity in SCLC metastasis using transplantation models: additional NE cell line for the TTX experiment for tumor burden chemogenetic manipulation of NE electrical activity in a liver metastasis assay

5. Chemogenetics:

Modulation of the electrical activity in NE cells *in vivo* using PSAM

6. Additional metabolic assays:

- a. Measurement of the oxygen consumption rates in NE and non-NE cells
- b. Validation of metabolic support model using MCT1 and MCT4 inhibitors

Referee #1 (Remarks to the Author):

In this manuscript, the authors describe a really fascinating story of an electrically active, neuronal-like, AP-generating SCLC cell subpopulation (NE cells) that collaborates with non-(active) NE cells to promote malignancy; in turn, the non-NE cells provide metabolic support for the NE cells to meet their high energy demand associated with this neuronal-like behavior (like astrocytes with neurons).

In general, the concept developed here is exciting, raising the possibilities of tumor cell-autonomous neuronal features to a new level, and linking the field of cancer neuroscience to cancer metabolism. The abstract and introduction read great and very convincing. The data provided however does often not live up to the expectations raised. It is not sufficiently experimentally addressed how exactly the NE and non-NE cells collaborate; whether this is really reflecting and impacting intra- (rather than inter-) tumor heterogeneity; how this influences important parameters of tumor biology, which are by and large missing (other than liver weight, see below, and some tumor cell number quantifications) – what about tumor burden throughout the organism, cancer cell proliferation, apoptosis, animal survival etc; and important control experiments are missing, too.

The following points should be addressed:

Major points:

The authors should discuss their findings more in the context of what is known about cancer cell intrinsic electrical activity in other cancers, including glioma. This appears particularly important, since the emerging fields of CNS and extracranial Cancer Neuroscience need to understand the similarities and potential differences better. In this respect I would suggest to discuss the preprint reporting action potentials recorded in a new population of glioma cells with a neuronal profile (Curry, et al. bioRxiv 2023); and also the work about neural-like networks of gliomas and potentially also extracranial cancer entities that also communicates with Ca transients and where neurodevelopmental mechanisms are hijacked, including for generation of an autonomous rhythmic activity (Osswald et al Nature 2015; Hausmann et al Nature 2023). Importantly, while they nicely demonstrate evidence for the former, the authors should provide experimental data whether those latter features of gliomas can be found in SCLC, too. ED Movie 1 could be a strong hint that they might. In any case, they should discuss the implications. Along these lines, the well-known axon-like protrusions of SCLC cells that the authors cite: do they resemble tumor microtubes (TMs) of incurable glioma types? Do these protrusions also interconnect SCLC cells to one communicating syncytium,

potentially involving NE and non-NE cells and thus providing an explanation how activity (Ca transients?) generated in one cancer cell subpopulation can govern the entire tumor? Does an anatomical network allow electrical auto-stimulation of the entire tumor? And, as important, is this a biological mechanism how the metabolic symbiosis between NE and non-NE is achieved, complementing the paracrine/autocrine mechanism that is also plausibly supported by the data of Fig. 2,3? This could (and probably should) be experimentally addressed, too, since these questions have broad implications for Cancer Neuroscience and would significantly strengthen this manuscript.

We appreciate the reviewer's suggestions and in the revised manuscript have included new discussion about these landmark glioma papers. We have added extensive new data exploring the nature and mechanisms of the spontaneous activity in SCLC. Firstly, we show, using simultaneous patch-clamp and calcium imaging, that individual action potentials produce unitary calcium transients (**Extended Data Fig. 2k - 2n**). Secondly, we show that there is ongoing asynchronous and synchronous calcium activity, as well as large-scale calcium waves which can be resolved in lightsheet microscopy recording of 3D tumouroids (**Extended Data Fig. 1p, 1q, Extended Data Movie 2**). Moreover, these waves can be triggered by electrical stimulation (**Fig. 1h, Extended Data Movie 3**). We show further that, *unlike mechanisms identified in glioma*, propagation of electrical activity is unlikely to occur via gap junctions or chemical synapses, as paired patch-clamp recordings of adjacent cells show no evidence of such connections (**Fig. 1j, 1k**), and we observe propagation of calcium waves across cell-free gaps in planar cultures (**Fig. 1n**). This implicates volume transmission by calcium-dependent secretion of diffusible factor(s) as reported, for example, in astrocyte calcium waves (Bowser & Khakh, 2007, *J. Gen. Physiol.* 129:485-491), and we suggest that these at least include ACh (**Fig. 2b**) and ATP (**Extended Data Fig. 5m**), both of which we show cause calcium transients when applied exogenously.

[Line: 201]

Collectively, these observations argue that diffusible factors are key in the initiation and propagation of calcium waves in SCLC NE cells, distinct from the mechanism of wave propagation mediated by tumor microtubes and gap junctions as reported in glioma^{40,41}.

40 Osswald, M. et al. Brain tumour cells interconnect to a functional and resistant network. *Nature* 528, 93-98 (2015). <https://doi.org:10.1038/nature16071>

41 Hausmann, D. et al. Autonomous rhythmic activity in glioma networks drives brain tumour growth. *Nature* 613, 179-186 (2023). <https://doi.org:10.1038/s41586-022-05520-4>

To establish the difference of NE vs non-NE cells with respect to generating APs, the authors establish an experimental setup in which they compare a (limited) number of established cell lines that are more NE vs less (non-) NE. The exciting point of how these differences influence intratumoral heterogeneity would be made much more convincing if they sort and then electrophysiologically compare NE vs non-NE cells from the SAME patient (cell line), ideally by using primary patient-derived cells. Similarly, this paradigm

should wherever possible also be applied to other parts of this manuscript.

We agree completely with the reviewer that the analysis of paired NE and non-NE models from the same SCLC tumour would constitute a major improvement for our manuscript. In the revised version, we now evaluate three such pairs: one pair derived from the *Trp53*^{-/-}, *Rb1*^{-/-} “PR” murine SCLC model (AD984LN), one well-established pair derived from a human cell line (NCI-H69), and a newly derived pair of cell lines derived from the same xenograft from a PDX model (MGH1505-1A). Reassuringly, comparison of these isogenic models confirmed that the distinctions between electrically excitable, hyperpolarised NE cells and unexcitable, depolarised non-NE cells are valid, even within the same tumour. Please see **Fig. 1c - 1f** and **Extended Data Fig. 2i**.

Similarly, it would be relevant to learn about differential effects of ChR expression in NE vs non-NE cells, preferably those coming from the same line/patient. Is optogenetically-induced cancer cell excitation in non-NE cells possible, and if (unexpectedly) so, can it transform non-NE cells to NE cells on a transcriptional and functional level (“rescue experiment”)? Otherwise this would be a good control experiment. In this context: Fig. 1h: adequate control experiments (light stimulation without ChR) are missing.

To address these points, we have expressed ChR2 in non-NE cell lines and found that although, as expected, depolarisation could be observed in response to blue light, reflecting the expression of functional ChR2, there were no action potentials induced in these unexcitable cells (**Extended Data Fig. 9b**).

To test this point, we used blue light, at intensities which produce maximal ChR2 activation, to illuminate an excitable NE cell line (AD984LN_fl) which did not express

ChR2, and as expected, no depolarisation, and therefore no excitation of action potentials, could be observed (**Extended Data Fig. 9a**).

Of similar importance, blue light stimulation of ChR expressing cells is also simulating what happens on the postsynaptic side (induction of EPSCs). In general, are these SCLC cells showing any indications of EPSCs, spontaneous or after electrical stimulation (including e.g. of NE-cell only)? Is optogenetic stimulation inducing EPSC(like) currents and not only APs?

We have included cell-attached patch-clamp recording of an isolated NE cell showing large inward (upward) single channel openings, which produce EPSP-like membrane potential transients (**Fig. 1i** and **Extended Data Fig. 1r**).

Last but not least, are there any synapses or synapse-like structures between cancer cells? Between NE cells, or – potentially even most interesting in light of the general concept of this study – between NE cells (presynaptic) and non-NE cells (postsynaptic)?

We have performed extensive paired patch-clamp (**Fig. 1j, 1k**) and rabies virus tracing (**Fig. 1l, 1m**) experiments to address this question. No synaptic interaction could be identified by paired recording, and rabies virus labelled < 1% of TVA⁻ NE cells co-cultured with TVA⁺ NE cells. These data collectively argued against synaptic interaction as the predominant mechanism of wave propagation in NE cells.

Moreover, we would like to clarify that we did not suggest synaptic interaction between NE and non-NE cells and we wouldn't consider that to be a relevant mechanism to the general concept of this current study. We provided evidence that non-NE cells metabolically support NE cells and promote their electrical activity of NE cells (**Fig. 4a – 4f, 4o**), which could be recapitulated by incubation with non-NE conditioned medium (**Fig. 3m, 3n, Extended Data Fig. 8a**) or lactate/pyruvate addition (**Fig. 4g – 4j, Extended Data Fig. 8c, 8d**). *Neither of these conditions requires the physical presence of*

non-NE cells, hence excluding the possibility that synaptic interaction between NE and non-NE cells, if any, plays a role in the metabolic shuttle hypothesis we proposed.

With respect to TTX, one important fundamental experiment is missing: does TTX influence AP generation in SCLC NE cells?

We now show in voltage-clamp experiments that TTX blocks the large fast inward (Na) current expressed in NE cells, and indeed, blocks fast action potentials (**Fig. 1e**). Cav channels are also present in these cells, and contribute a smaller, more slowly-activating nonlinearity of membrane potential responses, which is not affected by TTX (**Extended Data Fig. 2o**).

Another crucial experiment that is missing is the proof that TTX removal is indeed increasing hyperactivity/firing frequency in the SCLC cells, as shown for some nonmalignant neuronal cells.

We thank the reviewer for raising this question, and would like to point out that the extent, nature and mechanisms of homeostatic plasticity are still unclear after decades of research in neurons. We did attempt to perform patch-clamp experiments after TTX washout but did not detect a significant boost in spontaneous firing in the 7 cells recorded within the timeframe of our analysis. This could have been for several reasons: the potential burst firing event reported in neurons might be transient and might have been missed during the timeframe of our recording, or any homeostatic plasticity occurring in these conditions could have been too subtle to be detected by this method. However, considering the various experimental results are consistent with the possibility of homeostatic plasticity, including:

1. The highly specific targeting of Nav by TTX (**Fig. 1e**).
2. The striking upregulation of CREB phosphorylation after TTX washout (**Fig. 6d**).
3. The clear, consistent potentiating effect of TTX pre-treatment both on colony formation *in vitro* and on tumor progression *in vivo* (**Fig. 6a – 6e, Extended Data Fig. 10a – 10c**).

Based upon all the evidence, we suggested an involvement of homeostatic plasticity as a highly likely possibility. We have now added *in vivo* chemogenetic experiments to show that continuous suppression of electrical activity suppressed liver metastasis progression (**Fig. 6f – 6i**), which is again completely in line with our hypothesis.

The concept of particularly high energy demand of AP-firing SCLCs is intriguing. However, I am struggling to understand whether the authors claim that the OXPHOS pathway is particularly relevant here, or the lactate shuttle (“metabolic symbiosis”) mechanism. I am also missing data on differential relevance of glycolysis (another way of ATP production) between NE vs non-NE cells.

We have now added **Fig. 3a and 4a** for better illustration of our hypothesis, and have also extensively edited the manuscript as below, and hope that these points would now be more accessible to a general audience:

[Line: 272]

A major source of cellular ATP production is glucose catabolism⁵², which can generate ATP in two ways: oxygen-independent glycolysis and oxygen-dependent oxidative phosphorylation (OXPHOS). After glucose import, glycolysis takes place in the cytoplasm to catabolize 6-carbon glucose into two molecules of 3-carbon pyruvate. This process can generate two molecules of ATP per molecule of glucose. In anaerobic conditions, pyruvate can be converted into lactate by lactate dehydrogenase (LDH) and excreted, a pathway necessary to maintain redox balance and allow ATP production in the absence of oxygen. Under aerobic conditions, pyruvate can enter the mitochondria to be further catabolized in the tricarboxylic acid (TCA) cycle, which is coupled to the mitochondrial electron transport chain (ETC) to generate a proton gradient/chemiosmotic potential that can be used for ATP production, a process collectively referred to as oxidative phosphorylation (OXPHOS), which can generate up to 36 molecules of ATP per molecule of glucose⁵³ (Fig. 3a). Notably, aerobic glycolysis, or the Warburg effect, is commonly observed in cancer cells⁵⁴, whereby much of the imported glucose molecules in cells is secreted as lactate even under aerobic conditions. It has been proposed that these actively proliferating cancer cells would prioritise anabolic pathways for biomass accumulation, over catabolic pathways for ATP production. We hypothesized that the unique ATP demand in electrically active NE cells may lead to a distinct metabolic phenotype with a higher dependence on ATP-efficient OXPHOS, compared to the majority of other cancer cells, which are non-excitable and have been reported to be less-ATP-dependent⁵⁵ and which instead engage in aerobic glycolysis⁵³.

[Line: 333]

Surprisingly, while NE cells were more dependent on OXPHOS for ATP supply, their glucose uptake rate was significantly lower compared to SCLC non-NE (Fig. 3l) and LUAD/PDAC cells (Extended Data Fig. 7g). This led us to query if other metabolites serve as alternative fuels for ATP production in NE cells. Notably, previous studies in classical SCLC GEMMs demonstrated that either co-culture with non-NE cells⁴ or conditioned medium from non-NE cells⁵ promoted proliferation^{4,5} and metastasis^{4,15} of NE cells. Therefore, we wondered whether metabolite(s) from non-NE cells may underlie such co-operativity.

The authors claim that cholinergic signaling is the key driver of synaptic cell-cell communication between SCLC cells. The data to support this (other than cited previous work) is however very limited and seems to be limited to Fig. 4a – one single recording without statistics. It would be very important to substantially strengthen this part of the manuscript by providing much more robust data on the structural, molecular and electrophysiological level.

We appreciate the reviewer’s comment and have now included more data related to the cholinergic signaling in SCLC, providing electrophysiological characterization of AChR in SCLC (Fig. 2b), as well as survival analysis of AChR expression in SCLC patients (Fig. 6n).

In addition, we have also significantly reconstructed the manuscript to avoid potential confusion. We would like to clarify that, while we proposed cholinergic signaling to be one of the mechanisms inducing the electrical activity of SCLC NE cells, we have never claimed that there is synaptic cell-cell communication between SCLC cells, nor did we propose that cholinergic signaling was driving such interaction. In fact, our data suggested that *diffusile factors* are the key in cell-cell communication between SCLC cells,

which is typical of neuroendocrine cells and does not require cell-cell contact, let alone synapse formation (Fig. 1n, 1o).

The data provided In Fig. 4 (and else) is not really convincingly supporting one key hypothesis (if not conclusion) of the study: that a cancer-autonomous electrical self-stimulation is making SCLC independent from direct neuronal input during cancer progression. In fact, everything cited in favor for this concept is no direct proof, rather circumstantial. One way to address this interesting hypothesis would be to investigate different metastatic sites: are liver metastasis (where the main tumor biology – findings are made) are particularly hypo-/non-innervated, which makes NE cells particularly liver metastasis-proficient – while other metastatic (or primary) sites are much better innervated, which makes the electrical activity of NE cells less important? This should really be addressed in mouse studies covering multiple sites of tumor growth, characterizing innervation patterns and at the same time recording tumor growth kinetics and finally overall animal survival.

We completely agree that it would be particularly interesting to assess how the innervation pattern of different organs is affecting how cancer cells metastasise and colonize different sites, so that it is worthy of a full new project to address this question properly. We have expanded the analysis of lung innervation in more detail as shown in Fig. 2; however, we believe that properly examining the impact of innervation in metastatic sites would be an entirely new project that would be hard to work into this already data dense study. Further, because our focus is on *cancer cell intrinsic electrical activity* in SCLC, and external ACh supply from the cholinergic innervation is *one of the mechanisms* to activate the electrical activity of SCLC, but not our key hypothesis.

Minor points:

Abstract: ...similar to astrocytes...: correct, but this applies to any other non-neuronal cell; the authors might want to stress this point that also implies an even wider relevance of their finding.

We thank the reviewer’s suggestion and have now modified the text accordingly.

Ext Data Fig 1h, but also Fig. 1l and EDF 3e etc.: the quantification of “metastasis” is not convincing; only liver size and weight seems be determined. EDF1h: What is the color code of the graph?

We thank the reviewer for their suggestion and have replaced EDF1h with the new **Fig. 5a**. Liver weight is a routine measurement for metastatic burden in the liver, especially when the metastasis is extensive¹⁻³; on the other hand, to address the reviewer's concern, we have now normalized the liver weight to body weight, and further included measurements of *tumor area, proliferation and apoptosis markers* in **Fig. 6a – 6c**, and **Extended Data Fig. 10c**.

Referee #2 (Remarks to the Author):

Small cell lung cancer is a highly aggressive form of lung cancer, characterised by a neuroendocrine phenotype. The precise molecular mechanisms that underlie its aggressive nature are largely unknown and elucidating such mechanism pertinent for designing improved therapeutic interventions for patients. Given that SCLC exhibits neuronal characteristics, Peinado and colleagues, explored whether, like neurons NE cells present within SCLC tumors, exhibit similar features and activities, with a focus on whether they can fire action potentials. The authors explore this question using the well-described NE and non-NE cellular populations present in SCLC tumors and demonstrate that indeed NE cells can fire action potentials and this activity requires a heightened ATP demand. Together, these processes promote tumor progression.

Major points:

1. The authors utilise NE and non-NE cell lines derived from PRp130 SCLC lung tumors (maintained on a mixed genetic background). B6129SF1/J mice are utilised for intrasplenic injection studies, which demonstrate that NE, but not no-NE tumor cells seed metastatic lesions in the livers of recipient mice (Extended data Fig. 1h). The authors attribute this in part to differences in the electrical activity of NE vs non-NE cells. However, given that non-NE tumor cells have been previously shown to express high levels of MHC-I, and fail to engraft tumors when transplanted into immune-competent recipient mice (PMID: 33707236). The difference therefore seen in metastatic potential may not be solely attributed to differences in the electrical activity of the NE vs non-NE cells. To more directly address this: is a similar difference in metastatic potential seen following transplantation of NE and non-NE cells into immune-deficient recipient mice?

The reviewer raised an interesting and important point. We have performed the experiment in NOD-SCID mice as suggested, and observed the same result (**Fig. 5a**):

We have also modified the main text to highlight the fact that in the original paper⁴, which described that only NE cells, but not non-NE cells, are capable to establish liver metastasis from subcutaneous tumors, was performed in Balb/c Nude immunosuppressed mice⁴. This was later confirmed by a follow-up paper, in which intravenous injections were performed in Balb/c nude immunosuppressed mice as well⁵:

[Line: 423]

Previous studies have shown that NE transformation of adenocarcinoma promotes metastasis and treatment resistance²⁵ and adenocarcinoma cells adopt a NE-like transcriptomic signature when they become more metastatic³. Interestingly, despite having the same oncogenic driver mutations, SCLC non-NE cells fail to establish liver metastases when transplanted subcutaneously⁴, in sharp contrast to SCLC NE cells. Since the liver is one of the most frequent distant metastatic sites in SCLC patients, we recapitulated the final steps of liver metastasis using intrasplenic transplantation⁵⁶. Indeed, we corroborated that when NE and non-NE cells were intrasplenically injected into immunocompromised NOD-SCID mice, only NE cells could efficiently establish liver metastasis (Fig. 5a).

2. Related to the above, the authors show in Fig. 1l, that inhibiting the action potential firing in SCLC NE cells using tetrodotoxin (TTX) leads to a suppression in the tumorigenic capacity of SCLC NE cells, what time-point was examined, as the liver burden seen following injection of A3062C cells in Extended Data Fig. 1h, was significantly higher than the control level seen in Fig. 1l. Has this experiment been repeated with an additional independent NE cell line e.g., AF1165?

We appreciate the important suggestion and have now included another NE line AD984LN_fl, and also repeated the same experiment using the original AF3062C, as shown in **Fig. 6a**.

We have also added in the Method session:

For the transplantation of SCLC NE cell after TTX pretreatment, cells were treated with 1 μ M TTX for 24 hours prior to the injection. Liver was sampled 25 days (AF3062C) or 34 days (AD984LN_fl) after injection.

2. “non-NE” human SCLC cell lines were used to determine whether the differences in electrical activity were conserved in human tumor cells. Several of the human SCLC cell lines utilised for this analysis (original classification as YAP1-expressing SCLC subtype by PMID: 30926931), such as H841, SB-5, have recently been re-classified as SMARCA4-UT tumors (BioRxiv;<https://www.biorxiv.org/content/10.1101/2022.10.09.511505v1>) or were misclassified, and thus should not be used as a model of SCLC.

We thank the reviewer for pointing out the changing landscape in SCLC research and have replaced these data with other cell lines that should better reflect genuine “non-NE” hSCLC cells (**Fig. 1a**). This suggestion led directly to an in-depth analysis of paired NE and non-NE cell lines from the same SCLC tumours. In addition to comparisons in murine SCLC models, we investigated a well-established pair of NE and non-NE cell lines derived from the human SCLC cell line NCI-H69, as well as a novel pair of NE and non-NE cell lines derived from the same xenograft of the PDX MGH1505-1A. Please see new data in **Fig. 1b - 1d**, and **Extended Data Fig. 2**.

3. Expression of MPC1 and MPC2 is higher in NE cells, compared to their non-NE counterparts (Fig. 2g), however, this expression was variable with some NE lines, exhibiting low (MPC2) to barely detectable levels. Can the authors speculate on the heterogeneity observed and how that may impact/relates to the metabolic capacity of these tumor cell lines?

We have decided to replace the MPC2 part in the revised manuscript with more functional analysis to demonstrate the differential dependence on OXPHOS in NE and non-NE cell lines. Please see the updated version of **Fig. 3**.

4. Some of the findings presented in this manuscript are leveraged from a prior CRISPR screen performed by several of the co-authors. However, in several places, additional context/details are warranted to support some of the findings presented. For example, and please mind my ignorance, but it is unclear how the genes, shown in Extended Data Fig. 6 are less sensitive to KO of glycolysis pathway genes.

We thank for reviewer's suggestion and have now added more details to help interpret the data resented:

[Main text, Line: 297]

Coincidentally, we have previously performed a CRISPR-mediated genetic screen to look for preferential vulnerabilities in SCLC NE cells compared to LUAD and PDAC⁵⁶. Vulnerability is defined as genes essential for cell survival, so that cells harboring knockouts of these genes will become disadvantaged for survival and selected against, or “dropped-out” from the population over time⁵⁷. We further defined preferential vulnerability for SCLC NE cells as genes heavily selected against in SCLC NE cells but which were non-essential for LUAD and PDAC⁵⁶. In that screen, three out of the top eight candidate genes we identified as having SCLC NE-preferential vulnerability encoded ETC complex proteins: Ndufa1 (complex I; ubiquinone oxidoreductase), Sdha and Sdhb (complex II; succinate dehydrogenase, SDH), rendering ETC the most prominent hallmark of SCLC NE cell vulnerability in the gene ontology analysis (Fig. 3d, Extended Data Fig. 7b). On the other hand, consistent with the hypothesis that NE cells depend more on OXPHOS and less on glycolysis, the SCLC NE cells were less sensitive to knockout of glycolysis pathway genes compared to LUAD and PDAC (Extended Data Fig. 7c).

[Legend for Extended Data Fig. 7c]

(c) CRISPR-screen results of genes involved in the glycolysis pathway in SCLC NE cell lines, LUAD and PDAC cell lines^{4,5}. Gene scores (\log_2 fold change, L2FC) for the indicated genes for SCLC, LUAD and PDAC (Mean \pm SEM; each dot represents a cell line). A negative L2FC score suggests the gene to be essential since knocking out this gene is deleterious for cell survival and hence selected against in the entire population; a L2FC score close to zero suggests that the gene is neutral for cell survival; a positive L2FC score suggests the gene to be a tumor-suppressor, since knocking out this gene promotes cell survival/proliferation. In all these glycolysis-related genes, their gene scores were more negative in LUAD and PDAC cell lines compared to those in the SCLC NE cell lines, suggesting that these genes are more “essential” for LUAD/PDAC survival compared to their role in SCLC NE cells.

As the reviewer has pointed out, it was unexpected that SCLC NE cells are less sensitive to KO of glycolysis pathway genes but highly sensitive to OXPHOS-related gene KO compared to LUAD and PDAC, and we reason that it was because SCLC NE cells are less reliant on glycolysis than on OXPHOS, which is in line with our findings.

5. In several of the studies, non-excitable, PDAC and LUAD cell lines served as “control” samples. while non-NE cells, which the authors also show are non-excitable are not used/shown and would serve as far superior controls in several of the experiments, and thus for consistency it would be favourable to include, where possible a side-by-side comparison of non-NE cell lines.

We have performed additional experiments using paired NE/non-NE cell lines and have modified the figures accordingly. We found that all “non-excitable” cell lines we examined, including SCLC non-NE cells and LUAD/PDAC cells, show similar metabolic phenotypes: glycolysis index (**Fig. 3b, 3c**); sensitivity to rotenone with (**Extended Data Fig. 7d**) and without (**Fig. 3e, 3f**) pyruvate (**Fig. 3g**); and oxygen consumption rate (**Fig. 3i – 3k**).

6. Quantification of the findings shown in Fig. 4f-k and Extended data Fig. 9 would increase the robustness of the findings. Particularly the expression of MCT4 which looks quite abundant in Fig. 4k, but how reproducible that that pattern of expression across a series of progressed lesions?

We completely agree with the reviewer and have now expanded the original Fig. 4f - 4h and Extended Data Fig. 9 into **Fig. 2d - 2l** and **Extended Data Fig. 5a - 5j**, with extensive characterization and quantification throughout SCLC initiation and progression:

We also expanded and quantified the original Fig. 4i, which is now **Fig. 6 k - 6m**:

The original Fig. 4j, 4k became the current **Fig. 4m, 4n**:

Minor points:

1. Light field images are provided of NE cells exposed to varying light pulse frequencies and duration (extended Fig. 2f-g) from which it is concluded that blue light simulation alone resulted in "...duration dependent growth reduction or even cell death". Quantitative evidence should be provided to support this apparent growth reduction and/or cell death, e.g., FACS analysis.

We thank the reviewer's suggestion and have now included the SRB quantification in **Extended Data Fig. 9e**.

2. Images in Extended Data Fig. 3b do not capture the bottom the well and could be improved.

We have performed additional experiments for **Extended Data Fig. 9** (the original Extended Data Figure 3), including more WB and ChR2-expressing non-NE cells, and decided to remove these pictures along with those from the LUAD and PDAC cell lines.

3. References missing, NE cells as cells of origin of SCLC – PMID: 21665149 and PMID: 21822053.

We thank the reviewer's suggestion and have now updated the references:

[Line: 52]

SCLC is highly heterogeneous, both intra- and intertumorally. While rare pulmonary NE cells (PNECs) are thought to be prominent cells of origin for SCLC tumors^{8,10,11}, cancer cells with varying levels of NE gene expression have been identified.

8. Sutherland, K. D. et al. Cell of origin of small cell lung cancer: inactivation of Trp53 and Rb1 in distinct cell types of adult mouse lung. Cancer Cell 19, 754-764, doi:10.1016/j.ccr.2011.04.019 (2011).

10. Ouadah, Y. et al. Rare Pulmonary Neuroendocrine Cells Are Stem Cells Regulated by Rb, p53, and Notch. Cell 179, 403-416 e423, doi:10.1016/j.cell.2019.09.010 (2019).

11. Park, K. S. et al. Characterization of the cell of origin for small cell lung cancer. Cell Cycle 10, 2806-2815, doi:10.4161/cc.10.16.17012 (2011).

Referee #3 (Remarks to the Author):

This paper identifies differences in intrinsic electrical activity between small cell lung cancer (SCLC) cells containing and lacking neuroendocrine (NE) features. The authors provide evidence that excitability is required for some aspects of tumorigenicity of NE cancer cells, including growth after implantation into the liver. They also demonstrate metabolic differences between NE and non-NE cells, particularly a reduced rate of lactate secretion and increased sensitivity to oxidative phosphorylation (OXPHOS) inhibitors in NE cells. The authors postulate that an increased dependence on OXPHOS helps NE cells meet the metabolic demands of generating action potentials. They show that unlike non-NE cells, NE cells can undergo net lactate uptake, and they propose a model akin to the astrocyte-neuron lactate shuttle (ANLS), in which glycolytic non-NE cells secrete lactate, which NE cells take up and oxidize to meet their energetic demands. The paper proposes an interesting mode of cooperativity between NE and non-NE cells within the same tumors. However, key aspects of the model are not supported by the data, and in other areas the descriptions of the experiments are inadequate. Specific comments:

1. It is unclear how the authors obtained the GEMM-derived NE and non-NE cell lines. Were these from independent clones from the same isogenic GEMMs?

We have updated the information in **Fig. 1a** and explained the method of NE/non-NE derivation in the main text:

[Line: 113]

Previous studies revealed functional ITH in classical SCLC GEMMs^{4,5}, in which the crosstalk of NE and non-NE cells (both segregated from tumours according to their culture phenotype: either forming floating aggregates –fl–, or being adherent –adh–, respectively), could influence the metastatic potential of these tumours⁴. To further explore the underlying mechanism, we derived a panel of cell lines from GEMMs of

classical SCLC, including two independent pairs of NE and non-NE cells originally derived from the same tumor, and a pair of NE and non-NE lines from the same patient-derived xenograft (PDX) model (Fig. 1a, Extended Data Fig. 1).

Detailed description of the source of the cell lines could be found in the Methods:

[Methods]

Mouse SCLC lines: The following cell lines were derived from GEMMs as previously described¹. In particular, AF1165 was derived from a primary tumor of a PR; Rosa26^{LSL-Tom/+} mouse. AF3062C was derived from liver metastases of a male PR; Rosa26^{LSL-Tom/+} mouse. AF1281m1 was derived from a relapse tumour after chemotherapy in a PR; Rosa26^{LSL-Tom/+} mouse. TP2031T2 was derived from a primary PR tumor. AD984LN_fl/AD984LN_adh were derived from the same lymph node metastasis in a PR mouse⁴. AF3291LN_fl/AF3291LN_adh were also derived from the same lymph node metastasis in a PRPTEN mouse.

The PRM2.1a cell line was derived in-house from a primary tumour in a Trp53^{fl/fl}; Rb1^{fl/fl}; Myc^{LSL-T58A/+} (PRM) mouse, as previously described⁵. Briefly, the primary tumour was dissected, cut into small pieces, and incubated for 30 minutes at 37°C in 6 ml digestion solution [10% TrypLE (Gibco, 12605010), 1 mg/ml Collagenase IV (Gibco, 17104019), 1 mg/ml Dispase II (Sigma-Aldrich, D4693) in HBSS]. The digestion reaction was quenched by adding 4 ml of ice-cold quenching medium [10% FBS (Gibco, 11320033), 18.75 µg/ml DNase I (Sigma-Aldrich, DN25) in DMEM]. The cell suspension was passed several times through a 18G needle before being filtered with a 100 µm strainer. Cells were centrifuged at 800 g for 5 minutes, resuspended in 1X RBC Lysis Buffer (BioLegend, 420301) and incubated for 3 minutes at 37°C. After washing with ice-cold PBS, the cells were plated in complete culture medium on tissue-treated culture vessels, selecting for cells growing as floating aggregates.

2. Independent human SCLC cell lines with high or low NE scores are also assessed, and the NE score is correlated with electrical activity. But these experiments are susceptible to idiosyncrasies unrelated to the NE state. A better experiment with more relevance to the idea that ITH supports the growth of NE cells would be to select NE and non-NE cells from the same parental population, then compare electrical activity between the two sub-populations. In several human SCLC cell lines, straightforward methods have been described to enrich the NE and non-NE populations (see Pongor LS et al., Cancer

Discovery 2023, PMID 36715552; Cai L et al., Communications Biology 2021, PMID 33750914).

We thank the reviewer's comment and have now included analysis for paired hSCLC cell lines and PDX models originally derived from the same tumor, as shown in **Fig. 1c, 1d, 1f, Extended Data Fig. 2i**:

In addition, we would also like to highlight the status quo of the SCLC field: unfortunately, the enrichment method the reviewer has referred to in those publications were based on *pure bioinformatic analysis*; while paired NE and non-NE mSCLC cell lines have been reported, so far it remains challenging in the field to establish paired NE and non-NE hSCLC cell lines. One of our co-authors, Dr. Benjamin Drapkin, has generated a large panel of PDX models from SCLC patients. From one of these models, MGH1505-1A, his group has generated paired NE and non-NE cell lines from the same xenograft. The derivation of these cell lines is reported here for the first time, and their electrophysiologic analysis in Figure 1 and Extended Figure 2 strongly supports our initial conclusions. Dr. Drapkin works closely with the SCLC team at UT Southwestern, including Dr. John Minna, who along with Dr. Adi Gazdar established the vast majority of hSCLC cell lines that are used to study this disease. Their laboratory graciously supplied us with a second pair of NE and non-NE hSCLC cell lines, derived from the parental cell line NCI-H69. Unfortunately, to the best of our knowledge, these are the only cell line pairs available to address this question. Overall, the challenges of generating paired cell lines from human SCLC tumours have made it difficult to experimentally validate the ITH using human cell lines or PDX models. Such data remains lacking in the field, even in the previous two landmark papers describing functional ITH in SCLC. We have revised the main text to highlight this limitation for a general audience.

3. For perspective/scale, can the authors show a typical current-clamp recording of neurons in Fig. 1?

We have now included the graph in **Fig. 1b**.

4. SCLC NE cells express high levels of ATP synthetase subunits, but expressing ATP synthetase alone (i.e. without components of the ETC) would have little effect on ATP production. Did this proteomics experiment also reveal increased expression of other components of the ETC in NE vs non-NE cells? This is worth knowing, because in some contexts ATP synthetase has functions distinct from its ability to produce ATP.

We thank the reviewer for the interesting comment. Indeed, in our list of differentially expressed proteins between SCLC NE and other cancer cells, we did not observe a major difference in other ETC proteins; nor did we see a difference at RNA level in the vulnerable genes we found in the CRISPR screen. Although in human SCLC data, many ETC proteins are indeed expressed at higher levels in NE cells compared to non-NE cells. However, *we have removed the protein mass-spec data related to ATP synthetase to better focus on the genetic vulnerability of ETC components in NE cells*, which indeed may or may not correlate with their RNA expression.

5. In the experiments describing ATP synthetase and OXPHOS dependence, the authors seem to imply that NE cells have higher respiration/higher OXPHOS activity than non-NE cells, but this is never directly shown. It is conceivable that NE cells have enhanced dependence on OXPHOS without having higher OXPHOS rates. A direct comparison of OCR between NE and non-NE cells would close the loop on these experiments.

As the reviewer suggested, we have included the Seahorse experiments in **Fig. 3i**, and confirmed that NE cells have both higher dependence on OXPHOS and higher OXPHOS rates (**Fig. 3k**).

6. If true, it is important that OXPHOS dependence is unrelated to NAD⁺ recycling. But these experiments are superficial. What happens to the NAD⁺/NADH ratio in rotenone-treated NE cells with and without pyruvate? The argument would be more compelling if the ratio changed when sensitivity to rotenone did not. Another way to exclude the role of NAD⁺ recycling would be to test the effect of LbNOX on rotenone-treated NE cells. This bacterial enzyme converts NADH to NAD⁺ independently of ETC complex I.

We have now validated that LbNOX expression (**Extended Data Fig. 7e**) could barely rescue the unusual sensitivity of NE cells to rotenone, as shown in **Fig. 3h**. This argues this is a dependence on OXPHOS (i.e. ATP production coupled to mitochondrial respiration).

7. Key experiments related to the proposed ANLS-like mechanism are missing. First, is adding lactate to dialyzed serum sufficient to drive proliferation of NE cells, as in Fig. 3e? And how does adding pyruvate compare to adding lactate in these experiments? Second, if NE cells do not express MCT4, do they express MCT1 or some other monocarboxylate transporter to allow them to take up lactate? Third and related, the role of the lactate shuttle in NE cell excitability and growth is not tested. The key experiment would be to inactivate lactate import and test whether this suppresses some of the features of NE cells described in the paper (electrical activity; ATP levels; the ability of electrical activity to stimulate growth; etc).

We thank the reviewer for these interesting questions.

We have now added the WB for MCT1 expression in paired SCLC NE and non-NE cells (**Extended Data Fig. 7j**).

We have also supplied lactate to dialysed serum, but didn't observe consistently increased proliferation in the SRB assay. In addition, when we boiled the conditioned medium, we found that the proliferation effect was completely abolished. Therefore, we reason that for the pro-proliferation effect, both the protein and metabolite components are necessary and neither alone was sufficient.

On the other hand, lactate addition did sustain ATP levels in culture in serum-free medium as measured by CTG assays:

In addition, we validated the role of lactate in NE cell excitability using direct electrophysiology measurement (Fig. 4, Extended Data Fig. 8), showing that both lactate and pyruvate maintain more hyperpolarized resting membrane potential of NE cells, and the ATP-demanding electrophysiological status (Fig. 4g, 4h, Extended Data Fig. 8c, 8d).

Importantly, the effect of lactate could be abolished by co-incubation with an MCT1 inhibitor (**Fig. 4i, 4j**).

While lactate clearly sustains the ATP level in starved NE cells and maintains/promotes the electrical activity of NE cells, we think that the direct effect of lactate addition on proliferation might also be confounded by these two agonistic forces: pro-survival signals from increased electrical activity counterbalanced by decreased availability for anabolic reactions for biomass production, as a result of increased ATP demand in more electrically active cells and hence suppressed catabolic reactions.

Now we have added new data using MCT4 inhibitors in *ex vivo* lung slices with autochthonous SCLC tumors, which clearly demonstrated the MCT4 inhibitor diclofenac suppressed the electrical activity of autochthonous SCLC tumors *ex vivo* (**Fig. 4o**):

Minor:

1. Line 261 says that NE cells have an increased “fraction of glucose utilized for OXPHOS,” but this was not demonstrated. The NE cells secrete a lower fraction of glucose as lactate, but glucose has more metabolic fates than lactate and the TCA cycle.

We appreciate the reviewer’s comment and have removed the statement.

2. The images in Fig. 3d and ED Fig 7f are difficult to interpret. The authors need a better way of arguing that SOX1 and HES1 are expressed in different cells than MCT4.

We apologise for the potential confusion – We would like to show that SOX1 is expressed in NE cells, while *both HES1 and MCT4 are expressed in non-NE cells*. Therefore, in the original Fig. 3d and ED Fig. 7f, the current **Fig. 4n** and **Extended Data Fig. 8f**, we intend to demonstrate that *SOX1 (expressed by NE cells) and MCT4 (expressed by non-NE cells), do not co-localized* in autochthonous SCLC tumors. We have modified the text for better clarity:

[Line: 358]

In the central nervous system, monocarboxylate transporter 4 (MCT4, encoded by Slc16a3), a major protein involved in lactate export, is exclusively expressed in astrocytes and not in neurons⁶², and the astrocyte-to-neuron lactate shuttle (ANLS) has been proposed to play a major role in neuronal energy supply⁶, especially when elevated activity is required³². In analogy to the ANLS hypothesis, the HES1-expressing⁵, astrocyte-like non-NE cells had higher MCT4 compared to the neuron-like NE cell lines, which expressed another monocarboxylate transporter, MCT1, a protein proposed to be important for lactate import (Fig. 3p, Extended Data Fig. 7j).

We have also created a new illustration in **Fig. 4a** to demonstrate the relationship:

3. In Fig. 3g, please confirm in the legend that the lactate-supplemented medium also contains glucose.

We have clarified that in the legend of the current **Fig. 3r**:

[Line: 1003]

(q-s) Changes in lactate (q, r) and glucose (s) concentration in media (25 mM glucose, no pyruvate, with 2% dialysed FBS) after overnight incubation of paired NE and non-NE SCLC cell lines in 20 mM lactate (q), and of SCLC NE cells with different concentrations of lactate (NE #1: AF1165; NE #2: AD984LN_fl; NE #3: AF3291LN_fl)(r, s).

Referee #4 (Remarks to the Author):

The manuscript by Peinado et al. addresses the relation between cellular heterogeneity and electrical activity in small-cell lung cancer (SCLC), and uncovers novel links to metabolic symbiosis between SCLC cells of neuroendocrine phenotype (NE) and non-neuroendocrine cells (non-NE). The study reports novel data that help clarify several open questions in the field of SCLC and particularly with respect to the role of electrical activity in driving tumor growth and progression, which they now report to be specific to NE cells. NE cells were reported to acquire a neuronal phenotype, while non-NE cells were reported to mimic astrocyte-like cells of the nervous system. Employing a number of novel techniques, including single cell sequencing-based cell classification, metabolic assays, electrophysiology and modelling of cell activity data, the study reports that NE and non-NE cells play non-redundant roles in tumor progression, whereby lactate derived from non-NE cells supports the energy metabolic demands of NE cells, which are high because electrical activity of NE cells requires large ATP consumption to maintain membrane potential. Tumor heterogeneity developing over the course of tumor growth was suggested to establish a self-sustaining system of symbiosis between NE and non-NE cells. A large majority of analyses are performed on cultured tumor cell lines and tumor cells classified via gene expression analyses.

The study uncovers several highly interesting principles and puts forward novel concepts. There is the promise of major conceptual advance, particularly if the data can be substantiated in a more in vivo setting. The study has the potential to make a major impact, not only on the field of SCLC, but also other types of cancers containing cells of NE origin. The following major and minor points can be considered.

Major points:

1. As such, it is known since several decades that both cell lines and tumors of the lung can show Ca²⁺ transients and electrogenesis. The authors have done a highly thorough and excellent job in electrophysiological analyses to uncover that NE type cells in SCLC (and other types of tumors, such as pancreatic neuroendocrine cells) but not non-NE cells possess the ability to generate electrical activity and calcium transients, leading to neuronal plasticity-like processes, indicating that this is a property that is specific to certain cells in a heterogenous environment. However, all of the work is based upon current injections in patch clamp analyses, which simulates an artificial situation where cells are driven to spike. Also, although spontaneous activity was confirmed as a phenomenon that is known previously, a major open question in the field remains unanswered as to what leads to natural electrical activity in cancer cells. Nowhere in the manuscript does it become why NE cells spike naturally, nor is activity analysed in the natural setting of a tumor. Which processes lead to natural depolarization of cancer cells? Which channels mediate electrical activity and why? A number of previous studies have reported expression of sodium, calcium and potassium channels in tumor cells, including

SCLC, while others have contradicted a role for electrical signals driven by channel activity in tumor progression. While the beauty of this study is its allocation of electrical activity and the downstream sequelae to specific cell types within SCLC tumors, it needs to work out the mechanisms as to how such activity can come about in the first place in a real tumor setting (i.e. independent of the cell culture media employed etc.).

We value the reviewer's suggestion and have performed *ex vivo* imaging on fresh lung slices from autochthonous SCLC models in **Fig. 2o and 4o**. We confirmed that SCLC tumors showed spontaneous calcium transients *ex vivo* (**Fig. 2o, Extended Data Movie 8**), which could be dampened when we blocked the lactate shuttle using an MCT4 inhibitor diclofenac (**Fig. 4o**). We have identified that both ACh and ATP, both are highly abundant in the tumor microenvironment, could initiate calcium activity in SCLC NE cells (**Fig. 2b, Extended Data Fig. 5m, Extended Data Movie 4**).

2. Another second concern is that SCLC cells are very broadly and somewhat crudely classified as NE and non-NE. All of the functional experiments imply that the non-NE population is a homogenous population that does not possess electrical activity, but is there to promote the growth of NE cells by providing lactate as a fuel. However, it is well established via single cell sequencing analyses both in cell lines and in *in vivo* tumors that the non-NE population is actually very heterogenous. Based on gene signatures, such as ASCL1, NEUROD1, and POU class 2 homeobox 3 (POU2F3), several subclasses are known. Heterogeneity is also given by origin (PNEC, tuft cells, chemosensory epithelial cells etc.). A more mechanistic and refined analysis would be timely, rather than a general NE vs. non-NE classification.

We thank the reviewer for having raised this crucial yet complicated aspect in SCLC biology. We have now added the clarification to this revised manuscript and included **Fig. 1a** to better present the models we used in this study.

We have based our experiments upon GEMM and GEMM-derived SCLC cells, which is less heterogeneous compared to the patient setting, and the NE/non-NE classification is routinely used in SCLC GEMM studies, as in Lim et al (2017)⁶. We have revised the main text according to the reviewer's suggestions:

[Line: 64]

Regardless of the molecular subtypes, tumors from SCLC patients demonstrate almost universal inactivating mutation of both tumor suppressor genes TP53 and RB1. GEMMs of SCLC were developed based on these mutations, including the Trp53^{-/-}, Rb1^{-/-} "PR" model, the Trp53^{-/-}, Rb1^{-/-}, Rbl2^{-/-} "PRP130" model and the Trp53^{-/-}, Rb1^{-/-}, Pten^{-/-} "PRPTEN" model. Tumors derived from these GEMMs express high ASCL1 and represent the classic SCLC. Later, a c-Myc overexpression model was developed (Trp53^{-/-}, Rb^{-/-}, Myc^{T58A} "PRM" model); tumors from the PRM model express high NEUROD1 and are typical of the variant SCLC¹³ (Fig. 1a). Studies in classical GEMMs and GEMM-derived cell lines revealed that both mouse SCLC (mSCLC) NE (correspond to the SCLC-A subtype in hSCLC) and non-NE (correspond to the SCLC-Y subtype¹⁴) subpopulations could arise from the same tumor and even the same clone⁴, and provided the first experimental evidence of functional intratumoral heterogeneity (ITH) in SCLC.

[Line: 467]

To further assess if the association could potentially be generalized, we performed the colony formation assay in an expanded cell line panel with distinct excitability, from different SCLC subtypes and even from different cancer types, including a variant mSCLC cell line (PRM model), a mouse pancreatic neuroendocrine tumor (mPanNET) cell line (β TC-3)⁷⁵, hSCLC-A cell lines (COR-L47, NCI-H889), hSCLC-N cell lines (NCI-H82, COR-L279), and non-excitable mLUAD and mPDAC lines. Indeed, TTX suppressed the long-term tumorigenic potential in all electrically active cells tested, regardless of their cell and tissues of origin; in stark contrast, non-excitable cells were unaffected by TTX treatment (Fig. 5l, 5m).

3. The optogenetic experiments claiming a functional role for electrical activity in driving tumor progression in colony forming in vitro assays and organ cultures addressing hepatic metastases are not conclusive. A major caveat, the authors report, is given by toxicity induced by blue light LEDs and the authors rely on complicated explanations and indirect observations to account for paradoxical observations. However, this should be easy to circumvent. In this era of multicolor optogenetics, it is easily feasible to use red-shifted variants of channelrhodopsin or even other substitutes for excitatory manipulations.

In the same vein, the authors find results contrary to their expectations with the use of TTX to silence neuronal activity and attribute it to homeostatic plasticity. To substantiate the latter, the authors will need to provide a series of additional evidence in the vein of whether all of the molecular elements mediating classical homeostatic plasticity are actually present in SCLC NE cells. Moreover, elsewhere in the manuscript, TTX treatment does not evoke homeostatic plasticity, but gives expected results. Again, it is advisable to use additional tools to circumvent the technical issues that currently occlude unequivocal interpretations, e.g. any of the myriad of optogenetic or chemogenetic activity silencing

tools. This is a critical point that needs to be adequately addressed to support functional claims.

We have added illustrations here for better clarification of the experimental settings. In experiments where we reason that homeostatic plasticity played a role, we performed short-term, overnight TTX treatment, and TTX was “washed out” before seeded for the colony formation assays (**Fig. 6e**).

REDACTED

On the other hand, in experiments where we didn't think homeostatic plasticity played a role, TTX treatment was continuously given during the entire course of the colony formation assay, and was never removed from the culture medium (**Fig. 5j, 5k**).

REDACTED

Finally, as the reviewer suggested, we have now included data using chemogenetics with inhibitory PSAM4 (iPSAM4), a ligand-activated ion channel (**Fig. 6g - 6i**). Indeed, our new data is in line with our previous observations using ChR2 and TTX, and demonstrated that suppressing the electrical activity of SCLC decreased metastasis formation and prolonged survival *in vivo*.

4. Along the lines of point 1 and point 4 above, while the authors attempt to make a case for neuron-like activity in NE cells in driving tumor growth, there are no mechanistic insights given in to how this takes place. CREB phosphorylation is reported, but how this or other pathways lead to cancer progression is not clarified. This is a critical deficit, also in the light of several previous studies that have already speculated on the ability of electrical activity to promote tumor growth. Key mechanistic analyses are needed to extend beyond previous studies and enable novel advance.

We thank the reviewer for the comment; indeed, several studies have been demonstrating that certain neuronal receptors and ion channels may impact cancer cell growth *in vitro*; less studies, including some from our own research, have demonstrated their roles *in vivo*. However, as we discussed in the manuscript, the vast majority of these studies were performed in cancer cells that could not fire action potentials; despite observed ion channel expressions or membrane potential fluctuations, the archetypical neuronal activity, namely action potential firing and propagation, was lacking in these studies. To the best of our knowledge, direct manipulation of the genuine electrical activity in excitable cancer cells, leading to changes in tumor progression in vivo, has not been properly demonstrated. We believe that this finding per se is a crucial advance not only in SCLC biology, but also for the first time moving beyond specific ion channels and minor membrane potential fluctuations that do not recapitulate the full spectrum of neuronal activity, directly interrogating the impact of neuron-like, cancer cell intrinsic electrical activity on cancer progression.

5. The remaining part of the study then focusses on major energy demands inflicted on NE cells by electrical activity. Here, the authors have performed a series of excellent experiments to propose models and address the role of Oxphos and glucose-pyruvate-lactate metabolism and shuttling of lactate from non-NE to NE cells. The data are exciting and the model they propose is indeed intriguing. Here, two questions arise: (i) the authors refer to these interactions as ‘metabolic symbiosis’, which would imply that both types of cells benefit from each other. While the study reports on how non-NE cells may be providing lactate to vulnerable, energy-depleted NE cells, like astrocytes provide for neurons, it does not become evident what non-NE cells derive from NE cells. Why is this a mutually beneficial arrangement and how does it promote intratumor heterogeneity over time?

We thank the reviewer’s comment and would like to point out that the term “metabolic symbiosis” is commonly (mis)used in the (cancer) metabolism field even when only one cell type is providing metabolites to the other and only one cell type benefits. We acknowledge that this term may be confusing for the general audience, and we thank the reviewer for having pointed this out. We have now modified the manuscript to only use “metabolic support” and “metabolite shuttle” instead.

Furthermore, we would also like to clarify that we did not claim that this co-operativity does not promote intratumoral heterogeneity; instead, we consider such interaction as a result of intratumoral heterogeneity, which was incurred by NOTCH signaling as reported in Lim et al.⁶, instead of triggering or promoting intratumoral heterogeneity.

6. At the end of the manuscript, nicotinic cholinergic signaling is suddenly brought into

picture, but the morphological analyses and in vitro experiments are sketchy at best. Potentially unrelated observations are strung together, and no evidence given in in vivo setting although the authors describe the availability of appropriate animal models. Based on presence of cholinergic innervation alone, major deductions are made. The reviewer would advise to either work this out in more detail or to drop this part. It is correlative at best, and does not give any functional or mechanistic insights.

We appreciate the reviewer's suggestion and have now expanded this part into a full **Fig. 2**. The potential clinical relevance of cholinergic signaling has been assessed in **Fig. 6**. We believe that these new data would also help to answer the first question regarding mechanisms of calcium wave initiation in the natural setting.

7. A major concern is that most of the data are restricted to cell lines and in vitro experiments. Can key findings be substantiated in a more natural, in vivo setting? For example, human SCLC types vary between NE-high and NE-low. Can the authors use biopsies from heterogenous types of SCLC as a basis to test the differential roles they ascribe based on in vitro analyses to NE and non-NE cells?

We appreciate the reviewer's valuable suggestion and have now included extensive *in vivo* data on both mouse and human SCLC tumors (**Fig. 2**, **Fig. 6**), as well as PDX models, and validated that the *in vivo* phenotypes correspond well to our *in vitro* analysis.

8. Finally, there is some concern about the low number of data points in several figures. N numbers are given only occasionally and mostly entail n of 3 technical replicates. What about biological replicates? In many experiments, the data points shown by individual dots on bar graphs (which are often too difficult to spot or to clearly decipher) appear to be very few – 2 or 3. By using appropriate means to make transparent and clear figures, these critical pieces of information can be made more clearly visible. Moreover, small n numbers need to be accounted for and ideally, substantiated in a more reliable data cohort.

We appreciate that the reviewer's concern and have updated the figure legends for better clarity. In addition, we would like to point out that in most of the experiments with n = 3 technical replicates were shown for each cell line, we have included analysis from multiple different cell lines as biological replicates to avoid potential idiosyncrasies specific to any one cell line.

Other comments:

(i) By the same token, data shown in Fig. 4f-k deserves to be in extended figures at best. Particularly the stainings for β tubulinIII and ChAT in panel 4h, at the magnification shown, do not reveal any insights. Moreover, it is surprising to see that putative cholinergic nerves show 100% identity with a marker for sensory nerves. The use of the term 'neurons' in the legend is incorrect – these are axons or efferents, not somata. Because the data do not show any contacts between the axons and the putative tumor cells, one does not understand what this figure achieves.

We have extensively edited the manuscript for better clarity and have now included high resolution staining of β tubulinIII and VChT with zoom in from different angles, to show

direct contacts between axons and SCLC cancer cells in **Fig. 2j, 2k** and **Extended Data Fig. 5g - 5j**.

In addition, we would like to clarify that we mostly saw overlapping of VAChT with the *pan-neuronal marker* (β tubulinIII, **Fig. 2j, 2k, Extended Data Fig. 5g - 5j**), but *not with the sensory nerve marker CGRP*, which we also used in some figures to mark SCLC cells (**Fig. 2d, Extended Data Fig. 5b**). We have also modified the text for better clarity:

(ii) Similarly, pCREB data and Sox/MCT4 data are purely qualitative. How many patients/mice show these findings? Do they stand up to statistical analyses?

We have updated the figures and figure legends with detailed quantifications.

p-CREB quantification in **Fig. 6 k - m**:

SOX1/MCT4 quantification in **Fig. 4m, 4n**:

- 1 Zhou, J. *et al.* GPR37 promotes colorectal cancer liver metastases by enhancing the glycolysis and histone lactylation via Hippo pathway. *Oncogene* **42**, 3319-3330, doi:10.1038/s41388-023-02841-0 (2023).
- 2 Li, L. *et al.* Identification of DHODH as a therapeutic target in small cell lung cancer. *Sci Transl Med* **11**, doi:10.1126/scitranslmed.aaw7852 (2019).
- 3 Yin, H. *et al.* Fusobacterium nucleatum promotes liver metastasis in colorectal cancer by regulating the hepatic immune niche and altering gut microbiota. *Aging (Albany NY)* **14**, 1941-1958, doi:10.18632/aging.203914 (2022).
- 4 Calbo, J. *et al.* A functional role for tumor cell heterogeneity in a mouse model of small cell lung cancer. *Cancer Cell* **19**, 244-256, doi:10.1016/j.ccr.2010.12.021 (2011).
- 5 Kwon, M. C. *et al.* Paracrine signaling between tumor subclones of mouse SCLC: a critical role of ETS transcription factor Pea3 in facilitating metastasis. *Genes Dev* **29**, 1587-1592, doi:10.1101/gad.262998.115 (2015).
- 6 Lim, J. S. *et al.* Intratumoural heterogeneity generated by Notch signalling promotes small-cell lung cancer. *Nature* **545**, 360-364, doi:10.1038/nature22323 (2017).

Response to Referees:

We thank the editor and the reviewers for their comments and feedback to improve this manuscript. We appreciate the reviewers' positive responses to our previous revision, indicating that *"the manuscript has benefitted significantly from the additions of so many control experiments and methodologies which present a tour-de-force of Cancer Neuroscience – relevant technologies today"*, *"these additional experiments and clarification have enhanced the impact of the work"*, *"this is a strong revision"*, and commenting that *"this is a very interesting study that will have impact on the field"*.

We have performed additional experiments to address further critiques and suggestions of the reviewer; we also significantly reconstructed the manuscript to provide better clarity to the story.

In particular, in the previous revision, a huge number of experiments had been conducted with many new figure panels added because we thought that we were requested to determine *if there were any synapses or synapse-like structures between cancer cells, and assess synaptic dependent cell-cell communication, especially by the EPSP in the (potential) post-synaptic cells, in order to compare our results with published literature in the Cancer Neuroscience field by providing experimental data whether those features of gliomas can be found in SCLC and understand the similarities and potential differences better*. Given that this might have been some misunderstanding in the first place, and that we appreciate the reviewer's new suggestion to reconstruct the manuscript and make it one coherent story, we decided to remove most of the figure panels related to addressing synaptic dependent cell-cell interactions and better focus on our major theme of how cancer cell intrinsic electrical activity promotes SCLC progression, and how the heterogeneous SCLC subpopulations cooperate via lactate shuttle, to alleviate the metabolic vulnerability of NE cells engendered by their electrical activity.

Major changes of the revised manuscript are summarized below:

1. Extensive edits to simplify the manuscript:

We have significantly trimmed the Main text from more than 7000 words down to around 4300 words and removed figure panels and discussions relevant to the synaptic dependent cell-cell communication to better focus the manuscript.

2. New figure panels: Fig. 2b, Ext. Data Fig. 3e, 3f

We have now included new patch clamp experiments demonstrating the electrophysiological effects after carbachol addition to NE cells.

3. Removed figure panels: Original Fig. 1i, 1l, 1m, 2i, Ext. Data Fig. 2r, 3, 5l, 5m

- a. EPSP-like activity (Fig. 1i, Ext. Data Fig. 2r)
- b. Retrograde rabies tracing (Fig. 1l, 1m, Fig. 2i, Ext. Data Fig. 3)
- c. Purinergic receptor expression and ATP responsive EP plots in SCLC cells (Ext. Data Fig. 5l, 5m)

4. Reorganization of figure panels:

We have reorganized and replaced some figure panels for better presentation and to facilitate data interpretation.

Referee #1 (Remarks to the Author):

The authors have responded well to my previous concerns. I apologize for a mis-wording that caused some confusion (..cholinergic signaling is the key driver of synaptic cell-cell communication: I did not mean “synaptic” here of course).

The manuscript has benefitted significantly from the additions of so many control experiments and methodologies which present a tour-de-force of Cancer Neuroscience – relevant technologies today. The authors also made changes that allowed to understand some of their concepts and conclusions better.

Having said that, the additions to my and the other reviewers’ concerns increased the problem that this manuscript is overall a difficult read, which is partly due to the extensive number of experiments performed, but also by data interpretation, presentation, ordering, and conclusions made. This manuscript continues to look like many studies combined into one paper. For this reviewer, it is really challenging to delineate a clear path out of this situation. One possibility would be to focus on the cancer cell intrinsic aspect and leave the (putative, see below) relation to neuron-cancer interactions for a future study.

We appreciate the reviewer’s concern and agree that this multidisciplinary manuscript is data heavy, and that better presentation and interpretation would be helpful. Indeed, this manuscript extensively covers research areas from **SCLC biology, cancer genetics, cancer metabolism, electrophysiology and cancer neuroscience**, making it really challenging to present all data within the figure limit, and explain all the background information and experimental details to non-expert readers from different fields within the word limit.

Despite the challenges, we appreciate the reviewer’s point of view and have significantly reconstructed the manuscript, leaving out figure panels that might be less pivotal to the storyline. On the other hand, we would also like to point out that these additions and changes were directly focused on addressing the concerns and points of all the referees collectively, which we and the other three reviewers all agreed that, by addressing all these reviewer comments, this paper has been significantly improved. While we also greatly value the power and beauty of simplicity, we do believe that the extensive data presented herein are important for this study, as the reviewer has also pointed out that *“The manuscript has benefitted significantly from the additions of so many control experiments and methodologies which present a tour-de-force of Cancer Neuroscience – relevant technologies today.”* Moreover, the complicated nature of SCLC biology also mandates thorough analysis, as requested by other reviewers in their previous comments. Without extensively addressing the well-known intra- and inter-tumoural heterogeneity of SCLC, it would be next to impossible for us to make any meaningful discovery and really advance the SCLC field.

The EPSP-like potentials in NE cells without cell contacts are puzzling, and in contrast to the many studies describing those potentials in tumor cells with similarly high input resistance and still only recorded in co-cultures with neurons but not in monocultures. Either the authors provide extensive ephys data what is exactly going on here (which single inward channels open, etc.), or they need to delete Fig. 1i and EDF 2r . Anyway, those two do not contribute to the story in a relevant way.

We thank the reviewer for pointing out that the EPSP figures are not directly relevant to the story and suggested revising the manuscript by deleting **Fig. 1i** and **EDF 2r**. These experiments were included in the previous revision due to the potential misunderstanding that we had to address the synaptic cell-cell interaction by showing EPSP-like potentials in NE cells. We apologize for the confusion and have removed all the relevant figures from the current revision exactly as the reviewer suggested. Furthermore, we would like to note here that **we certainly do not intend to imply anything about other studies involving cocultures of other tumour cell types with neurons**, which, on the basis of observing spontaneous EPSP-like events, have concluded that there are likely EPSPs in tumour cells. **Importantly, we are only discussing the possibility of transmission amongst tumour cells, not between neurons and tumour cells.**

In addition, we would like to point out that inward channels of just a few pA amplitude - which we commonly observe in cell-attached mode - when injected into a cell with passive membrane parameters as measured, will produce events closely matching the “EPSP”-like events observed in current-clamp, both in amplitude and shape, which could be found in these following papers demonstrating how single channel openings of just a couple of pA or less can drive action potential firing in small (high input resistance) neuroendocrine cells, going back to the classic paper of Fenwick, Marty & Neher (1982) A patch-clamp study of bovine chromaffin cells and of their sensitivity to acetylcholine, *J. Physiol.* 331:577-597. See their Fig. 3B for examples of stimulation of action potentials by openings of single AChR channels.

See also:

Johansson and Århem (1994) Single-channel currents trigger action potentials in small cultured hippocampal neurons, *PNAS* 91(5):1761-1765

and

Chow & White (1996) Spontaneous action potentials due to channel fluctuations. *Biophys. J.* 71:3013-3021.

It remains unclear what the tumor biological effect of the AP-generated intercellular Ca transients is. A diffusible factor is plausible, but do the authors suggest this is indeed Ach (this is at least how the story is presented)? I cannot see how, and evidence for this is missing. It is much more likely that these are extracellular Ca waves mediated by e.g. ATP/purinergic receptors, a well established mechanism of Ca wave propagation in unconnected cells. The authors should test this. In fact, later on in the manuscript, they are making first experiments into this direction (Extended Data F 5l,m), but only in the context of NE depolarization. Thus, I would also recommend to re-order the presentation of these datasets, too.

We agree with the reviewer that the mechanism of propagation of these waves in NE cells is not completely clear, and we also assume that the referee does not actually mean “extracellular Ca waves” – these are waves of *intracellular calcium transients*, propagated extracellularly. On the other hand, we would like to point out that we have done our best to answer the reviewer’s previous questions (cited below) in our last revision:

I would suggest to discuss the preprint reporting action potentials recorded in a new population of glioma cells with a neuronal profile (Curry, et al. bioRxiv 2023);

and also the work about neural-like networks of gliomas and potentially also extracranial cancer entities that also communicates with Ca transients and where neurodevelopmental mechanisms are hijacked, including for generation of an autonomous rhythmic activity (Osswald et al Nature 2015; Hausmann et al Nature 2023). Importantly, while they nicely demonstrate evidence for the former, **the authors should provide experimental data whether those latter features of gliomas can be found in SCLC, too.....**Along these lines, the well-known axon-like protrusions of SCLC cells that the authors cite: **do they resemble tumor microtubes (TMs) of incurable glioma types? Do these protrusions also interconnect SCLC cells to one communicating syncytium, potentially involving NE and non-NE cells and thus providing an explanation how activity (Ca transients?) generated in one cancer cell subpopulation can govern the entire tumor? Does an anatomical network allow electrical auto-stimulation of the entire tumor?**

All our answers have been documented in the previous version of our revised manuscript:

1. *The mechanism of calcium wave propagation in SCLC is completely different from those reported in glioma.*
2. *The axon-like protrusions in SCLC are different from the TMs found in glioma.*
3. *Diffusion factors, instead of synaptic-dependent mechanisms, are an explanation for the wave propagation in SCLC.*
4. *Electrical stimulation can evoke wave propagation in cultured SCLC cells.*

Importantly, we would like to point out that this is a story focused on **SCLC biology**, which is a **neuroendocrine cancer**, and is most likely very different from the biology of glioma. We decided to focus on acetylcholine pathway because it is a very reasonable candidate and an educated guess, due to its well-known role in SCLC biology. Please see below some examples of the role of ACh pathway both in normal PNECs and in SCLC:

Song et al, Cancer Research (2003) 63: 214, Acetylcholine is synthesized by and acts as an autocrine growth factor for small cell lung carcinoma.

Please also see illustration of the Ach pathway in PNEC/SCLC as shown below by Shuller, (2009) Nat Rev Cancer, 9(3):195-205 and Li et al (2023) Molecules 28(3), 1139:

REDACTED

We certainly do not wish to give the impression that we have solved the mechanism of these waves, which would have to be the subject of very extensive further studies; on the other hand, we would also like to point out that the reviewer has already commented that *the extensive number of experiments performed has made our manuscript difficult to read*, and that our manuscript *looks like many studies combined into one paper*. Therefore, we would like to refrain from further detailed investigation of potential additional molecular mechanisms of calcium wave propagation in SCLC NE cells, especially a plausible **ATP signaling pathway, which, while relevant for astrocytes, a cell-of-origin of glioma, may be less relevant in SCLC**.

We consider the key novelties of the manuscript lie in the following findings:

1. *Detailed EP analysis in heterogeneous SCLC subtypes.*
2. *Different metabolic phenotypes and vulnerabilities between excitable and non-excitable cancer types.*
3. *Identified metabolic support as a mechanism for the functional ITH in SCLC.*
4. *Cancer cell intrinsic electrical activity promotes SCLC progression.*

Besides those points, further in-depth interrogation of molecular mechanisms underlying calcium wave propagation in NE cells would be beyond the scope of our current study. We only conclude that: **a) these are definitely mediated by a diffusible factor or factors (because they cross cell-free gaps), and b) probably multiple factors to different extents**, including, but not necessarily limited to:

- ATP, because it is expected to be released from any depolarization-induced vesicular release, and causes calcium transients when applied exogenously, and in a couple of experiments, we have observed that spontaneous waves are reduced by applying extracellular apyrase;
- ACh because it is known to be secreted by SCLC cells and also causes calcium transients, as well as inward channel currents and action potentials when applied exogenously in the form of the stable cholinergic agonist CCh;
- K⁺ ions, which are obligatorily released by electrical activity and will depolarise any cell that has resting potassium conductance.

To avoid potential confusion, we decided to remove all ATP-related figures and discussions, but highlight in our discussion that multiple mechanisms might play a role in calcium wave propagation:

*[Pages 4, 5] To explore **possible** diffusible factors that could trigger the electrical activity of NE cells, we took a **candidate approach** and focused on **acetylcholine (ACh)**, since the **cholinergic signaling pathway has been extensively characterized and shown to be crucially involved in SCLC tumorigenesis**^{37,38}.*

37 Song, P. et al. Acetylcholine is synthesized by and acts as an autocrine growth factor for small cell lung carcinoma. *Cancer Res* 63, 214-221 (2003).

38 Song, P. et al. M3 muscarinic receptor antagonists inhibit small cell lung carcinoma growth and mitogen-activated protein kinase phosphorylation induced by acetylcholine secretion. *Cancer Res* 67, 3936-3944 (2007).

<https://doi.org/10.1158/0008-5472.CAN-06-2484>

*[Page 11] We also identified nAChR as an ion channel which participates in the initiation and propagation of the electrical activity in SCLC. Notably, SCLC is almost invariably associated with smoking in patients⁷, and tobacco nicotine is another potent agonist for nAChR. In addition, calcium activity in PNEC has recently been shown to be activated by various mechanical and chemical stimuli²³, many of which become pronounced within the tumor microenvironment⁷²⁻⁷⁴. Therefore, **as SCLC progresses, multiple mechanisms may be co-opted to stimulate and sustain the intrinsic electrical activity of SCLC.***

23 Seeholzer, L. F. & Julius, D. Neuroendocrine cells initiate protective upper airway reflexes. *Science* 384, 295-301 (2024).
<https://doi.org:10.1126/science.adh5483>

72 Boedtkjer, E. & Pedersen, S. F. The Acidic Tumor Microenvironment as a Driver of Cancer. *Annu Rev Physiol* 82, 103-126 (2020).
<https://doi.org:10.1146/annurev-physiol-021119-034627>

73 Eil, R. et al. Ionic immune suppression within the tumour microenvironment limits T cell effector function. *Nature* 537, 539-543 (2016).
<https://doi.org:10.1038/nature19364>

74 Shieh, A. C. Biomechanical forces shape the tumor microenvironment. *Ann Biomed Eng* 39, 1379-1389 (2011). <https://doi.org:10.1007/s10439-011-0252-2>

Moreover, do the non-NE cells also respond to the activation from NE cells, another interesting mechanism how both cancer cell subpopulations interact?

We have shown that non-NE cells are not electrically active. Other types of “response”, including changes in gene expression and secretome, have been shown in previous SCLC papers, including the **Calbo et al (2011) *Cancer Cell*.15;19(2):244-56; Kwon et al (2015) *Genes Dev.* 1;29(15):1587-92; Lim et al (2017) *Nature* 18;545(7654):360-364**. To properly address how non-NE cells may respond to the activation from NE cells via additional mechanisms would require another full manuscript to do so, and is beyond the scope of the current study.

Fig. 2a,b: why are no AP's recorded here in NE cells? Wouldn't this be expected after strong (like with CCh) Ach receptor stimulation?

We value to reviewer's concern and have performed additional recordings. The **current-clamp experiments** showing AP after CCh treatment are shown here and in **Fig. 2b**:

Moreover, new voltage-clamp experiments in NE cells after CCh treatment are shown in Fig. 2b and Extended Data Fig. 3d - 3f.

[Page 5]...application of a stable cholinergic agonist carbachol (CCh) elicited widespread calcium activity in NE cell cultures (Fig. 2b, Extended Data Fig. 3d - 3f, Extended Data Movie 4), and the evoked single channel openings demonstrated amplitudes and lifetimes consistent with those reported for nAChRs (Fig. 2b).

To prove that “classical synapses” exists between sensory neurons and SCLC cells, the authors need to go beyond rabies virus tracing, which is not a definite proof of synaptic connections but can simply show spatial proximity. EM and electrophysiology of acute sections of the tumor tissue is needed. Otherwise (since this is not a key aspect of this manuscript) they would need to discuss this data accordingly and phrase it much more cautiously.

We agree with the reviewer that EM and electrophysiology would be important to validate that the interaction between sensory neurons and SCLC cells is indeed a classical synapse. Such an in-depth characterisation of these potential synapses could be further addressed in a more specialised paper. Therefore, to avoid overcomplicating our work, we decided to phrase this interaction more cautiously and remove the all the rabies virus tracing experiments.

*[Page 5] The nerve axons were not only detected at the tumor periphery but were tightly woven into the core of the lesion, where we also observed **structures typical of terminal boutons, suggesting synapse formation at nerve endings** (Fig. 2j, 2k, Extended Data Fig. 4i, 4j, Extended Data Movie 6).*

Fig. 2m: the authors should clarify better in the text that the beta3 tubulin staining is not showing neuronal structures here but is a tumor cell-specific staining pattern.

We agree with the reviewer that we could further clarify in the text, and have made editorial changes as follows. We hope that by making these changes, it would be clearer now that beta 3 tubulin shows the neuronal structures as well and is not tumor cell-specific (please also see Fig. 2g, lower middle and right panels).

*[Page 5] ...the **overall innervation (identified by a pan-neuronal marker β 3-tubulin)** of hyperproliferative NEB/early SCLC lesions...dramatically increased after tumor induction ...concomitantly, **β 3-tubulin became highly expressed by cancer cells in these large tumors** (Fig. 2l)*

It is not clear why the authors assume that the Ca²⁺ activity measured by GCaMP6 signals in fresh tumor-bearing lung slices is an indication of “spontaneous” (i.e. tumor-intrinsic) electrical activity. It could be that those are generated by neuronal input, synaptic or otherwise, by the remaining (if transected) nerve endings in the tissue, or even neuronal somata that can be newly formed in tumor environments (see the works of C. Magnon and others).

We agree with the reviewer that we indeed have not exhausted all possibilities to exclude that the calcium activity in lung slices might be induced by neuronal input in any form. Therefore, we **have made editorial changes and removed these potentially controversial statements.**

[Page 5] To further assess calcium activity in autochthonous SCLC tumors, we generated the PRP130-Salsa6f (tdTomato-V5-GCaMP6f) animal, which expressed

*both the tdTomato lineage marker and calcium reporter GCaMP6f in SCLC cells (Fig. 2m). Indeed, ex vivo imaging on fresh lung slices revealed **spontaneous propagating calcium waves in SCLC tumors** (Fig. 2n, Extended Data Movie 8).*

On the other hand, given the 2D and 3D *mono-cultures of SCLC cells* also generate spontaneous calcium waves *without nerve inputs* (Ext. Data Movies 1-3), these results indicate that **external innervation was not necessary for calcium wave propagation in SCLC.**

Along these lines, it would really strengthen one major conclusion of this manuscript if the early tumor growth-promoting influences of neuronal innervation (namely cholinergic) that is not relevant for later tumor growth would have been supported by any data stemming from interventional experiments: in vivo growth differences under early vs late vagotomy, for example, or pharmacological or genetic interference. All data provided in this direction is descriptive. As it is, this concept remains hypothetical (albeit an interesting and even plausible hypothesis). When no more data in this direction is provided, the authors should reconsider the last sentence of the abstract, for example.

We appreciate the reviewer's suggestion and we reconsidered **the last sentence of the abstract as follows:**

*[Page 2] Lastly, we observe drastic changes in the innervation landscape during SCLC progression, which **might reflect a transition from the initial dependency of early-stage SCLC on external factors, namely cholinergic innervation in the tumor microenvironment, coincide with** increase intratumoral heterogeneity and elevated neuronal protein expression in SCLC cells, suggesting an induction of ~~to~~ a tumor autonomous vicious cycle, driven by cancer cell-intrinsic electrical activity, which confers long-term tumorigenic capability and metastatic potential.*

In the meantime, we would also like to point out that the early/late vagotomy experiment which the reviewer suggested, **has already been performed in an interesting preprint that we cited in the discussion of our previous revision:**

[Lines 570 - 573] These observations are consistent with a previous report showing that vagotomy before tumor initiation in the PRP130 model significantly suppressed SCLC development, while when performed at late stage, demonstrated an opposite trend to promote SCLC progression⁸⁴.

**84 Savchuk, S. et al. Neuronal-Activity Dependent Mechanisms of Small Cell Lung Cancer Progression. bioRxiv (2023).
<https://doi.org:10.1101/2023.01.19.524430>**

Below, we show their data figures and legends copied from bioRxiv:

REDACTED

[Legend:] Vagotomies or sham surgeries were performed approximately two months after the intratracheal administration of the Adeno-CMV-Cre vector but prior to tumor initiation. Animals were then followed for the next 25 weeks using in vivo bioluminescent imaging... Finally, as overall tumor burden was greatly reduced by vagotomy, we observed a striking survival benefit for the mice that had been denervated, with all denervated mice surviving the full length of the experiment compared to a median survival of 16 weeks for sham manipulated mice (Fig. 4k).

REDACTED

[Legend:] In contrast, when vagotomy was performed after tumor initiation in an independent cohort of animals, such reduction of tumor burden and survival benefit were not observed (Extended Data Fig. 8e-f).

In that paper, **the exact experiments have been performed in the same mouse model we use in this manuscript (PRP130), and their observations are well in line with our hypothesis.** Considering that the information is publicly available, to conform with the ethical rule of 3R in animal research, we think that it would be ethically inappropriate to repeat extensive animal experiments that have already demonstrated results that we expected. We have now further clarified that in our discussion:

[Page 11] Our data from different stages of mSCLC progression suggested that cholinergic innervation might be crucial for SCLC initiation but becomes dispensable in fully formed SCLC. This hypothesis is supported by evidence from an independent study, which shows that vagotomy before tumor initiation in the PRP130 model significantly suppressed SCLC development, while when performed at a late stage, demonstrated an opposite trend to promote SCLC progression⁷⁶

**76 Savchuk, S. et al. Neuronal-Activity Dependent Mechanisms of Small Cell Lung Cancer Progression. *bioRxiv* (2023).
<https://doi.org:10.1101/2023.01.19.524430>.**

Referee #2 (Remarks to the Author):

I take this opportunity to thank the authors for performing an extensive amount of additional experiments to address the concerns raised by myself and the other reviewers. These additional experiments and clarification have enhanced the impact of the work. Based on this, the paper is acceptable for publication.

Referee #3 (Remarks to the Author):

This is a strong revision and the authors have addressed my critiques.

Referee #4 (Remarks to the Author):

The authors have adequately responded to my questions and comments. This is a very interesting study that will have impact on the field.

Previous comments from Referee #1 (Remarks to the Author):

In this manuscript, the authors describe a really fascinating story of an electrically active, neuronal-like, AP-generating SCLC cell subpopulation (NE cells) that collaborates with non-(active) NE cells to promote malignancy; in turn, the non-NE cells provide metabolic support for the NE cells to meet their high energy demand associated with this neuronal-like behavior (like astrocytes with neurons).

In general, the concept developed here is exciting, raising the possibilities of tumor cell-autonomous neuronal features to a new level, and linking the field of cancer neuroscience to cancer metabolism. The abstract and introduction read great and very convincing. The data provided however does often not live up to the expectations raised. It is not sufficiently experimentally addressed how exactly the NE and non-NE cells collaborate; whether this is really reflecting and impacting intra- (rather than inter-) tumor heterogeneity; how this influences important parameters of tumor biology, which are by and large missing (other than liver weight, see below, and some tumor cell number quantifications) – what about tumor burden throughout the organism, cancer cell proliferation, apoptosis, animal survival etc; and important control experiments are missing, too.

The following points should be addressed:

Major points:

The authors should discuss their findings more in the context of what is known about cancer cell intrinsic electrical activity in other cancers, including glioma. This appears particularly important, since the emerging fields of CNS and extracranial Cancer Neuroscience need to understand the similarities and potential differences better. In this respect I would suggest to discuss the preprint reporting action potentials recorded in a new population of glioma cells with a neuronal profile (Curry, et al. bioRxiv 2023); and also the work about neural-like networks of gliomas and potentially also extracranial cancer entities that also communicates with Ca transients and where neurodevelopmental mechanisms are hijacked, including for generation of an autonomous rhythmic activity (Osswald et al Nature 2015; Hausmann et al Nature 2023). Importantly, while they nicely demonstrate evidence for the former, the authors should provide experimental data whether those latter features of gliomas can be found in SCLC, too. ED Movie 1 could be a strong hint that they might. In any case, they should discuss the implications. Along these lines, the well-known axon-like protrusions of SCLC cells that the authors cite: do they resemble tumor microtubules (TMs) of incurable glioma types? Do these protrusions also interconnect SCLC cells to one communicating syncytium, potentially involving NE and non-NE cells and thus providing an explanation how activity (Ca transients?) generated in one cancer cell subpopulation can govern the entire tumor? Does an anatomical network allow electrical auto-stimulation of the entire tumor? And, as important, is this a biological mechanism how the metabolic symbiosis between NE and non-NE is achieved, complementing the paracrine/autocrine mechanism that is also plausibly supported by the data of Fig. 2,3? This could (and probably should) be experimentally

addressed, too, since these questions have broad implications for Cancer Neuroscience and would significantly strengthen this manuscript.

To establish the difference of NE vs non-NE cells with respect to generating APs, the authors establish an experimental setup in which they compare a (limited) number of established cell lines that are more NE vs less (non-) NE. The exciting point of how these differences influence intratumoral heterogeneity would be made much more convincing if they sort and then electrophysiologically compare NE vs non-NE cells from the SAME patient (cell line), ideally by using primary patient-derived cells. Similarly, this paradigm should wherever possible also be applied to other parts of this manuscript.

Similarly, it would be relevant to learn about differential effects of ChR expression in NE vs non-NE cells, preferably those coming from the same line/patient. Is optogenetically-induced cancer cell excitation in non-NE cells possible, and if (unexpectedly) so, can it transform non-NE cells to NE cells on a transcriptional and functional level (“rescue experiment”)? Otherwise this would be a good control experiment. In this context: Fig. 1h: adequate control experiments (light stimulation without ChR) are missing.

Of similar importance, blue light stimulation of ChR expressing cells is also simulating what happens on the postsynaptic side (induction of EPSCs).

Last but not least, are there any synapses or synapse-like structures between cancer cells? Between NE cells, or – potentially even most interesting in light of the general concept of this study – between NE cells (presynaptic) and non-NE cells (postsynaptic)?

With respect to TTX, one important fundamental experiment is missing: does TTX influence AP generation in SCLC NE cells?

Another crucial experiment that is missing is the proof that TTX removal is indeed increasing hyperactivity/firing frequency in the SCLC cells, as shown for some nonmalignant neuronal cells.

The concept of particularly high energy demand of AP-firing SCLCs is intriguing. However, I am struggling to understand whether the authors claim that the OXPHOS pathway is particularly relevant here, or the lactate shuttle (“metabolic symbiosis”) mechanism. I am also missing data on differential relevance of glycolysis (another way of ATP production) between NE vs non-NE cells.

The authors claim that cholinergic signaling is the key driver of synaptic cell-cell communication between SCLC cells. The data to support this (other than cited previous work) is however very limited and seems to be limited to Fig. 4a – one single recording without statistics. It would be very important to substantially strengthen this part of the manuscript by providing much more robust data on the structural, molecular and electrophysiological level.

The data provided in Fig. 4 (and else) is not really convincingly supporting one key hypothesis (if not conclusion) of the study: that a cancer-autonomous electrical self-stimulation is making SCLC independent from direct neuronal input during cancer progression. In fact, everything cited in favor for this concept is no direct proof, rather circumstantial. One way to address this interesting hypothesis would be to investigate different metastatic sites: are liver metastasis (where the main tumor biology – findings are made) are particularly hypo/-non-innervated, which makes NE cells particularly liver metastasis-proficient – while other metastatic (or primary) sites are much better innervated, which makes the electrical activity of NE cells less important? This should

really be addressed in mouse studies covering multiple sites of tumor growth, characterizing innervation patterns and at the same time recording tumor growth kinetics and finally overall animal survival.

Minor points:

Abstract: ...similar to astrocytes...: correct, but this applies to any other non-neuronal cell; the authors might want to stress this point that also implies an even wider relevance of their finding.

Ext Data Fig 1h, but also Fig. 1l and EDF 3e etc.: the quantification of "metastasis" is not convincing; only liver size and weight seems to be determined. EDF1h: What is the color code of the graph?

Point-to-point response:

The authors have again responded very well to my points and concerns. Although the manuscript remains a difficult read, that might be the best possible.

All in all, this is a strong manuscript that will definitely contribute significantly to our understanding of SCC pathobiology and will also contribute to the field of Cancer Neuroscience in a very meaningful way.

We thank the reviewer for their compliment and constructive advice.

If have only three remaining points:

1. It is certainly not beyond the scope of the manuscript to test whether ATP or ACh or K⁺ ions are responsible for the intercellular Ca waves that are clearly conducted extracellularly. There are many drugs available that can easily be used to interfere with these three possibilities in in vitro experiments. Since this is not an unimportant point - how does the "tumor organism" of these two major cell types function? - I would still find it very nice data to have in the manuscript. However, if this is completely beyond the capabilities or would require many months for the authors (...?), this is not mandatory for manuscript acceptance in my view.

We completely agree with the reviewer that understanding whether ATP or Ach or K⁺ ions are responsible for the intracellular Ca waves are crucial for SCLC biology. On the other hand, we believe that it would be appropriate to address the important questions in future studies, so that the roles of these molecules could be interrogated in depth. Therefore, we appreciate the reviewer's understanding that this would take a huge amount of effort and is not mandatory for this current story.

2. Mechanism of how both cancer cell subpopulations interact: I was more thinking into the direction of intercellular Ca waves, too. Any data on this?

We agree that the cell-cell interaction between SCLC subpopulation is intriguing, whether it's dependent on interacellular Ca waves or not. Importantly, we have been focusing on metabolic interactions, so many key experiments (e.g. Fig.3m, 3n, 3o) were conducted with **conditioned medium from non-NE cells**, without the physical presence of non-NE cells, which excluded the possibility of interacellular Ca waves. That being said, we do think that it is crucial to study the direct cell-αcell interaction between these population as well; on the other hand, our data showed that non-NE cells are not excitable, could not fire action potentials, do not express the AChR as NE cells do, and unlike glioma, do not form microtubules. Therefore, we think that it is unlikely that intercellular Ca waves would play a major role between NE and non-NE cells. Considering the scope of this manuscript, which is mostly focused on exchange of secreted metabolites between these cell types, we think that these interesting questions of intercellular Ca waves would be appropriate to leave for future studies.

3. please rephrase: ...of termin boutons, which might point towards synapse formation at nerve endings (or a similar cautious phrasing).

We appreciate the reviewer's suggestion and will modify accordingly.